# Foliation-Based Approach to Quantum Gravity and Applications to Astrophysics

**Inyong Park** 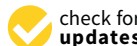

Department of Applied Mathematics, Philander Smith College, Little Rock, AR 72223, USA;
inyongpark05@gmail.com

**Abstract:** The recently proposed holography-inspired approach to quantum gravity is reviewed and expanded. The approach is based on the foliation of the background spacetime and reduction of the offshell states to the physical states. Careful attention is paid to the boundary conditions. It is noted that the outstanding problems such as the cosmological constant problem and black hole information can be tackled from the common thread of the quantized gravity. One-loop renormalization of the coupling constants and the beta function analysis are illustrated. Active galactic nuclei and gravitational waves are discussed as the potential applications of the present quantization scheme to astrophysics.

**Keywords:** quantum gravity; renormalization; ADM formalism; foliation; dimensional reduction

## 1. Introduction

The last several decades have witnessed multiple major advances in mathematical and theoretical physics. In particular, some far-reaching ideas have been proposed in string theory, leading to synthesized and synergetic results in the other areas of theoretical physics. It has recently come to light that the set of the new ideas is potent enough to penetrate several longstanding problems that often baffled the field researchers in spite of progress made. These problems include quantization of gravity, the cosmological constant, and black hole information. A remarkable picture that has emerged is that these problems may not be independent but may well have a common thread with a promise of solutions for all. The common thread is the holography-inspired foliation-based quantization of gravity.

The study of quantization of gravity has a prolonged history (see, e.g., [1–18] for reviews), yielding several approaches with which we will make brief comparisons to the present approach. As well known, non-gravitational gauge theories such as a Yang–Mills theory were successfully quantized in the seventies. Since a gravity theory also takes the form of a gauge theory with diffeomorphism being the gauge symmetry, one may wonder what the difference between the two theories is that makes the quantization of the former so much less straightforward. Could it be, for instance, that although both the gravitational and non-gravitational gauge theories have infinite dimensional Lie symmetry groups, the diffeomorphism symmetry of a gravity theory belongs to the spacetime, whereas the gauge symmetry of a non-gravitational gauge theory—acting only on the abstract field space—does not? Certainly a gravitational system is more complex in many ways than a non-gravitational one. The real reason for the difficulty in the gravity quantization has turned out, as we will review, to be the large amount of gauge symmetry, and it's been realized that the diffeomorphism symmetry can be fixed in a manner that accomplishes the long-sought renormalizability of gravity (in its physical sector[1]) [19–21].

---

[1]    More on this in Section 2.

One of the early attempts to quantize gravity was the covariant approach in which attempts were made to replicate the success of non-gravitational gauge theories. There, the offshell renormalizability, i.e., the renormalizability of the Green's functions, was undertaken just as in non-gravitational gauge theories. As is well known, the endeavor led to non-renormalizability instead. Another direction, the so-called canonical quantization, employed the canonical Hamiltonian formalism. In particular, a 3+1 splitting of the spacetime dimensions was introduced and the Dirac bracket formalism for constraints was used. One of the obstacles in this approach was how to deal with the constraints associated with the nondynamical fields. Various obstructions in these early attempts motivated another, more radical and ambitious approach, loop quantum gravity (LQG) [9,10], where the basic degrees of freedom of the theory were taken to be the so-called the loop variables. Although the set of the ideas of LQG is attractive, there remain some challenges and ongoing problems.

As stated, there have been various critical developments in theoretical and mathematical physics in the last several decades. One of the most critical developments was AdS/CFT correspondence, which conjectures duality between non-gravitational and gravitational theories. Another was the steadier progress made in differential geometry and foliation theory. In fact, some of the crucial results in this branch of mathematics that are critical for the present work were obtained as early as the seventies in the mathematical literature. In spite of these developments ,these results have not, until recently, been assembled into arsenals for tackling the longstanding problems. In the newly proposed quantization some of the old techniques have been put together with the recent ones to provide the missing fulcrum for quantizing gravity in a renormalizable manner.

Among other developments, it was holography and AdS/CFT correspondence that had provided the strongest motivation and momentum to the present approach of the quantization. The correspondence can be approached from the fact that the branes in string/M theory admit two different descriptions, dubbed in [22] as the fundamental and solitonic descriptions. Whereas it was the symmetries of the dual theories that played a central role in motivating the correspondence in the original proposal, our focus has been the actual dualization procedure itself. As we will further comment on in the next section, investigation of the actual dualization procedure fills the gaps in our understanding of AdS/CFT-type dualities in the literature. At the same time it provides one of the philosophical pillars of the present quantization scheme.

Gravitational theories harbor some of the most fascinating aspects of Nature, such as holography [23,24]. As demonstrated by Einstein's formulation of general relativity, geometry makes the subject richer and deeper than it would have been otherwise. One naturally expects that geometry will also play a key role in quantization; mathematical foliation theory provides the needed geometrical ingredient. As will be shown, holography can be explicitly realized by appropriately fixing the diffeomorphism in the foliation-based setup of the Arnowitt–Deser–Misner (ADM) formalism. Once the diffeomorphism symmetry is appropriately fixed, it brings projection of the physical states onto a 3D "holographic screen" in the asymptotic region and the reduction[2] provides the missing leverage for the renormalizability.

The fact that the physical states are associated with the hypersurface in the boundary region brings along the issue of boundary conditions: for quantization and various quantum-level analyses, a systematic analysis of the boundary conditions is highly desirable. As we will see, non-Dirichlet boundary conditions, in particular Neumann-type boundary conditions, will play an important role in the complete quantum description of the system. The Neumann-type boundary conditions are the ones that allow construction of the reduced form of the action.

On more technical side, the present approach to the 1PI effective action calculation is a direct Feynman diagrammatic one and we employ what we call the refined background field method

---

[2]　As stressed in [19], the reduced theory is not a genuine 3D theory, but still a 4D theory whose dynamics can be described through the hypersurface.

(BFM) [25,26]. The Feynman diagram techniques were, of course, employed in the early literature as well. However, they played the subsidiary role of computing the coefficients of the counterterms with the forms of the counterterms predetermined by dimensional analysis and covariance, a procedure that cannot be fully justified due to the employment of the "traceful" propagator—as we will review. Our refined BFM allows one to directly compute, as one normally does in non-gravitational theories, the counterterms, including their coefficients.[3] (We employ the "traceless" propagator and the values of the coefficients computed this way are different in general from those of the earlier works.) It also leads, as a byproduct, to a simple solution of the long-known gauge-dependence issue [27–34] in computing the 1PI effective action.

　　　The remainder of the paper is organized as follows. Section 2 is devoted to an overview of what has been done differently to achieve the renormalizability of the physical sector. We also highlight salient features of the present quantization procedure. The focus of Section 3 is the reduction of the offshell states to the physical states. The ADM formalism [35] is employed; the shift vector and lapse function are gauge-fixed, then the constraints associated are solved, leading to the reduction. In Section 4 we analyze the boundary conditions and boundary dynamics. We review some of the recent developments in boundary conditions and dynamics in gravitational theories. It turns out that the complementary roles of the active and passive gauge transformations are highly instructive. Natural foliations and boundary conditions come with the coordinates system adapted to the reference frame of the observer and the pertinence of different reference frames implies that the Hilbert space must be enlarged by incorporating the states of different boundary conditions. In Section 5 we present a detailed analysis of one-loop counterterms for relatively simple two-point diagrams. The renormalization involves a field redefinition, which is not necessary in the non-gravitational renormalizable theories. We carry out renormalization of the coupling constants and, in particular, the beta function analysis of the vector coupling constant for an illustration. The one-loop renormalizability is achieved at the offshell level with the use of the Euler–Gauss–Bonnet identity given in (1). As for the two- and higher-loop-order renormalizability, no such identity is available and one must therefore rely on the reduction of the offshell states to the physical states. The two- and higher-loop renormalizability in an asymptotically flat background can be achieved by following the outlines presented in [36], a task postponed to the future. In Section 6, we discuss several potential astrophysical applications of the new quantization scheme. There is accumulating evidence that an infalling observer will in general encounter trans-Planckian radiation from near the event horizon of a time-dependent black hole. We propose that the phenomenon should be the mechanism for the large energy radiation of active galactic nuclei. Also, incident waves may well produce reflected waves if the horizon is not a featureless place as conventionally conceived. We set the stage for the future exploration of the gravitational waves with the boundary conditions different from that of the conventional perfect infall.

　　　Throughout we try to present more universal types of issues; for finer and more detailed issues, one may consult the original papers in which the issues were analyzed in greater detail.

## 2. Overview of "What Has Been Done Differently"

　　　Before embarking on the long and involved enterprise starting with Section 3, it may be useful to have an overview of some of the hallmark features of the present quantization procedure. Through the intensive efforts of the seventies and eighties, non-renormalizability of the Green's functions was established. The core rationale behind the non-renormalizability was the following. To be specific, let us consider gravity without matter. The crux of the problem leading to the non-renormalizability was the appearance of the Riemann tensor, as opposed to the Ricci tensor or Ricci scalar, in computation

---

[3]　Such a direct calculation is normally done in non-gravitational theories. However, it appears that the same procedure has not been implemented in a gravitational theory. The reason is presumably that such an attempt would have run into non-covariance, as we will review below.

of the counterterms to the ultraviolet divergences. If one computes the counterterms to, say, various one-loop divergences, one sees that they are required to take the forms of $R^2$, $R_{\mu\nu}R^{\mu\nu}$, or $R_{\mu\nu\rho\sigma}R^{\mu\nu\rho\sigma}$. One can show that with an appropriate metric field redefinition (a la 't Hooft) the first two types of counterterms can be absorbed into the Einstein–Hilbert term. As a matter of fact, the Riemann tensor square term, $R_{\mu\nu\rho\sigma}R^{\mu\nu\rho\sigma}$, can also itself be absorbed because (in 4D) it can be replaced by $R_{\mu\nu}R^{\mu\nu}$ and $R^2$ through the following Euler–Gauss–Bonnet topological identity,

$$R_{\mu\nu\rho\sigma}R^{\mu\nu\rho\sigma} - 4R_{\mu\nu}R^{\mu\nu} + R^2 = \text{total derivative}, \tag{1}$$

and this is why, e.g., the Einstein gravity was declared one-loop renormalizable. However, let us suppose for the sake of the argument that such an identity were not available. (Indeed, for the two-loop counterterms involving the Riemann tensor, no such analogous identity is available.) Then the $R_{\mu\nu\rho\sigma}R^{\mu\nu\rho\sigma}$ counterterm could not be removed. Things get worse and worse as the number of loops increases; the proliferation of the Riemann tensor-containing counterterms spins things out of control and the theory loses predictive power due to the infinite number of required counterterms.

Here holography comes to the rescue. What we have shown in a series of recent works is that the physical sector of the theory is associated with a 3D hypersurface—often called the holographic screen—in the boundary region (see Figure 1). This implies that for the renormalization of the S-matrix one can replace the Riemann tensor in the 1PI action essentially by the 3D Riemann tensor (more details and references can be found in the main body). As is well known, the 3D Riemann tensor can be expressed in terms of the 3D Ricci tensor, Ricci scalar, and metric, thus leading to the renormalizability[4] through a metric field redefinition a la 't Hooft [38]. To recap, instead of the more ambitious goal of the offshell renormalizability of the Green's function, one can aim at the more moderate goal of renormalizability of the S-matrix (which nevertheless is experimentally uncompromising since only the physical states can be measured) for which the problematic Riemann tensor can be expressed in terms of the "benign" Ricci tensor and Ricci scalar.

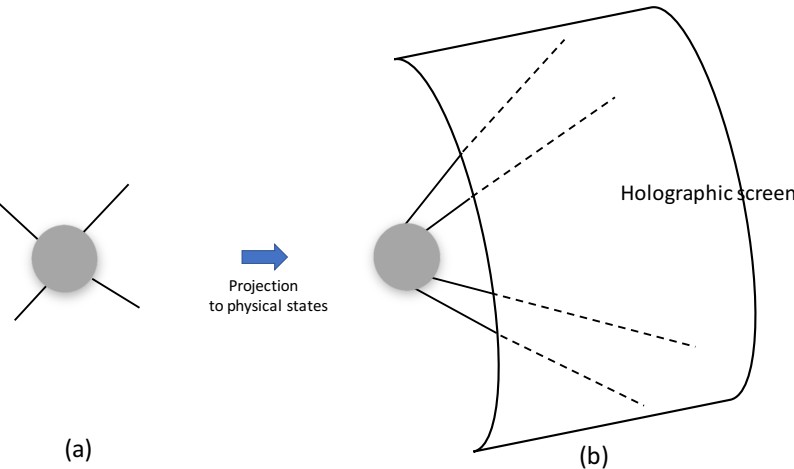

**Figure 1.** (**a**) Scattering of offshell states; (**b**) scattering of physical states.

Thus, establishing the reduction of the physical states is quite central to the present quantization scheme, making it the focus of Section 3. We employ the ADM formalism for the reduction. There are

---

[4] Although the present scheme of the "holographic quantization" may sound similar to the well-known idea of holographic renormalization [37], the two ideas are not directly related. While in the holographic renormalization one usually studies the renormalization of the composite operators of the boundary theory (assisted by the classical bulk theory), the focus of the present holographic quantization is the renormalizability and renormalization of the bulk theory.

two unusual features about the ADM formalism to be employed. The first is that the split direction (to be denoted by $x^3$) is not the time coordinate but one of the spatial coordinates. For a Schwarzschild geometry, for instance, it is the radial coordinate. The reason for considering such unusual splitting is basically, as will become clearer, to conduct appropriate gauge-fixing and at the same time to retain the dynamism of the 3D surface. The second unusual feature is that instead of the Hamiltonian formalism, the Lagrangian formalism is ultimately employed for analyzing the constraints; the lapse function and shift vector are nondynamical and can be gauge-fixed, leading to the constraints that correspond to the Hamiltonian and momentum constraints, respectively, in the standard ADM Hamiltonian formalism. Those constraints are then solved in the Lagrangian formalism.

When AdS/CFT correspondence was initially proposed, the actual dualization procedure—namely, how to produce, even in principle, one theory starting from the other—was not spelled out. This issue has been addressed in [36,39,40]. Substantial efforts were made to derive one of the dual theories by starting with the other and going through a cerain dualization procedure. For instance, in [40] where IIB supergravity was considered, it was shown that the gauge field appears as the moduli field of the solution of the gravity side field equations. At the philosophical level, this dualization procedure provides an important rationale for the present work: the dual gauge theory degrees of freedom are obtainable from the starting gravity theory. (The procedure employs the apparatus of the Hamilton–Jacobi formalism.) In the present case, the theory "dual" to the starting gravity theory is realized as the theory describing the boundary dynamics. In particular it takes the form of a lower-dimensional gravity theory. (The "dual" degrees of freedom in this case are, compared to the standard AdS/CFT, more akin to the those of the original theory. This must be due to the fact that here they are obtained without employing the Hamilton–Jacobi formalism. This difference between the two cases is a technical, rather than fundamental, one.) The dualization procedure signifies two things: firstly it serves as a framework of the proof of the AdS/CFT type conjecture. Secondly it suggests that the gravity theory should be renormalizable. This is because it cannot be that one theory has a predictive power, i.e., renormalizable, whereas its dual does not.

On the technical side several improvements have been made in our approach. What had not been noticed in the past was that the shift vector constraint can be explicitly solved in the Lagrangian formalism (for which the commutator of a Lie derivative and covariant derivative plays an important role); the shift vector can be gauged away and the resulting Hamiltonian then becomes identical to the lapse function constraint. In other words the gauge-fixed Hamiltonian is the operator that governs the $x^3$-evolution, and acts as a constraint at the same time. We will see that what brings along the reduction is the dual role of the lapse function constraint (or the Hamiltonian), the physical state constraint as well as the "usual" translational generator along the direction separated out.

Another noteworthy improvement is the employment of the propagator that is constructed out of the traceless fluctuation metric, which is essentially a gauge-fixing [41]. We loosely call the propagator the "traceless propagator." The problem caused by the presence of the trace piece of the fluctuation metric is the pre-loop divergence [42–44]. The necessity of the imposition of the traceless condition was previously mentioned in [45]. Our analysis reveals that the presence of the trace piece is incompatible with the covariance: it interferes with it. The use of the traceless propagator is a necessary condition for the 4D covariance.

## 3. Foliation-Based Quantization

As discussed in the previous section, the key to the renormalizability is the Holography-induced reduction of the physical states. In this section we review the steps of the reduction for two cases: a pure Einstein gravity [19] and an Einstein–Maxwell system [46]. The backgrounds, such as a flat, Schwarzschild, and dS (in the static coordinates) are among the examples to which the procedure below can be applied. A broader range of backgrounds is discussed [36]: the method should be applicable, e.g., to any asymptotically flat background.

The ADM formalism provides a convenient framework for the reduction: the split direction serves as the dimension to be reduced. Although the genuine time direction is separated out in the standard practice of the ADM formalism, in the present analysis, one of the spatial directions, denoted by $x^3$, will be split out. The associated "Hamiltonian" governs the evolution of the system along that direction. For the reduction of the physical states it is crucial to note that the lapse function constraint becomes identical to the $x^3$-Hamiltonian itself once the shift vector constraint is solved and the gauge-fixing is explicitly enforced. On the one hand, the lapse constraint generates the "time"-translation; on the other hand, the constraint serves as the definition of the physical states. They are the states invariant under this "time"-translation. We follow the standard procedure of starting with a classical theory, then moving to the operator quantization (and ultimately to the path integral quantization in Section 5).

### 3.1. Einstein Gravity Case

Consider the Einstein–Hilbert action

$$S_{EH} \equiv \frac{1}{\kappa^2} \int d^4x \sqrt{-g}\, R. \tag{2}$$

Let us employ the ADM formalism [35] where one of the spatial coordinates is split out,

$$x^\mu = (y^m, x^3) \quad \mu = 0, .., 3, \ m = 0, 1, 2; \tag{3}$$

the 4D metric is parameterized as

$$g_{\mu\nu} = \begin{pmatrix} \gamma_{mn} & N_m \\ N_n & n^2 + \gamma_{mn} N^m N^n \end{pmatrix}, \quad g^{\mu\nu} = \begin{pmatrix} \gamma^{mn} + \frac{1}{n^2} N^m N^n & -\frac{1}{n^2} N^m \\ -\frac{1}{n^2} N^n & \frac{1}{n^2} \end{pmatrix}. \tag{4}$$

The fields $n$ (not to be confused with the index $n$) and $N_m$ denote the lapse function and shift vector, respectively. The Einstein–Hilbert action can be written as (see, e.g., [47] for a review)

$$S_{EH} = \int d^4x \, n\sqrt{-\gamma} \left[ \mathcal{R} + K^2 - K_{mn}K^{mn} + 2\nabla_\alpha(n^\beta \nabla_\beta n^\alpha - n^\alpha \nabla_\beta n^\beta) \right]. \tag{5}$$

where $\nabla_\alpha$ denotes the 4D covariant derivative and $K_{mn}$ the second fundamental form given by

$$K_{mn} = \frac{1}{2n} \left( \mathcal{L}_3 \gamma_{mn} - D_m N_n - D_n N_m \right), \qquad K = \gamma^{mn} K_{mn}, \tag{6}$$

where $\mathcal{L}_3$ denotes the Lie derivative along the vector field $\partial_{x^3}$ and $D_m$ is the 3D covariant derivative. The surface term, $2\nabla_\alpha(n^\beta \nabla_\beta n^\alpha - n^\alpha \nabla_\beta n^\beta)$, where $n^\alpha$ denotes the unit normal to the boundary, will be set aside for now[5]; it will play an important role in the 3D reduction discussed in Section 4.4.

Let us fix the bulk gauge symmetry by the de Donder gauge; at the full nonlinear level the gauge condition is given by

$$g^{\rho\sigma} \Gamma^\mu_{\rho\sigma} = 0. \tag{7}$$

---

[5]　A careful treatment of the boundary term can be found in [41].

It reads, in the ADM fields [48],[6]

$$(\partial_{x^3} - N^m \partial_m)n = n^2 K$$
$$(\partial_{x^3} - N^n \partial_n)N^m = n^2(\gamma^{mn}\partial_n \ln n - \gamma^{pq}\Gamma^m_{pq}). \tag{10}$$

As is well known, the lapse function and shift vector are non-dynamic. Because of this, it is natural to gauge-fix them to their background values, which can be accomplished by using the gauge symmetry generated by the $x^3$-component of the diffeomorphism parameter that has a residual 3D coordinate dependence [25]. We fix the lapse function to

$$n = n_0, \tag{11}$$

where $n_0$ denotes the background value (e.g., $n_0 = 1$ for a flat background). For the shift vector we adopt the synchronous-type gauge-fixing of the residual 3D gauge symmetry:

$$N_m = 0. \tag{12}$$

The backgrounds such as flat, Schwarzschild, and dS (in the static coordinates) are among the examples for which this gauge can be chosen. Let us illustrate the quantization procedure with a flat background. The procedure for more general curved backgrounds can be found in [36,41,50]. With these gauge-fixings, namely, $n_0 = 1, N_m = 0$ for a flat background, the shift vector constraint (usually called the momentum constraint in the standard ADM Hamiltonian formalism),

$$D^n(-K_{mn} + K\gamma_{mn}) = 0, \tag{13}$$

is automatically satisfied [19]. In other words, the shift vector constraint is solved by the gauge-fixings. This can be seen as follows. Substitution of $N_m = 0$ into (13) leads to

$$\nabla^m \left[ \frac{1}{n} \left( \mathscr{L}_{\partial_3} \gamma_{mn} - \gamma_{mn} \gamma^{pq} \mathscr{L}_{\partial_3} \gamma_{pq} \right) \right] = 0. \tag{14}$$

Although the covariant derivative and Lie derivative do not commute in general, they do commute in the present case. (See [20,51] for more details.) Because of this, the only surviving term is the one with the covariant derivative acting on the lapse function. For the class of the backgrounds under consideration, that term vanishes as well.

As for the lapse function constraint (i.e., the field equation of the lapse field; it is called the Hamiltonian constraint in the standard ADM Hamiltonian formalism), we impose it as the physical state condition:

$$\left[ \mathcal{R} - K^2 + K_{mn}K^{mn} \right] |phys> = 0 \tag{15}$$

---

[6]　For the perturbative quantization one needs to compute the propagator. However, it was noticed long ago [42–44] that the path integral is not well defined due to the trace mode of the fluctuation metric $h_{\mu\nu}$. The problematic trace piece must be gauge-fixed. (The need for gauge-fixing of the trace piece is already revealed at the classical level, as we will note in Section 4.) In fact, the set of the gauge-fixings just mentioned leads to natural gauge-fixing of the trace piece as well. To see this let us consider the first equation of the de Donder gauge in (10), which we quote here for convenience:

$$(\partial_{x^3} - N^m \partial_m)n = n^2 K. \tag{8}$$

Since the lapse function $n$ has been gauge-fixed to its classical value, this equation implies the trace piece of the second fundamental form is also gauge-fixed to its classical value [41,49]:

$$K = K_0, \tag{9}$$

where $K$ and $K_0$ are the trace of the second fundamental form associated with offshell and onshell metrics, respectively.

The condition will be illuminated by the mathematical picture described in Section 3.3.

Before the lapse and shift gauge-fixings, the bulk part of the "Hamiltonian of $x^3$-evolution" takes

$$H = \int d^3y \left[ n(-\gamma)^{-1/2}(-\pi^{mn}\pi_{mn} + \frac{1}{2}\pi^2) - n(-\gamma)^{1/2}R^{(3)} - 2N_m(-\gamma)^{1/2}D_n[(-\gamma)^{-1/2}\pi^{mn}] \right], \quad (16)$$

where $\pi^{mn}$ denotes the momentum field,

$$\pi^{mn} = \sqrt{-\gamma}\,(-K^{mn} + Kh^{mn}). \quad (17)$$

As well known the Hamiltonian can be expressed in terms of the lapse and shift constraints; omitting the surface terms, the Hamiltonian density is given by

$$\begin{aligned} \mathcal{H} &= \sqrt{-\gamma}\,n(K^2 - K_{mn}K^{mn} - \mathcal{R}) + 2\sqrt{-\gamma}\,N_m D_n(K^{mn} - K\gamma^{mn}) \\ &= \sqrt{-\gamma}\left[ -n\mathcal{C}_0 - 2N^m\mathcal{C}_{0m} \right], \end{aligned} \quad (18)$$

where

$$\begin{aligned} \mathcal{C}_0 &\equiv \mathcal{R} - K^2 + K_{mn}K^{mn} \\ \mathcal{C}_{0m} &\equiv D^n(-K_{mn} + K\gamma_{mn}). \end{aligned} \quad (19)$$

As discussed in details in [41], the gauge-fixing of the trace piece of the metric leads to the following constraint:

$$K = 0. \quad (20)$$

This is consistent with the first equation of (10). With $n = 1$ for the flat background, for example, the shift vector constraint is automatically satisfied and the Hamiltonian density takes

$$\mathcal{H}_{g.f.} = -\sqrt{-\gamma}\,\mathcal{C} = -\sqrt{-\gamma}\,(\mathcal{R} - K^2 + K_{mn}K^{mn}), \quad (21)$$

where $\mathcal{H}_{g.f.}$ denotes the gauge-fixed Hamiltonian density.

The lapse function constraint (15) with the shift vector gauge-fixing implies, in the full nonlinear sense (see [52] for an earlier related analysis),

$$H_{g.f.}|phys >= 0, \quad (22)$$

where $H_{g.f.}$ denotes the gauge-fixed Hamiltonian.

Note the dual roles of the gauge-fixed Hamiltonian: it governs the $x^3$-evolution, and at the same time serves as a constraint. It implies that the physical states are the ones associated with the "holographic screen" in the asymptotic boundary region [46]. It is this reduction that allows a description of the 4D physics through the 3D window.

*3.2. Einstein–Maxwell Case*

Consider an Einstein–Maxwell action

$$S = \int d^4x \sqrt{-g} \left[ R - \frac{1}{4}F_{\mu\nu}F^{\mu\nu} \right]. \quad (23)$$

The 3+1 splitting yields

$$S = \int d^4x \, n\sqrt{-\gamma}\Big[\mathcal{R} + K^2 - K_{mn}K^{mn} \tag{24}$$

$$-\frac{1}{4}F_{mn}F^{mn} - \frac{1}{2n^2}\big(F_{3n}F_{3p}\gamma^{np} - 2F_{mn}F_{3p}N^m\gamma^{np} + F_{mn}F_{pq}N^nN^q\gamma^{mp}\big)\Big],$$

where the indices are raised and lowered by the 3D metric $\gamma_{mn}$. Let us fix the $U(1)$ gauge invariance of the vector field by imposing the axial gauge

$$A_3 = 0. \tag{25}$$

The canonical momentum is defined in the usual manner:

$$\Pi^m \equiv \frac{\mathcal{L}}{\partial(\mathscr{L}_{\partial_3}A_m)} = -\frac{\sqrt{-\gamma}}{n}\,\gamma^{mp}F_{3p} + \frac{\sqrt{-\gamma}}{n}\,N^q\gamma^{mn}F_{qn}. \tag{26}$$

As before, the lapse function and shift vector field equations serve as the constraints; at the classical level they are given, respectively, by

$$\mathcal{C}_0 - \frac{1}{4}F_{mn}F_{pq}\gamma^{mp}\gamma^{nq} + \frac{1}{2(\sqrt{-\gamma}\,)^2}\Pi^m\Pi^n\gamma_{mn} = 0 \tag{27}$$

and

$$-D^k(K_{km} - \gamma_{km}K) - \frac{1}{\sqrt{-\gamma}}F_{mk}\Pi^k = 0, \tag{28}$$

where $\mathcal{C}_0$ is defined in (19). The Hamiltonian can be computed in the standard manner

$$\mathscr{H} = \pi^{ab}\mathscr{L}_{\partial_3}\gamma_{ab} + \Pi^m\mathscr{L}_{\partial_3}A_m - \mathcal{L}$$

$$= -\sqrt{-\gamma}\,(n\,\mathcal{C}_0 + 2N^m\mathcal{C}_{0m}) - \frac{n}{2\sqrt{-\gamma}}\Pi^m\Pi^n\gamma_{mn} + \Pi^kN^qF_{qk} + \frac{1}{4}n\sqrt{-\gamma}\,F_{mn}F_{pq}h^{mp}\gamma^{nq}$$

$$= -\sqrt{-\gamma}\,n\Big[\mathcal{R} - K^2 + K_{mn}K^{mn} + \frac{1}{2(\sqrt{-\gamma}\,)^2}\Pi^m\Pi^n\gamma_{mn} - \frac{1}{4}F_{mn}F_{pq}\gamma^{mp}\gamma^{nq}\Big]$$

$$-\sqrt{-\gamma}\,N^m\Big[2D^n(-K_{mn} + K\gamma_{mn}) - \frac{1}{\sqrt{-\gamma}}F_{mk}\Pi^k\Big], \tag{29}$$

and can be rewritten as

$$\mathscr{H} = -\sqrt{-\gamma}\,n\mathcal{C}_A - \sqrt{-\gamma}\,N^m\Big[2D^n(-K_{mn} + K\gamma_{mn}) - \frac{1}{\sqrt{-\gamma}}F_{mk}\Pi^k\Big], \tag{30}$$

where

$$\mathcal{C}_A \equiv \mathcal{C}_0 - \frac{1}{4}F_{mn}F_{pq}\gamma^{mp}\gamma^{nq} + \frac{1}{2(\sqrt{-\gamma}\,)^2}\Pi^m\Pi^n\gamma_{mn}. \tag{31}$$

With the following gauge-fixing $N_m = 0$ by exploiting the 3D diffeomorphism, the Hamiltonian becomes proportional to the lapse function constraint. Once the shift vector is set to zero, Equation (28) is satisfied[7] and, as in the pure Einstein, case the reduction takes place.

---

[7]  As for a curved background such as a Schwarzschild or Kerr black hole, the constraint can be satisfied by choosing the fluctuation part of the vector field to be independent of $r$. (For this see the analysis in [50]—which is reviewed below.) The leading order part is just the classical field equation.

*3.3. Mathematical Foundations for Reduction*

Although the content of the present subsection lies outside the main stream of the paper, it offers an enlightening mathematical perspective for the gauge-fixing and reduction. Above we saw that the shift vector constraint leads to $\partial_m n = 0$. This condition implies that the foliation of the spacetime is of a special type known as Riemannian in foliation theory. Interestingly, Riemannian foliation admits "dual" foliation, known as totally geodesic foliation (see Figure 2), a result relatively recent in the timeframe of mathematics [53]. One of the facts that make these special foliations interesting is the presence of the so-called parallelism [54], and in the context of the totally geodesic foliation under consideration the parallelism is "tangential" [53] and has the associated abelian Lie algebra. In other words, a totally geodesic foliation has the so-called tangential parallelism and the corresponding Lie algebra (the duals of the transverse parallelism and its Lie algebra of the Riemannian foliation [54]). It was proposed in [20] that the abelian symmetry be associated with the gauge symmetry that allows the gauge-fixing of the lapse and shift: the lapse and shift gauge symmetry should somehow be related to the fibering by the group action that generates the "time" direction, i.e., the tangential parallelism. The gauge-fixing then corresponds to taking the quotient of the bundle by the group, bringing us to the holographic reduction.

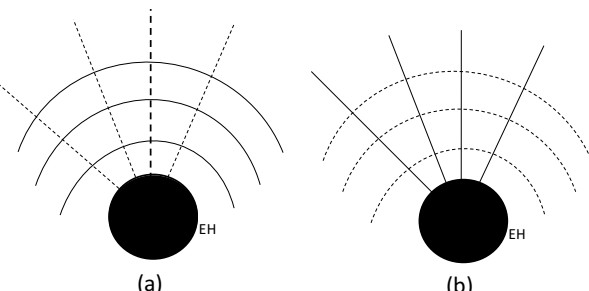

**Figure 2.** (**a**) Constant-*r* surfaces as leaves; (**b**) radial lines as leaves.

In fact, the potential significance of this abelian algebra for the physics context becomes much clearer once the whole mathematical setup is reconstructed in the framework of jet bundle theory [21] (see, e.g., [55–57] for reviews of jet bundle theory). For example, the jet bundle setup brings it to light that the abelian symmetry is a gauge symmetry. The key result of [21] is the proposal in [20] that the abelian symmetry be associated with the gauge symmetry is now on firmer ground, and we have a refined confirmation of the reduction delineated in [20]. In particular, the modding-out procedure, which is central for the reduction picture but only qualitatively outlined in [20], was made quantitative and precise in [21].[8]

## 4. Boundary Conditions and Dynamics

The fact that the physical states of a gravitational system are associated with a hypersurface in the asymptotic region is indicative of the likelihood that the roles of boundary conditions are far more important than conventionally presumed. This makes it imperative to re-examine the issues of the boundary conditions and dynamics. The analysis is not only meaningful in its own right but should also precede certain quantum-level manipulations such as partial integrations performed on the quantum-level action.

The standard Dirichlet boundary condition has been widely used in non-gravitational and gravitational field theories. However, it has become clear that in gravitational theories a large portion

---

[8]　With the relevance of the totally geodesic foliation now understood, it must be possible to conduct the modding-out procedure by employing the symplectic quotient approach [58,59].

of the dynamics is not accounted for in limiting the boundary condition to that of the Dirichlet. There are several prominent indications of the relevance of the non-Dirichlet boundary conditions. For instance, it has recently been shown that quantum corrections are at odds with the Dirichlet boundary condition [60,61]. Another indication comes from the recent development along the line of Bondi–Metzner–Sachs (BMS) symmetry [62,63] where renewed attention has been given to large gauge transformations (LGTs). Suppose one has a solution that satisfies the Dirichlet boundary condition. An inequivalent solution can be obtained by performing an LGT on the solution. Since an LGT acts nontrivially at the asymptotic region, the new solution will have nontrivial dynamics and this is at odds with the Dirichlet boundary condition initially imposed.

The boundary condition is changed through several different channels. For instance, the change can be induced by different foliations within the same coordinate system (such as the $t$- vs $r$-foliation of a Schwarzschild geometry). The boundary condition can also be changed by an ordinary (i.e., "non-large") or large gauge transformation. Another quite obvious way of changing the boundary condition is to add a different boundary term. We will take a closer look at some of these aspects in what follows.

To get to the bottom of the matter, we start by recalling the well-known fact that there are two complementary ways of dealing with a spacetime symmetry. Let us collectively denote by $\Phi$ the fields of the system under consideration. The philosophy of the each picture can be seen through the notations: the passive viewpoint is denoted by $\Phi'(x')$ and the active viewpoint by $\Phi'(x)$. For the present case, the field $\Phi$ is the metric and the symmetry is the diffeomorphism; the two complementary forms of the gauge transformations are the passive and active transformations, $g'_{\mu\nu}(x')$ and $g'_{\mu\nu}(x)$, respectively. They contain complementary pieces of information: the metric $g'_{\mu\nu}(x')$ satisfies the Dirichlet boundary condition in the new coordinates, whereas the metric $g'_{\mu\nu}(x)$—which can be interpreted as a "new" solution in the original coordinates—may satisfy a non-Dirichlet boundary condition in the original coordinate system $x^\mu$. This can be seen by considering an infinitesimal 4D diffeomorphism:

$$g'_{\mu\nu}(x) = g_{\mu\nu}(x) + \nabla_\mu \epsilon_\nu + \nabla_\nu \epsilon_\mu, \tag{32}$$

which is generated by a small parameter $\epsilon_\mu = \epsilon_\mu(t, r, \theta, \phi)$ with a non-trivial $t$-dependence at the boundary region. In the new coordinate system $x'^\mu$ the passively-transformed metric $g'_{\mu\nu}(x')$ satisfies the Dirichlet boundary condition, and thus such a transformation should definitely be allowed. The active form $g'_{\mu\nu}(x)$ would not, in general, satisfy the Dirichlet boundary condition in the original coordinates $x^\mu$. This has interesting implications, as we will discuss.

In Section 4.1, we examine certain aspects of the active transformation in detail. Afterwards we review the standard Dirichlet boundary condition in Section 4.2. This sets the stage for the discussion of the Neumann boundary conditions in Section 4.3. Section 4.4 is devoted to obtaining the reduced form of the action. Finally we ponder in Section 4.5 the ramifications of the results. In particular we consider their implications for the Noether charge and black hole information.

## 4.1. Actively-Transformed Metric

A careful examination of the active transformation turns out to be quite informative. Consider the active form of the transformation

$$g'_{\mu\nu}(x) = g_{0\mu\nu}(x) + h_{\mu\nu}, \quad h_{\mu\nu} \equiv \nabla_\mu \epsilon_\nu + \nabla_\nu \epsilon_\mu. \tag{33}$$

Naively, the actively-transformed metric $g'_{\mu\nu}(x)$ would automatically, i.e., without any further condition on $\epsilon_\mu$, satisfy the metric field equation. In contrast, the analysis below unravels that $g'_{\mu\nu}(x)$ satisfies the metric field equation only with gauge-fixing of the fluctuation field $h_{\mu\nu} \equiv \nabla_\mu \epsilon_\nu + \nabla_\nu \epsilon_\mu$. In other words, the parameter $\epsilon_\mu$ must be constrained by the gauge-fixing conditions on $h_{\mu\nu}$ (Equation (40) below).

Let us explicitly check that the actively-transformed metric satisfies the field equation by considering the $h_{\mu\nu}$-linear order Einstein equation:

$$\int d^4x' h_{\alpha\beta} \frac{\delta}{\delta g_{\alpha\beta}} \left( R^{\mu\nu} - \frac{1}{2} g^{\mu\nu} R \right) \big|_{g_{\alpha\beta}=g_{0\alpha\beta}}.$$

The symbol $|_{g_{\alpha\beta}=g_{0\alpha\beta}}$ indicates that $g_{0\alpha\beta}$ is substituted into $g_{\alpha\beta}$ after taking the derivative; it will be suppressed from now on. At the end we will set $h_{\mu\nu} = \nabla_\mu \epsilon_\nu + \nabla_\nu \epsilon_\mu$. Carrying out the functional derivative, one gets

$$= \int h_{\alpha\beta} \left( R^{\mu\alpha\nu\beta} + g_{\rho\sigma} \frac{\delta}{\delta g_{\alpha\beta}} R^{\mu\rho\nu\sigma} + \frac{1}{2} g^{\alpha\mu} g^{\beta\nu} R + \frac{1}{2} g^{\mu\nu} R^{\alpha\beta} - \frac{1}{2} g^{\mu\nu} g^{\rho\sigma} \frac{\delta R_{\rho\sigma}}{\delta g_{\alpha\beta}} \right). \tag{34}$$

Since the background satisfies the Einstein equation, one can set $R = 0 = R^{\rho\sigma}$; the above becomes

$$= \int h_{\alpha\beta} \left( R^{\mu\alpha\nu\beta} + g_{\rho\sigma} \frac{\delta}{\delta g_{\alpha\beta}} R^{\mu\rho\nu\sigma} - \frac{1}{2} g^{\mu\nu} g^{\rho\sigma} \frac{\delta R_{\rho\sigma}}{\delta g_{\alpha\beta}} \right). \tag{35}$$

The second and third terms can be further manipulated by utilizing

$$\delta R^\rho{}_{\sigma\mu\nu} = \nabla_\mu \delta\Gamma^\rho_{\nu\sigma} - \nabla_\nu \delta\Gamma^\rho_{\mu\sigma}, \quad \delta R_{\mu\nu} = \nabla_\rho \delta\Gamma^\rho_{\mu\nu} - \nabla_\nu \delta\Gamma^\rho_{\rho\mu} \tag{36}$$

$$\delta\Gamma^\lambda_{\mu\nu} = \frac{1}{2} g^{\lambda\kappa} (\nabla_\mu \delta g_{\nu\kappa} + \nabla_\nu \delta g_{\mu\kappa} - \nabla_\kappa \delta g_{\mu\nu}); \tag{37}$$

one can show that

$$\int h_{\alpha\beta} g_{\rho\sigma} \frac{\delta}{\delta g_{\alpha\beta}} R^{\mu\rho\nu\sigma} = \frac{1}{2} \left( 2\nabla^\rho \nabla^{(\nu} h^{\mu)}_\rho - \nabla^\mu \nabla^\nu h^\gamma_\gamma - \nabla^2 h^{\mu\nu} \right) - 2R^{\mu\alpha\nu\beta} - h^{\alpha\beta} R^{(\mu}_\alpha g^{\nu)}_\beta$$

$$= \frac{1}{2} \left( 2\nabla^\rho \nabla^{(\nu} h^{\mu)}_\rho - \nabla^\mu \nabla^\nu h^\gamma_\gamma - \nabla^2 h^{\mu\nu} \right) - 2h_{\alpha\beta} R^{\mu\alpha\nu\beta}, \tag{38}$$

where in the second equality, the Ricci tensor term has been omitted for the same reason stated previously, and

$$-\frac{1}{2} \int h_{\alpha\beta} g^{\mu\nu} g^{\rho\sigma} \frac{\delta R_{\rho\sigma}}{\delta g_{\alpha\beta}} = -g^{\mu\nu} \left( \nabla^p \nabla^q h_{pq} - \nabla^2 h^\gamma_\gamma \right). \tag{39}$$

Let us now impose the following gauge conditions,

$$h^\gamma_\gamma = 0 \quad , \quad \nabla^\kappa h_{\kappa\mu} = 0. \tag{40}$$

The first is the traceless condition of the fluctuation metric; the second is the (linear-order) de Donder gauge. (More on this in Section 5.) The reason we need the gauge-fixing (40) in order for (33) to satisfy the field equation should be that inversion of the kinetic operator—for which the gauge-fixing is required—is somehow built-in to the functional Taylor expansion. With these, one gets

$$\int h_{\alpha\beta} g_{\rho\sigma} \frac{\delta}{\delta g_{\alpha\beta}} R^{\mu\rho\nu\sigma} = \frac{1}{2} \left( 2\nabla^\rho \nabla^{(\nu} h^{\mu)}_\rho - \nabla^2 h^{\mu\nu} \right) - 2h_{\alpha\beta} R^{\mu\alpha\nu\beta} \tag{41}$$

and

$$-\frac{1}{2} \int h_{\alpha\beta} g^{\mu\nu} g^{\rho\sigma} \frac{\delta R_{\rho\sigma}}{\delta g_{\alpha\beta}} = 0. \tag{42}$$

For the present purpose the fluctuation field is given by $h_{\mu\nu} = \nabla_\mu \epsilon_\nu + \nabla_\nu \epsilon_\mu$. Using this and the following identities,

$$
\begin{aligned}
(\nabla_\alpha \nabla_\beta - \nabla_\beta \nabla_\alpha) T^{\rho_1 \cdots \rho_n}{}_{\lambda_1 \cdots \lambda_m} &= -\Sigma_{i=1}^n R_{\alpha\beta\kappa}{}^{\rho_i} T^{\rho_1 \cdots \kappa \cdots \rho_n}{}_{\lambda_1 \cdots \lambda_m} \\
&\quad + \Sigma_{j=1}^m R_{\alpha\beta\lambda_j}{}^\kappa T^{\rho_1 \cdots \rho_n}{}_{\lambda_1 \cdots \kappa \cdots \lambda_m} \\
\nabla_\alpha R_{\beta\gamma\lambda}{}^\alpha &= -\nabla_\beta R_{\gamma\lambda} + \nabla_\gamma R_{\beta\lambda},
\end{aligned}
\tag{43}
$$

the first two terms in (41) can be reduced further. First note that the gauge conditions constrain $\epsilon_\mu$ by

$$
\nabla^\kappa \epsilon_\kappa = 0 \quad , \quad \nabla^2 \epsilon_\nu = 0
\tag{44}
$$

Combining (43) and (44), one gets the desired result. For instance, one of the terms to be computed is

$$
\begin{aligned}
\nabla^\rho \nabla^\nu \nabla_\rho \epsilon^m &= \left( [\nabla^\rho, \nabla^\nu] + \nabla^\nu \nabla^\rho \right) \nabla_\rho \epsilon^\mu \\
&= -R^{\kappa_1 \nu \kappa_2 \mu} \nabla_{\kappa_1} \epsilon_{\kappa_2},
\end{aligned}
\tag{45}
$$

where the second equality is obtained after using $R^{\mu\nu} = 0$ and $\nabla^2 \epsilon^\mu = 0$. By evaluating the other terms in (41) and using the identities given above, one can show that (41) vanishes, which completes the proof.

*4.2. Dirichlet Boundary Condition*

To set the stage for the subsequent analysis, let us review the variational procedure that had led to the introduction of the standard Gibbons–Hawking–York (GHY)—term. Consider an Einstein–Hilbert action with the GHY boundary term,

$$
S_{EH+GHY} \equiv S_{EH} + S_{GHY}
\tag{46}
$$

$$
S_{EH} \equiv \int_{\mathcal{V}} d^4 x \sqrt{-g}\, R \quad , \quad S_{GHY} \equiv 2 \int_{\partial\mathcal{V}} d^3 x \sqrt{|\gamma|}\, \varepsilon K
\tag{47}
$$

where $\mathcal{V}$ denotes the 4D manifold. The rationale for the introduction of the GHY-terms will be spelled out shortly. For now one can take the genuine time coordinate $t$ as the split direction, which leads to the usual foliation with $\varepsilon = -1$. (Later we will consider an $r$-foliation with $\varepsilon = 1$; the Dirichlet boundary condition in that case is along the $r$ direction.) The variation of $S_{EH}$ yields (see, e.g., [64])

$$
\delta S_{EH} = \int_{\mathcal{V}} \sqrt{-g}\, G_{\mu\nu} \delta g^{\mu\nu} - 2 \int_{\partial\mathcal{V}} d^3 x\, \delta(\sqrt{|\gamma|}\, \varepsilon K) + \int_{\partial\mathcal{V}} d^3 x \sqrt{|\gamma|}\, \varepsilon \left( K \gamma^{mn} - K^{mn} \right) \delta\gamma_{mn},
\tag{48}
$$

where

$$
G_{\mu\nu} \equiv R_{\mu\nu} - \frac{1}{2} R g_{\mu\nu}.
\tag{49}
$$

Setting both of the boundary terms in (48) to zero amounts to over-imposing the boundary conditions. Historically the Dirichlet boundary condition was adopted and used more or less exclusively up until very recently. The addition of the GHY-term is to remove the second term above:

$$
\delta S_{EH+GHY} = \int_{\mathcal{V}} \sqrt{-g}\, G_{\mu\nu} \delta g^{\mu\nu} + \int_{\partial\mathcal{V}} d^3 x \sqrt{|\gamma|}\, \varepsilon \left( K \gamma^{mn} - K^{mn} \right) \delta\gamma_{mn}.
\tag{50}
$$

With the second term cancelled out, one gets the Einstein equation with the Dirichlet boundary condition, $\delta\gamma_{mn}|_{\partial\mathcal{V}} = 0$,

$$
G_{\mu\nu} = 0.
\tag{51}
$$

Although the Dirichlet boundary condition is widely used, the form of the solution (33) indicates that not all is well. Suppose that the background solution is obtained, as it normally would be,

with the standard Dirichlet boundary condition. If one chooses the gauge parameter $\varepsilon_\mu$ such that it vanishes in the asymptotic region, the resulting solution $g'_{\mu\nu}(x)$ will be a gauge-equivalent solution. However, what if one considers a $\varepsilon_\mu$ that has a non-dying time-dependence in the asymptotic region? A large gauge transformation may deform the original solution by such a time-dependent parameter. This observation naturally sets the stage for non-Dirichlet boundary conditions.

*4.3. Neumann Boundary Conditions and Generalization*

Although the use of the Dirichlet boundary condition in gravity theories is in line with its use in non-gravitational theories, one may wonder whether it is the only possible and/or preferable boundary condition. In fact it may seem that the addition of the GHY-term is technically contrived, and one may ask whether or not a different-type boundary condition could be defined without such a term. It turns out that it is not only possible to impose different-type boundary conditions but indeed necessary to include them.

One can rather easily see the need for various boundary conditions and thus the need for the enlarged Hilbert space through the gauge transformations that change the given boundary condition. (Previously we had an infinitesimal-level discussion on the same point.) Let us consider the (finite) passive general coordinate transformation:

$$g'_{\mu\nu}(x') = \frac{\partial x^\rho}{\partial x'^\mu} \frac{\partial x^\sigma}{\partial x'^\nu} g_{\rho\sigma}(x). \tag{52}$$

In the new coordinates $x'^\mu$, one will have a new time coordinate and the foliation associated with the new time coordinate serving as the base space of the 4D manifold viewed as a fiber bundle. Because of the new time, the foliation content of the GHY-term is different from the original, although the form of the GHY-term is the same. If the hypersurface on which the boundary condition is imposed is kept the same (other than being expressed in the new coordinates) two boundary conditions would be considered the same. Normally, however, a natural boundary condition is associated with a different hypersurface in the new coordinates, and because of this the boundary condition is changed. For instance a Schwarzschild metric,

$$g_{S\mu\nu}(r,\theta)dx^\mu dx^\nu \equiv -\left(1 - \frac{2M}{r}\right)dt^2 + \left(1 - \frac{2M}{r}\right)^{-1}dr^2 + r^2 d\Omega^2 \tag{53}$$

can be written as

$$ds^2 = -\left(1 - \frac{2M}{r}\right)dudv + r^2 d\Omega^2, \tag{54}$$

where $(u, v)$ are the null coordinates:

$$u \equiv t - r^* \quad , \quad v \equiv t + r^*, \tag{55}$$

with

$$r^* \equiv r + 2M \ln\left|\frac{r}{2M} - 1\right|. \tag{56}$$

In the null coordinates, the boundary hypersurface on which the boundary condition is imposed is changed due to the fact that a natural boundary condition is in terms of either $u$ or $v$ instead of $t$. In other words, one would choose $(u, r)$ or $(v, r)$ coordinates and impose the boundary conditions accordingly. (As will be elaborated on later, certain foliations are associated with observer-dependent effects at the quantum level.) As we have discussed at the infinitesimal level around (32), one has two complementary forms of the gauge transformations: the passive and active transformations, $g'_{\mu\nu}(x')$ and $g'_{\mu\nu}(x)$. They contain complementary pieces of information: the metric $g'_{\mu\nu}(x')$ satisfies the Dirichlet boundary condition in the new coordinates, whereas the metric $g'_{\mu\nu}(x)$—which can be interpreted as a "new" solution in the original coordinates—may satisfy a non-Dirichlet boundary condition in the original coordinate system $x^\mu$.

The non-Dirichlet or Neumann-type boundary condition above has arisen from a different foliation induced by an active form of a gauge transformation within the same form of the GHY-term. A different type of Neumann boundary condition is obtained by performing a boundary Legendre transformation [65,66]. Consider

$$\delta S_{EH} = \int_{\mathcal{V}} \sqrt{-g}\, G_{\mu\nu}\delta g^{\mu\nu} - 2\int_{\partial\mathcal{V}} d^3x\, \delta(\sqrt{|\gamma|}\,\varepsilon K) + \int_{\partial\mathcal{V}} d^3x\, \sqrt{|\gamma|}\,\varepsilon\left(K\gamma^{mn} - K^{mn}\right)\delta\gamma_{mn}. \tag{57}$$

We employ the Hamilton–Jacobi procedure [40,67]; the momentum is given by

$$\pi^{mn} = \frac{\delta S_{EH+GHY}}{\delta\gamma_{mn}} = \varepsilon\sqrt{|\gamma|}\left(K\gamma^{mn} - K^{mn}\right) \tag{58}$$

With this the boundary terms can be written as

$$-\int \gamma_{mn}\,\delta\pi^{mn}. \tag{59}$$

Let us check this:

$$
\begin{aligned}
\int \gamma_{mn}\,\delta\pi^{mn} &= \int \gamma_{mn}\,\delta\left[\varepsilon\sqrt{|\gamma|}\left(K\gamma^{mn} - K^{mn}\right)\right]\\
&= \int 3\delta(\varepsilon\sqrt{|\gamma|}\,K) + \varepsilon\sqrt{|\gamma|}\,\gamma_{mn}K\delta\gamma^{mn} - \int \gamma_{mn}\,\delta\left[\varepsilon\sqrt{|\gamma|}\,K^{mn}\right]\\
&= \int 3\delta(\varepsilon\sqrt{|\gamma|}\,K) + \varepsilon\sqrt{|\gamma|}\,\gamma_{mn}K\delta\gamma^{mn} - \int \delta\left[\varepsilon\sqrt{|\gamma|}\,\gamma_{mn}K^{mn}\right] + \int \left[\varepsilon\sqrt{|\gamma|}\,K^{mn}\delta\gamma_{mn}\right]\\
&= \int 2\delta(\varepsilon\sqrt{|\gamma|}\,K) - \varepsilon\sqrt{|\gamma|}\,\gamma^{mn}K\delta\gamma_{mn} + \int \left[\varepsilon\sqrt{|\gamma|}\,K^{mn}\delta\gamma_{mn}\right].
\end{aligned} \tag{60}
$$

Note that the boundary terms in (48) can be written as

$$\int_{\partial\mathcal{V}} d^3x\left(\pi^{mn}\delta\gamma_{mn} - \delta\pi\right). \tag{61}$$

This of course implies that the boundary terms of $S_{EH+GHY}$ can be written

$$\int_{\partial\mathcal{V}} d^3x\, \pi^{mn}\delta\gamma_{mn}. \tag{62}$$

This implies that the boundary terms in (48)—which can be viewed as resulting from the boundary Legendre transformation of $S_{EH+GHY}$—can be removed by requiring the following Neumann-type boundary condition

$$\delta\pi^{mn} = 0. \tag{63}$$

It is possible to extend the Einstein–Hilbert case above to a more general system considered in [68]. Such an extension should be useful when dealing with the 3+1 splitting of the quantum-level action. Consider the following form of the action:

$$S = \frac{1}{2}\int \sqrt{-g}\, f(R_{\mu\nu\rho\sigma}) \tag{64}$$

where $f$ is an arbitrary function of the Riemann tensor. Instead of (64), one may consider the following first-order form of the action [68]:

$$S = \frac{1}{2}\int \sqrt{-g}\left[f(\varrho_{\mu\nu\rho\sigma}) + \varphi^{\mu\nu\rho\sigma}(R_{\mu\nu\rho\sigma} - \varrho_{\mu\nu\rho\sigma})\right], \tag{65}$$

where $\varrho_{\mu\nu\rho\sigma}$, $\varphi^{\mu\nu\rho\sigma}$ are auxiliary fields. Let us consider the genuine time-splitting; the spatial splitting case can be similarly analyzed. One can show that the 3+1 split form of the action (65) is

$$S = \int \left[ \mathcal{L}_{bulk} + \partial_\mu \left( \sqrt{-g}\, n^\mu K_{pq} \Psi^{pq} \right) \right], \tag{66}$$

where

$$\begin{aligned}
\mathcal{L}_{bulk} &= \sqrt{\gamma} N \left[ \frac{1}{2} f(\varrho_{\mu\nu\rho\sigma}) + \frac{1}{2} \phi^{ijkl}(R_{ijkl} - \rho_{ijkl}) - 2\phi^{mnp}(n^\kappa R_{mnp\kappa} - \rho_{mnp}) \right. \\
&\left. - \Psi^{mn}\left( KK_{mn} + K_{mp}K_n{}^p + n^{-1}D_m D_n n - \Omega_{mn} \right) - n^{-1}K_{mn}(\dot{\Psi}^{mn} - \mathcal{L}_{N^q \partial_q} \Psi^{mn}) \right], \tag{67}
\end{aligned}$$

with

$$\rho_{mnpq} \equiv \varrho_{mnpq} \quad \rho_{mnp} \equiv n^\mu \varrho_{mnp\mu} \quad \Omega_{pq} \equiv n^\mu n^\nu \varrho_{p\mu q\nu} \tag{68}$$

$$\phi^{mnpq} \equiv \gamma^{mm'} \gamma^{nn'} \gamma^{pp'} \gamma^{qq'} \varphi_{m'n'p'q'} \quad \phi^{mnp} \equiv \gamma^{mm'} \gamma^{nn'} \gamma^{pp'} n^\mu \varphi_{m'n'p'\mu} \quad \Psi^{pq} \equiv \gamma^{pp'} \gamma^{qq'} n^\mu n^\nu \varphi_{p'\mu q'\nu}.$$

By considering the 3D metric variation of (67) and collecting the results one gets, for the momentum conjugate to $\gamma_{mn}$,

$$p^{mn} = \sqrt{\gamma} \left[ -\frac{1}{2}\gamma^{mn}\Psi^{rs}K_{rs} - \frac{1}{2}\Psi^{mn}K - \Psi^{ml}K_l{}^n - \frac{1}{2n}(\dot{\Psi}^{mn} - \mathcal{L}_\beta \Psi^{mn}) + \phi^{mjnl}K_{jl} + \frac{2}{n}D_l(N\phi^{lmn}) \right]. \tag{69}$$

The trace part is given by

$$p \equiv \gamma_{mn} p^{mn} = \sqrt{\gamma} \left[ -\frac{5}{2}\Psi^{rs}K_{rs} - \frac{1}{2}\Psi K - \frac{1}{2n}\gamma_{mn}(\dot{\Psi}^{mn} - \mathcal{L}_\beta \Psi^{mn}) + \gamma_{mn}\phi^{mjnl}K_{jl} + \frac{2}{n}\gamma_{mn}D_l(N\phi^{lmn}) \right]. \tag{70}$$

Unlike the Einstein–Hilbert case, the present GHY-term,

$$S_{GHY} \equiv -\int \partial_\mu \left( \sqrt{-g}\, n^\mu K_{pq} \Psi^{pq} \right), \tag{71}$$

and the boundary term that converts the Dirichlet action to the Neumann action (namely the Legendre transformation term) are different.[9] Therefore, adding this term to the original action does not lead to the Dirichlet boundary condition. For the boundary Legendre transformation one can consider, just as in the Einstein–Hilbert case, the variation of $S + S_{GHY}$ with respect to the 3D metric; this variation leads to the following boundary term:

$$p^{mn} \delta \gamma_{mn}. \tag{72}$$

Then to go to the action with the Neumann boundary condition one should perform the Legendre transformation by adding the negative of $p$, $-p$, to $S + S_{GHY}$.

### 4.4. Reduced Action and Boundary Dynamics

It has been shown in Section 3 that the physical sector of the theory is associated with a 3D hypersurface in the boundary region. Given this, a concrete understanding of the boundary dynamics and its coupling to the bulk is necessary for the complete picture. In what follows, we apply the dimensional reduction technique developed in [69,70], and work out the explicit form of the reduced theory—the Lagrangian of the 3D theory. The explicit form of the reduced action can be obtained by consistently reducing the 4D action in two steps: reduction of the 4D field equations to 3D ones and construction of the 3D action that reproduces the resulting 3D equations. Below we illustrate the procedure for an Einstein gravity. We first carry out the reduction at the level of an infinitesimal fluctuation and subsequently discuss the full nonlinear extension.

---

[9]    Recall that even in the Einstein–Hilbert case, they are different for $D \neq 4$ [65,66].

Considering the (3+1) splitting: the ADM form of the Einstein–Hilbert action

$$S_{EH} = \int d^4x \sqrt{-g}\, R \tag{73}$$

can be written

$$S_{EH} = \int_{\mathcal{V}} d^4x\, n\sqrt{|\gamma|}\left[\mathcal{R} + K^2 - K_{mn}K^{mn}\right] - 2\int_{\partial\mathcal{V}} d^3x\sqrt{|\gamma|}\,\varepsilon K. \tag{74}$$

The second term is a boundary term that arises in the ADM description. It is not to be confused with the GHY-term: the GHY-term, if added, cancels this boundary term. We consider the 3+1 splitting where, say, $x^3 = r$. The $n, N_m, \gamma_{mn}$ field equations are, respectively,

$$\mathcal{R} - K^2 + K_{mn}K^{mn} = 0 \tag{75}$$

$$D_a(K^{ab} - \gamma^{ab}K) = 0 \tag{76}$$

$$\begin{aligned}
&\mathcal{R}_{ab} - \frac{1}{2}\mathcal{R}\gamma_{ab} - \frac{1}{2}\gamma_{ab}\left[K^2 - K_{pq}K^{pq}\right] + 2KK_{ab} - 2K_{pa}\gamma^{pq}K_{qb} \\
&+ \frac{1}{n\sqrt{|\gamma|}}\gamma_{pa}\gamma_{qb}\,\partial_r\left[\sqrt{|\gamma|}\gamma^{pq}K\right] - \frac{1}{n\sqrt{|\gamma|}}\gamma_{pa}\gamma_{qb}\,\partial_r\left[\sqrt{|\gamma|}K^{pq}\right] \\
&- \frac{2}{n}\gamma_{ab}D_e(KN^e) + \frac{2}{n}KD_{(a}N_{b)} + \frac{2}{n}\nabla^d(K_{ab}N_d) - \frac{2}{n}K_{n(a}D^nN_{b)} = 0,
\end{aligned} \tag{77}$$

where the symmetrization in (77), $(a\ b)$, is with a factor $\frac{1}{2}$. The reduction procedure has two components. The first is the requirement that the reduction ansatz satisfies the 4D $n, N_m, \gamma_{mn}$ field Equations (75)–(77). The requirement that the ansatz satisfy (77) leads to the 3D version of the $\gamma_{mn}$ field equation. The second component is the construction of the 3D action that yields the 3D $\gamma_{mn}$ field equation. We first consider the reduction that covers static backgrounds including a Schwarzschild or Reissner-Nordström black hole. More general backgrounds, including a Kerr black hole and potentially even time-dependent backgrounds, require additional care and are discussed afterwards.

Static Backgrounds

Let us start with a static background metric of a diagonal form such as a Schwarzschild or Reissner-Nordström geometry. For such backgrounds, a convenient gauge-fixing is to gauge away the shift vector $N_m$ and the fluctuation part of the lapse function:

$$N_m = 0 \quad , \quad n = n_0(r), \tag{78}$$

where $n_0(r)$ denotes the background solution for the lapse function $n$. For a Schwarzschild background, for instance, it is given by

$$n_0 = \left(1 - \frac{2M}{r}\right)^{-\frac{1}{2}}. \tag{79}$$

With $N_m = 0$, the $\gamma^{ab}$ field Equation (77) becomes

$$\begin{aligned}
&\mathcal{R}_{ab} - \frac{1}{2}\mathcal{R}\gamma_{ab} - \frac{1}{2}\gamma_{ab}\left[K^2 - K_{pq}K^{pq}\right] + 2KK_{ab} - 2K_{pa}\gamma^{pq}K_{qb} \\
&+ \frac{1}{n\sqrt{|\gamma|}}\gamma_{pa}\gamma_{qb}\,\partial_r\left[\sqrt{|\gamma|}\gamma^{pq}K\right] - \frac{1}{n\sqrt{|\gamma|}}\gamma_{pa}\gamma_{qb}\,\partial_r\left[\sqrt{|\gamma|}K^{pq}\right] = 0.
\end{aligned} \tag{80}$$

The lapse function constraint (75) is contained above because it can be obtained by taking the trace of (80). (See the analysis around Equation (86).) Substituting $N_a = 0$ into (76) yields

$$D^a \left[ \frac{1}{n} \left( \mathscr{L}_r \gamma_{ab} - \gamma_{ab} \gamma^{cd} \mathscr{L}_r \gamma_{cd} \right) \right] = 0, \tag{81}$$

which is satisfied [19] (namely, the shift vector constraint is solved) by the gauge-fixing above:

$$\partial_a n = \partial_a n_0 = 0. \tag{82}$$

As for the reduction ansatz of the 3D hypersurface metric, first consider the following linear-level reduction ansatz:

$$\gamma_{mn}(t, r, \theta, \phi) = \gamma_{0mn} + \tilde{h}_{mn}(t, \theta, \phi), \tag{83}$$

where $\gamma_{0mn}$ denotes the solution of the field equation, and $\tilde{h}_{mn}(t, \theta, \phi)$ the fluctuation. For the Schwarzschild case, for instance, $\gamma_{0mn}$ is given by a diagonal metric $\gamma_{0mn} = \gamma_{0mn}(r, \theta) = diag(n_0^2, r^2, r^2 \sin^2 \theta)$. We have previously discussed that such an ansatz must be allowed even though it does not obey the original Dirichlet boundary condition. Let us choose $\tilde{h}_{mn}(t, \theta, \phi)$ such that

$$\tilde{h}_{mn} = D_m \epsilon_n + D_n \epsilon_m, \tag{84}$$

for a parameter $\epsilon_m = \epsilon_m(t, \theta, \phi)$. This step of setting $\tilde{h}_{mn}$ to $\tilde{h}_{mn} = D_m \epsilon_n + D_n \epsilon_m$ is not strictly necessary for finding the reduced form of the action: the fact that $\gamma_{mn}(t, r, \theta, \phi)$ with $\tilde{h}_{mn} = D_m \epsilon_n + D_n \epsilon_m$ satisfies the bulk and later the 3D field equations can be checked after the reduced action is obtained. The ansatz (83) is guaranteed to satisfy the $\gamma_{mn}$ field Equation (80) for the following reason.[10] The right-hand side of (83), $\gamma_{0mn}(r) + \tilde{h}_{mn}(t, \theta, \phi)$ is guaranteed to be a solution of the $\gamma_{mn}$ field equation since it takes a form of the gauge transformation of a solution $\gamma_{0mn}(r)$ with the gauge parameter $\epsilon_m$. This also suggests how to obtain the nonlinear ansatz; just borrow the finite form of the gauge transformation of the metric solution. A word of caution is in order. Note that we are choosing $\tilde{h}_{mn}$ to be given by (84), not generating $\tilde{h}_{mn}$, e.g., by utilizing the 3D gauge symmetry after starting with $\gamma_{0mn}$. The 3D symmetry is reserved for the gauge-fixing $N_m = 0$ and thus cannot be utilized for such a purpose. (For the same reason, one cannot gauge away $\tilde{h}_{mn}$.)

Finally, let us show that the trace part of (80) yields the lapse function constraint. The trace part is given by

$$\frac{1}{2} \left( - \mathcal{R} + K^2 - K_{mn}^2 \right) + \frac{\gamma_{pq}}{n \sqrt{|\gamma|}} \partial_r \left( \sqrt{|\gamma|} \left( K \gamma^{pq} - K^{pq} \right) \right) = 0. \tag{85}$$

If one can show that the background $g_{0\mu\nu}$ satisfies

$$\frac{\gamma_{pq}}{n \sqrt{|\gamma|}} \partial_r \left( \sqrt{|\gamma|} \left[ K \gamma^{pq} - K^{pq} \right] \right) = 0, \tag{86}$$

then one gets

$$- \mathcal{R} + K^2 - K_{mn}^2 = 0, \tag{87}$$

which is nothing but (75), the lapse constraint. One can show that (86) is satisfied, e.g., in the Schwarzschild case. Of course this must be generally true.

The remaining task is to show that the field Equation (80)—which is now viewed as a 3D field equation—can be derived from the reduced action to be constructed. In other words, we should construct the 3D action whose metric field equation yields (80). The following point is important before getting to the detailed steps of the construction. What we try to accomplish is to work out the

---

[10]　We have essentially checked this before by using the 4D covariant setup in Section 4.1.

3D Lagrangian that describes the physical fluctuations around a given solution. To elucidate the point let us use the 4D language for the moment. The fluctuations to be considered are the ones that would be generated by a gauge transformation if there were such a residual symmetry:

$$g'_{\mu\nu}(x) = g_{0\mu\nu}(x) + h_{\mu\nu}(x) \quad , \quad h_{\mu\nu} \equiv \nabla_\mu \epsilon_\nu + \nabla_\nu \epsilon_\mu, \tag{88}$$

where $g_{0\mu\nu}(x)$ and $h_{\mu\nu}(x)$ denote a given solution and the fluctuation, respectively. Although the passively-transformed metric $g'_{\mu\nu}(x')$ satisfies a Dirichlet boundary condition, this will not be the case for the actively-transformed metric $g'_{\mu\nu}(x)$. This is because the background $g_{0\mu\nu}(x)$ satisfies the Dirichlet boundary condition, whereas the fluctuation $h_{\mu\nu}(x)$ does not. For this reason it is not a priori clear whether one should start with an action with or without the GHY- term. It turns out, as we will see below, that for consistent reduction one should use the action *without* the GHY- term (74), i.e., the action with the Neumann boundary condition. Perhaps this may not be entirely surprising: the reduced action is the action for the fluctuation fields that satisfy the Neumann-type boundary condition. In other words, the 3D action describes the boundary theory whose dynamics are what causes the boundary condition—from the bulk point of view—to deviate from the Dirichlet. Since the reduction is carried out in the original coordinates, the action without the GHY-term may somehow become relevant, and indeed this is what happens.

For the construction of the 3D action, note that the derivation of the field Equation (80) after starting with the 4D action (74) involves partial integrations along $r$. We will now show that the form of the 3D action that we are after is what is inherited from the 4D action $S_{EH}$ without the GHY-term: the field Equation (80) is then obtained from that 3D action without performing the partial integration along $r$, as required by the fact that an $r$-partial integration cannot be performed since the 3D integration measure does not have the $r$-component. It is thus the Neumann boundary condition that is crucial for getting the field Equation (80). To summarize, if one starts with the action without the GHY-term (74), and takes the $\gamma_{mn}$ variation, one gets the 3D metric field Equation (80) without performing the $r$-partial integration. To explicitly show this, consider the following 3D action,

$$S_{3D} = \int d^3x \, n_0 \sqrt{|\gamma|} \left[ \mathcal{R} + K^2 - K_{mn} K^{mn} \right] - 2 \int d^3x \sqrt{|\gamma|} \, \varepsilon K, \tag{89}$$

where we have set $n = n_0$ and $N_m = 0$. The first term has been inherited from the bulk action and the second term from the boundary term. The boundary term in (74) is not to be confused with the GHY- term; it is a boundary term that appears when expressing the Einstein–Hilbert term in the ADM formalism. It is the negative of the GHY-term; the GHY-term, if added, would cancel out this boundary term. The explicit steps leading to the $\gamma_{mn}$ field equation are as follows. The variation of the first term in (89) is

$$\int d^4x \, n \sqrt{|\gamma|} \, \mathcal{R}_{ab} \, \delta\gamma^{ab} - \frac{1}{2} n \sqrt{|\gamma|} \gamma_{ab} \left[ \mathcal{R} + K^2 - K_{mn} K^{mn} \right] \delta\gamma^{ab}$$
$$+ 2n \sqrt{|\gamma|} \left( KK_{ab} - K_{pa} \gamma^{pq} K_{qb} \right) \delta\gamma^{ab} + 2n \sqrt{|\gamma|} \left[ K\gamma^{ab} \delta K_{ab} - K^{pq} \delta K_{pq} \right]. \tag{90}$$

The third term can be rewritten as

$$\sqrt{|\gamma|} \left[ K\gamma^{ab} \delta\partial_r \gamma_{ab} - K^{pq} \delta\partial_r \gamma_{pq} \right] = \sqrt{|\gamma|} \left[ K\gamma^{ab} - K^{pq} \right] \delta\partial_r \gamma_{ab} = \pi^{ab} \delta\partial_r \gamma_{ab}, \tag{91}$$

where $\varepsilon$ in (58) has been set to $\varepsilon = 1$ in the second equality. The variation of the second term in (89) is

$$-2\delta \int_{\partial\mathcal{V}} d^3x \sqrt{|\gamma|} K = -\delta \int_{\partial\mathcal{V}} d^3x \gamma_{ab} \pi^{ab} = -\delta \int_{\mathcal{V}} d^4x \partial_r (\gamma_{ab} \pi^{ab})$$
$$= -\int_{\mathcal{V}} d^4x \partial_r \left[ (\delta\gamma_{ab}) \pi^{ab} + \gamma_{ab} \delta\pi^{ab} \right]. \tag{92}$$

We have used the trace of (58) in the first equality above. The second term in the second line can be written as a 3D integral:

$$- \int_{\mathcal{V}} d^4x \partial_r \left[ \gamma_{ab} \delta \pi^{ab} \right] = - \int_{\partial \mathcal{V}} d^3x \left[ \gamma_{ab} \delta \pi^{ab} \right]. \tag{93}$$

It thus vanishes upon imposition of the Neumann boundary condition $\delta \pi^{ab} = 0$. The first term in the 2nd line takes

$$- \int_{\mathcal{V}} d^4x \left[ (\partial_r \delta \gamma_{ab}) \pi^{ab} + \delta \gamma_{ab} \partial_r \pi^{ab} \right]. \tag{94}$$

This with (91) yields

$$- \int_{\mathcal{V}} d^4x \, (\delta \gamma_{ab}) \partial_r \pi^{ab}. \tag{95}$$

The metric field Equation (80) is reproduced once one combines all the results above and uses the expression for the momentum field (58).

General Backgrounds

With the discussion so far we are ready to generalize the steps above in order to cover more general backgrounds such as Kerr or time-dependent backgrounds. For this, some of the steps need modification. To see the need for modification, let us take the Kerr case to be specific and contrast it with the Schwarzschild case. The difference between the two cases is the manner in which the gauge-fixings of the lapse function and shift vector satisfy the shift vector constraint (76). While the gauge-fixing $N_m = 0, n^2 = n_0^2(r) = (1 - 2M/r)^{-1}$ for the Schwarzschild case makes each term in (76) separately vanish, the same is not true for the Kerr case, and the manner in which the shift vector constraint is satisfied differs as we now discuss. For a Kerr metric

$$g_{0\mu\nu} = \begin{pmatrix} \frac{2Mr}{\rho^2} - 1 & 0 & 0 & -\frac{2aMr \sin^2(\theta)}{\rho^2} \\ 0 & \frac{\rho^2}{\Delta} & 0 & 0 \\ 0 & 0 & \rho^2 & 0 \\ -\frac{2aMr \sin^2(\theta)}{\rho^2} & 0 & 0 & \frac{\Sigma \sin^2(\theta)}{\rho^2}, \end{pmatrix} \tag{96}$$

where

$$\rho^2 = r^2 + a^2 \cos^2 \theta \quad , \quad \Delta = r^2 - 2Mr + a^2 \quad , \quad \Sigma = (r^2 + a^2)^2 - a^2 \Delta \sin^2 \theta \tag{97}$$

one can adopt the analogous gauge-fixings:

$$N_m = 0 \quad , \quad n^2 = n_0^2(r, \theta) = \frac{\rho^2}{\Delta}. \tag{98}$$

It is not difficult to show that the shift vector constraint (76) is satisfied at the $\tilde{h}_{mn}$-linear order. At the $\tilde{h}_{mn}$-linear order, Equation (76) becomes the leading, i.e., zeroth-order, field equation (since the linear part is trivially removed by the $\partial_r$-derivative appearing in the definition of $K_{ab}$) which is of course satisfied by the Kerr background; the reduced action is again given by (89). However, the $\tilde{h}_{mn}$-higher order status is not so obvious. Moreover, for a more general background such as a time-dependent background, a similar gauge-fixing that solves the constraint may not be available and it may be necessary to keep the shift vector instead of setting it to a fixed value too early. Keeping $N_m$, the reduced action is again given by

$$S_{3D} = \int d^3x \, n_0 \sqrt{|\gamma|} \left[ \mathcal{R} + K^2 - K_{mn} K^{mn} \right] - 2 \int_{\partial \mathcal{V}} d^3x \sqrt{|\gamma|} \, \varepsilon K, \tag{99}$$

in which, although the form is the same as (89), the shift vector $N_m$ is nonzero, acting as a Lagrange multiplier. This reduction procedure should cover quite general backgrounds, potentially including those that are time-dependent.

*4.5. Ramifications of Boundary Dynamics*

With the reduced 3D theory Lagrangian obtained, it is now possible to analyze various aspects of the original 4D system in a way that is not otherwise possible. In Section 4.5.1 we look, from the reduced theory's perspective, into the asymptotic symmetry aspect of the 4D theory. This is done by examining the present analogue of the conformal generalization [71] of the BMS-type symmetry. Afterwards we analyze the implication of a Neumann-type boundary condition for the Noether theorem. In Section 4.5.2, we ponder the quantum aspects of the theory relevant for black hole information. The tie between the boundary condition and the foliation indicates that the Hilbert space must include all of the different foliation-induced boundary conditions. The enlarged Hilbert space is important for black hole information.

4.5.1. Symmetry Aspects of the 3D Theory

The symmetry aspects of the original 4D theory can be studied by utilizing (89), the Lagrangian of the reduced action. The asymptotic conformal Killing symmetry or the conformal BMS group [71]—which contains the BMS group as a subgroup— provides an illuminating setup for studying the clear physical meaning and role of the BMS symmetry. We also examine the implication of the non-Dirichlet boundary condition for the Noether theorem. The standard Noether theorem is based on the Dirichlet boundary condition. We discuss modifications of the conservation law; this is needed for proper understanding of the decrease of the "conserved" charges under the quantum effects such as Hawking radiation.

BMS Symmetry in the Context of Boundary Dynamics

The generalities on an unbroken symmetry should be useful to recall. Let us again denote by $\Phi$ the field of the system under consideration and split the field into a fixed background $\Phi_0$ and the fluctuation $\tilde{\Phi}$:

$$\Phi = \Phi_0 + \tilde{\Phi}. \tag{100}$$

The symmetry group of the theory gets broken into a subgroup that leaves $\Phi_0$ invariant. The BMS symmetry is unbroken in this sense except that it is an asymptotic symmetry instead of a precise symmetry. In [71] the BMS symmetry has been extended to the so-called conformal BMS group, which is basically the asymptotic conformal Killing symmetry. The conformal BMS symmetry (and its present analogue (since we are considering the infinity associated with $r \to \infty$); see below) have an intuitively clear meaning and thus further elucidate the meaning of the BMS group itself.

The diffeomorphism contains a particular form of the conformal transformation. To see this we rewrite the diffeomorphism transformation as

$$\delta g_{\mu\nu} = \frac{1}{2}(\nabla_\kappa \epsilon^\kappa) g_{\mu\nu} + (Lg)_{\mu\nu}, \tag{101}$$

where

$$(Lg)_{\mu\nu} \equiv \nabla_\mu \epsilon_\nu + \nabla_\nu \epsilon_\mu - \frac{1}{2}(\nabla_\kappa \epsilon^\kappa) g_{\mu\nu}. \tag{102}$$

Suppose for the moment that $\epsilon_\nu$ is a precise, instead of asymptotic, conformal Killing vector; then it satisfies

$$\nabla_\mu \epsilon_\nu + \nabla_\nu \epsilon_\mu = \frac{1}{2}(\nabla_\kappa \epsilon^\kappa) g_{\mu\nu}. \tag{103}$$

For such $\epsilon^\mu$, the diffeomorphism takes the form of the following conformal-type transformation:

$$\delta g_{\mu\nu} = \frac{1}{2}(\nabla_\kappa \epsilon^\kappa) g_{\mu\nu}. \tag{104}$$

The asymptotic conformal Killing symmetry is a symmetry generated by $\epsilon^\mu$ that satisfies (103) not precisely but asymptotically [71].

Next, let us examine the physical meaning of an asymptotic symmetry of the present system, the analogue of the conformal BMS group. As reviewed above, for an exact symmetry, one would consider

$$g_{\mu\nu} \equiv g_{0\mu\nu} + h_{\mu\nu}, \tag{105}$$

and look for a subgroup that leaves the background $g_{0\mu\nu}$ invariant. Compared with this, there are several subtleties that make the analysis of an asymptotic symmetry more unwieldy than it would be otherwise. The first and most obvious is the fact that the symmetry under consideration is not an exact symmetry but an asymptotic one. If the asymptotic conformal Killing symmetry were an exact symmetry, it, and therefore the BMS symmetry, would be the unbroken symmetry. Since it is an asymptotic symmetry, it may be dubbed the "asymptotic unbroken symmetry." The second subtlety is the fact that presently one is considering the boundary at a spatial infinity unlike in [71] where the null infinity was considered. As often demonstrated in the literature, however, different asymptotic regions usually have corresponding quantities [72]. Lastly there is the fact that the ADM formalism makes it less transparent to apply the results of [71] to the present setup. Because of these reasons, the statements above on the meaning of the analogue BMS symmetry are not entirely rigorous. Nevertheless, we believe the picture is valid and has certain enlightening perspectives. In Section 4.5.2 we discuss the implication of the analogue BMS symmetry for black hole information.

Noether Charge Non-Conservation

The usual Noether theorem and associated conserved charge are based on the standard Dirichlet boundary condition. As we have previously seen, one encounters a Neumann boundary condition if one considers the dynamics from the active coordinate transformation. The foliation-induced Neumann boundary condition has an interesting implication for the Noether theorem; the Noether current that is conserved in a setup with the standard Dirichlet boundary condition is no longer conserved in a setup with the Neumann boundary condition. The implication seems useful for a deeper understanding of certain aspects of the conserved quantities (or quantities viewed to be conserved in the conventional description with the Dirichlet boundary conditions) of a black hole when they are subjected to quantum effects such as the Hawking radiation. We illustrate this with the mass or entropy of a black hole.

With the Hawking radiation, the mass and entropy of the black hole will decrease. This seems incompatible with the conventional non-dynamic boundary, i.e., a boundary with a Dirichlet boundary condition. This is because a non-dynamic boundary would imply conservation of the charges. This suggests that the mass or entropy decrease must have something to do with a Neumann-type boundary condition [36]. To convey the idea with minimum complications, let us first briefly review the Noether theorem for a non-gravitational system in a flat background. Suppose the system described by a field $\Phi$ has a global symmetry:

$$\Phi \to \Phi + \epsilon \delta \Phi. \tag{106}$$

One way of obtaining the conservation law is to make the parameter $\epsilon$ local; on general grounds, the variation of the action must take the following form:

$$\delta S = \int J^\mu \partial_\mu \epsilon, \tag{107}$$

where $J^\mu$ is the Noether current. If $\Phi$ is taken to be a solution of the field equation, the action must be stationary, from which it follows

$$\delta S = 0. \tag{108}$$

Suppose, as normally assumed, that the field $\Phi$ and its variation $\delta\Phi$ satisfy the Dirichlet boundary condition. Then the two equations above imply

$$\partial_\mu J^\mu = 0, \tag{109}$$

which in turn leads to the standard charge conservation. For a Neumann boundary condition, however, the boundary term does not vanish because the variation

$$\delta S = \int_{\mathcal{V}} J^\mu \partial_\mu \epsilon = \int_{\partial\mathcal{V}} n_\mu J^\mu \epsilon - \int_{\mathcal{V}} \epsilon \partial_\mu J^\mu = 0 \tag{110}$$

implies, instead of (109),

$$\int_{\mathcal{V}} (\partial_\mu J^\mu) \epsilon = \int_{\partial\mathcal{V}} (n_\mu J^\mu) \epsilon. \tag{111}$$

This means that the bulk current is not conserved, but coupled with the corresponding boundary quantity.

Let us turn to the present Einstein–Hilbert case. We consider the foliation-induced Neumann boundary condition. From the fact that the diffeomorphism variation (under $x^\mu \to x'^\mu = x^\mu - \xi^\mu$) is essentially a Lie dragging, it follows that

$$\delta_\xi(\sqrt{-g}\, L) = \sqrt{-g}\, \nabla_\mu(\xi^\mu L) \tag{112}$$

On the other hand the diffeomorphism variation $\delta_\xi$ is a special case of an arbitrary variation $\delta$; thus

$$\delta_\xi S_{EH} = \int_{\mathcal{V}} \sqrt{-g} \left[ G_{\mu\nu}\, \delta_\xi g^{\mu\nu} + \nabla^\rho v_\rho \right], \tag{113}$$

where

$$v_\rho = \nabla^\lambda \delta g_{\rho\lambda} - g^{\alpha\beta} \nabla_\rho \delta g_{\alpha\beta}. \tag{114}$$

The equivalence of (112) and (113) leads to the following current $J^\mu$ [73]

$$J^\mu \equiv -2G^{\mu\nu}\xi_\nu + v^\mu - L\xi^\mu, \tag{115}$$

with

$$\nabla_\mu J^\mu = 0. \tag{116}$$

Let us show that the current associated with the "new solution" with the Neumann boundary condition, given in (32) (here we use $\xi^\mu$ instead of $\epsilon^\mu$), does not satisfy the standard current conservation, which of course implies that the charge, i.e., mass or entropy, is not conserved. This can be seen by considering the active transformation of the current

$$J'^\mu = J^\mu + \delta J^\mu = J^\mu + \mathscr{L}_\xi J^\mu, \tag{117}$$

and the volume integral of its divergence:

$$\int d^4x \sqrt{-g}\, \nabla_\mu J'^\mu = \int d^4x \sqrt{-g}\, \nabla_\mu(\mathscr{L}_\xi J^\mu), \tag{118}$$

where $\nabla_\mu J^\mu = 0$ has been used to obtain the right-hand side. By applying the Stokes' theorem one gets, for the right-hand side,

$$\int d\Sigma_\alpha \, \mathcal{L}_\xi J^\alpha |. \tag{119}$$

The vertical line '|' indicates that the integrand is evaluated at the boundary $\Sigma$. The expression above vanishes for a Dirichlet class, since for the Dirichlet boundary condition, $\xi = 0$ at the boundary. However, the same is not true for the $\xi^\mu$ of the Neumann boundary condition, showing that the mass or entropy decrease via Hawking radiation is linked with the Neumann boundary condition.

### 4.5.2. Implications for Black Hole Information

There are several facets of the quantum-level dynamics and boundary conditions that are relevant to the black hole information problem. In [36,74] we have raised the possibility that the information may be bleached through a quantum gravitational process in the vicinity of the horizon and released before entry of the matter into the horizon. There has been a proposal in loop quantum gravity that the Hilbert space must be enlarged to include all those states associated with the 'extended Gibbons–Hawking' boundary term [75]. What we observe in the present work is in line with this proposal; consideration of the enlarged Hilbert space must be a necessary condition for solving the information problem. This point can be elaborated on by utilizing the 3D action obtained in Section 4.4. Let us first examine the symmetry aspect of the reduced theory, which we quote here for convenience:

$$S_{3D} = \int d^3x \, n_0 \sqrt{|\gamma|} \left[ \mathcal{R} + K^2 - K_{mn}K^{mn} \right] - 2 \int_{\partial\mathcal{V}} d^3x \sqrt{|\gamma|} \, \varepsilon K. \tag{120}$$

Now it is to be understood that

$$\gamma_{mn}(t,r,\theta,\phi) = \gamma_{0mn} + \tilde{h}_{mn}(t,\theta,\phi) \tag{121}$$

is substituted into $\gamma_{mn}(t,r,\theta,\phi)$. For simplicity we again consider the infinitesimal fluctuation case. The conformal Killing group will act as the symmetry of the boundary theory. Although this is not an exact symmetry of the bulk theory in the usual sense, it should be the closest analogy one can get for the asymptotic conformal Killing group. The symmetry will generate a set of inequivalent vacua, which will be an important part of the 3D description of the 4D dynamics. The 3D Fock space will then be built on these inequivalent vacua. Previously we have discussed that the rationale for the enlarged Hilbert space is the boundary condition-changing gauge transformations. A change between the reference frames with the accompanying transformation between the adapted coordinate systems brings observer-dependent effects. This is well known, e.g., in the descriptions of a quantized scalar field in a Schwarzschild black hole background by employing Schwarzschild and Kruskal coordinates [12]. Each coordinate system has the associated vacuum: the Schwarzschild vacuum (a.k.a a Boulware vacuum) and the Kruskal vacuum (a.k.a a Hartle–Hawking vacuum). The Kruskal vacuum appears to be thermally radiating to a Schwarzschild observer. The presence of such inequivalent vacua is an essential part of the setup that ultimately leads to the black hole information paradox. By the same token, the BMS transformations introduce many different inequivalent vacua [76] and the BMS charges or their conformal extension will be observable to a Schwarzschild observer. In each of those vacua, it will be possible to perform the transformation between Schwarzschild and Kruskal coordinate systems; the transitions between all these different vacua must be of an information-minimal type [36]. The information-carrying gravitons must be the ones that are associated with the 3D fluctuations.

## 5. One-Loop Renormalization

With the reduction established, one can follow the perturbative renormalization procedure. The renormalization procedures of pure Einstein gravity and an Einstein-scalar system have been carried out in [25,26,41,46,51], respectively. Here we carry out renormalization of an Einstein–Maxwell

system, the most complex system among the three; its matter part itself is a gauge system and this poses additional hurdles not present in, e.g., an Einstein-scalar system; the analysis requires all the techniques (and more) used in the pure Einstein gravity and Einstein-scalar cases. As pointed out in Section 2, the offshell renormalizability didn't work due to the fact that the metric field redefinition a la 't Hooft could not absorb all of the counterterms. In contrast, we will see that the physical sector of the theory can be renormalized. A detailed and explicit analysis of the running of the cosmological constant, Newton's constant, and vector gauge coupling is conducted toward the end.

At the initial stage of the renormalization procedure, the central task is to compute the one-particle-irreducible (1PI) effective action, in particular the counterterms. (Reviews of various methods of computing the effective action can be found in , e.g., [14,77–79].) The counterterms to the ultraviolet divergences were analyzed long ago in [80]. They were determined essentially by dimensional analysis and covariance. However, it is more desirable to directly work them out as one does in non-gravitational theories.[11] In our approach they are calculated by employing the refined BFM Feynman diagrammatic scheme. As a byproduct, the long-known gauge-dependence issue [27–34] is resolved in the present framework.

Consider the Einstein–Maxwell action,[12]

$$S = \int \sqrt{-\hat{g}} \left( \frac{1}{\kappa^2} \hat{R} - \frac{1}{4e^2} \hat{F}_{\mu\nu}^2 \right), \tag{123}$$

where $e$ denotes the vector coupling constant and work with the usual covariant (i.e., non-ADM) Lagrangian formalism. The 3+1 splitting and reduction of the physical states will play a role once the 1PI action is obtained.[13] In the conventional covariant perturbative renormalization analysis, the metric $\hat{g}_{\mu\nu}$ is shifted to

$$\hat{g}_{\mu\nu} \equiv g_{\mu\nu} + h_{\mu\nu}, \tag{124}$$

where $g_{\mu\nu}$ and $h_{\mu\nu}$ denote a solution of the metric field equation and the fluctuation, respectively. For reasons that will become clearer we actually introduce the shift of the following form:

$$\hat{g}_{\mu\nu} \equiv \varphi_{\mu\nu} + g_{\mu\nu} + h_{\mu\nu}, \tag{125}$$

where $\varphi_{\mu\nu}$ is the background field of the refined BFM. Basically the idea of computing the effective action is to integrate out $h_{\mu\nu}$ with the field $\varphi_{\mu\nu} + g_{\mu\nu}$ serving as the eternal legs. Evidently the methodology can be applied to an arbitrary solution $g_{\mu\nu}$.

There are several salient features of the analysis that deserve mentioning. The results of the counterterms are obtained without the help of dimensional analysis or presumed 4D covariance. It turns out that the 4D covariance—which is usually presumed in the related literature—is quite a nontrivial issue, and presuming it hides a good deal of required work under the rug. As will be detailed, taking altogether three different measures is required for ensuring the 4D covariance: removal of the trace part of the metric, employment of the refined BFM, and enforcement of the strong form of the gauge-fixing. The third requirement also provides an important clue as to how to solve the gauge choice-dependence of the effective action.

---

[11]  As we will show, the direct analysis requires several crucial steps. In retrospect, those steps must have obstructed one's attempts to calculate directly in the past.

[12]  To carry out renormalization, one starts with the renormalized form of the action:

$$S = \int \sqrt{-\hat{g}_r} \left( \frac{1}{\kappa_r^2} \hat{R}_r - \frac{1}{4e_r^2} \hat{F}_{r\mu\nu}^2 \right), \tag{122}$$

where the renormalized quantities are indicated by the subscript $r$ that has been omitted from (123) for simplicity of notation.

[13]  This is so at two- and higher- loops. At one-loop, the problematic Riemann tensor square term can be expressed in terms of other terms through the topological identity as discussed in Section 2.

In Section 5.1 we show how to expand the action around the given background by utilizing functional differentiation. We also elaborate on the gauge-fixing. In Section 5.2 the first several relatively simple diagrams and their relevant vertices are identified. In Section 5.3 we consider a flat background carrying out the explicit one-loop counterterm computation by taking $g_{\mu\nu} = \eta_{\mu\nu}$.[14] We directly calculate the counterterms in the refined background field method; dimensional analysis and covariance play the subsidiary role of result-checking. As an unexpected spin-off of our direct approach, we will see how the long-known problem of the gauge choice-dependence of the effective action arises in the present framework via inspection of a certain diagram that yields a non-covariant expression. The origin of the gauge choice-dependence is attributed to the limitation of the background field method, refined or not, in a gravitational system; the problem is resolved by enforcing the strong form of the gauge condition. In Section 5.4 we examine the renormalization of the Newton's, cosmological, and vector coupling constants. We then carry out the renormalization by a metric field redefinition. The two- and higher-loop aspects are commented on. Dimensional regularization has a technical subtlety (elaborated below); the flat propagator yields vanishing results for the vacuum-to-vacuum and tadpole diagrams. For this reason the shifts in the constants are introduced through finite renormalization. The analysis shows that the cosmological constant is generically generated and required for the renormalizability. This in turn suggests that the cosmological constant should be included in the starting renormalized action. Once it is included, it can be treated as the "graviton mass" term. With this arrangement, the vacuum-to-vacuum and tadpole diagrams yield non-vanishing results. For an illustration, we carry out in Section 5.5 the beta function calculation of the vector coupling in this alternative procedure of the renormalization. The analysis yields the same result as that in the literature.

*5.1. Gravity Sector*

Let us first review how the action is expanded under

$$\hat{g}_{\mu\nu} \equiv h_{\mu\nu} + \tilde{g}_{\mu\nu}. \tag{126}$$

We illustrate the procedure by obtaining the part quadratic in the fluctuation $h_{\mu\nu}$. As in Section 4.1, the main tool is the functional Taylor expansion:

$$S(\tilde{g}_{\mu\nu}+h_{\mu\nu}) = S(\tilde{g}_{\mu\nu}) + \frac{1}{2} \int \int h_{\alpha\beta}h_{\mu\nu} \frac{\delta}{\delta\hat{g}_{\alpha\beta}} \frac{\delta S}{\delta\hat{g}_{\mu\nu}}\bigg|_{\hat{g}_{\rho\sigma}=\tilde{g}_{\rho\sigma}} + \cdots, \tag{127}$$

where $|_{\hat{g}_{\rho\sigma}=\tilde{g}_{\rho\sigma}}$ denotes the substitution $\hat{g}_{\rho\sigma} = \tilde{g}_{\rho\sigma}$ after taking the derivative; it will be suppressed in what follows. For the quadratic terms[15]

$$\frac{1}{2} \int \int h_{\alpha\beta}h_{\mu\nu} \frac{\delta}{\delta\hat{g}_{\alpha\beta}} \frac{\delta S}{\delta\hat{g}_{\mu\nu}} = \frac{1}{2} \int h_{\alpha\beta}h_{\mu\nu} \frac{\delta}{\delta\hat{g}_{\alpha\beta}} \left[ -\sqrt{-\hat{g}} \ (\hat{R}^{\mu\nu} - \frac{1}{2}\hat{g}^{\mu\nu}\hat{R}) \right], \tag{128}$$

let us consider

$$\int h_{\alpha\beta}h_{\mu\nu} \sqrt{-\hat{g}} \ \frac{\delta}{\delta\hat{g}_{\alpha\beta}} \hat{R}^{\mu\nu} = \int h_{\alpha\beta}h_{\mu\nu} \sqrt{-\hat{g}} \left[ \hat{R}^{\mu\mu'\nu\nu'} \frac{\delta}{\delta\hat{g}_{\alpha\beta}} \hat{g}_{\mu'\nu'} + \hat{g}_{\mu'\nu'} \frac{\delta}{\delta\hat{g}_{\alpha\beta}} \hat{R}^{\mu\mu'\nu\nu'} \right]. \tag{129}$$

---

[14]   The analysis can also be viewed as the computation of the divergences in a curved background; the flat space analysis captures them since the ultraviolet divergence is a short-distance phenomenon.

[15]   The linear terms are removed by appropriate counterterms (see, e.g., [81] for the comments on this point).

The first term contributes to the first Riemann tensor-containing term in the final result given in (134) below. The second term can be further expanded to

$$\int h_{\alpha\beta} h_{\mu\nu} \sqrt{-\hat{g}} \, \hat{g}_{\mu'\nu'} \frac{\delta}{\delta \hat{g}_{\alpha\beta}} \hat{R}^{\mu\mu'\nu\nu'} \tag{130}$$

$$= \int h_{\alpha\beta} h_{\mu\nu} \sqrt{-\hat{g}} \, \hat{g}_{\mu'\nu'} \left[ \hat{R}^{\mu}{}_{\kappa_1\kappa_2\kappa_3} \frac{\delta}{\delta \hat{g}_{\alpha\beta}} \hat{g}^{\kappa_1\mu'} \hat{g}^{\kappa_2\nu} \hat{g}^{\kappa_3\nu'} + \hat{g}^{\kappa_1\mu'} \hat{g}^{\kappa_2\nu} \hat{g}^{\kappa_3\nu'} \frac{\delta}{\delta \hat{g}_{\alpha\beta}} \hat{R}^{\mu}{}_{\kappa_1\kappa_2\kappa_3} \right].$$

The second term on the right-hand side of (130) can be computed by using (36). Previously we have utilized the full nonlinear form of the de Donder gauge (10). With the metric shifted as in (126), the gauge-fixing condition (7) is translated, at the linear level, into

$$\tilde{\nabla}_\nu h^{\mu\nu} - \frac{1}{2} \tilde{\nabla}^\mu h = 0 \tag{131}$$

where raising or lowering is carried out with $\tilde{g}_{\mu\nu}$. For the path integral we add the following gauge-fixing action,

$$\mathcal{L}_{g.f.} = -\frac{1}{2} \left[ \tilde{\nabla}_\nu h^{\mu\nu} - \frac{1}{2} \tilde{\nabla}^\mu h \right]^2. \tag{132}$$

By adding a gauge-fixing term, one gets the following kinetic terms:

$$2\kappa^2 \mathcal{L}_{kin} = \sqrt{-\tilde{g}} \left( -\frac{1}{2} \tilde{\nabla}_\gamma h^{\alpha\beta} \tilde{\nabla}^\gamma h_{\alpha\beta} + \frac{1}{4} \tilde{\nabla}_\gamma h^{\alpha}_{\alpha} \tilde{\nabla}^\gamma h^{\beta}_{\beta} \right). \tag{133}$$

Including the rest of the quadratic terms, one gets

$$2\kappa^2 \sqrt{-\tilde{g}} \, \mathcal{L} = \sqrt{-\tilde{g}} \left( -\frac{1}{2} \tilde{\nabla}_\gamma h^{\alpha\beta} \tilde{\nabla}^\gamma h_{\alpha\beta} + \frac{1}{4} \tilde{\nabla}_\gamma h^{\alpha}_{\alpha} \tilde{\nabla}^\gamma h^{\beta}_{\beta} \right.$$
$$\left. + h_{\alpha\beta} h_{\gamma\delta} \tilde{R}^{\alpha\gamma\beta\delta} - h_{\alpha\beta} h^{\beta}{}_{\gamma} \tilde{R}^{\kappa\alpha\gamma}{}_{\kappa} - h^{\alpha}{}_{\alpha} h_{\beta\gamma} \tilde{R}^{\beta\gamma} - \frac{1}{2} h^{\alpha\beta} h_{\alpha\beta} \tilde{R} + \frac{1}{4} h^{\alpha}_{\alpha} h^{\beta}_{\beta} \tilde{R} + \cdots \right). \tag{134}$$

Several comments are in order. In [25], the conventional BFM was employed and the counterterms turned out noncovariant. Although the double shift of the metric was implemented, it is not that of the refined BFM, (125). The difference is as follows. Since a flat case was considered in [25], we will focus on the difference in the flat case. In [25], the action is first expanded around the given background, i.e., a flat spacetime in the present case. This way one loses some of the terms. To see this, let us examine the analysis in [25] more closely. In the conventional BFM, one shifts the metric according to (124):

$$\hat{g}_{\mu\nu} \equiv g_{\mu\nu} + h_{\mu\nu}, \tag{135}$$

where $g_{\mu\nu} = \eta_{\mu\nu}$ presently. Then instead of (134), one gets

$$2\kappa^2 \sqrt{-g} \, \mathcal{L} = \sqrt{-g} \left( -\frac{1}{2} \nabla_\gamma h^{\alpha\beta} \nabla^\gamma h_{\alpha\beta} + \frac{1}{4} \nabla_\gamma h^{\alpha}_{\alpha} \nabla^\gamma h^{\beta}_{\beta} \right.$$
$$\left. + h_{\alpha\beta} h_{\gamma\delta} R^{\alpha\gamma\beta\delta} - h_{\alpha\beta} h^{\beta}{}_{\gamma} R^{\kappa\alpha\gamma}{}_{\kappa} - h^{\alpha}{}_{\alpha} h_{\beta\gamma} R^{\beta\gamma} - \frac{1}{2} h^{\alpha\beta} h_{\alpha\beta} R + \frac{1}{4} h^{\alpha}_{\alpha} h^{\beta}_{\beta} R + \cdots \right) \tag{136}$$

with the following gauge-fixing

$$-\frac{1}{2} \left[ \nabla_\nu h^{\mu\nu} - \frac{1}{2} \nabla^\mu h \right]^2. \tag{137}$$

Since the background is flat, all the terms in the second line of (136) vanish, and this shows that one loses some of the terms in this approach. If the second line is the only loss one may hope (with adoption of the traceless propagator) for the covariance. However, the loss is more serious; all

of the Christoffel symbols, for instance, in the covariant derivatives in (136) are lost as well. In the conventional BFM one considers the shift of the form

$$h_{\mu\nu} \to h_{\mu\nu} + \varphi_{\mu\nu}, \tag{138}$$

after (135) and losing the terms. The loss of the terms, with the employment of the traceful propagator, is to blame for the noncovariance of the outcome of the diagram calculations in the conventional approach.

### 5.2. Refined BFM-Based Loop Computation Setup

With the action (134), it is now possible to go ahead and compute various diagrams and their counterterms. As observed in [25,26] there is a cautious step that one must take. It was noticed [43,44] that the path integral is ill-defined due to the presence of the trace piece of the fluctuation metric. Once the trace piece is removed, it is possible to proceed with the direct Feynman diagrammatic method as one normally does in non-gravitational theories. To maintain the 4D covariance of the 1PI action and in particular of the counterterms, it is necessary to employ the refined BFM. In the refined BFM, one shifts the metric according to (125) from the beginning instead of shifting according to (124) followed by (138). (The latter procedure loses terms and thus covariance as explained above.)

We first lay broader outlines of the counter-term computation in an arbitrary unperturbed metric $g_{\mu\nu}$, i.e., a solution of the metric field equation. Afterwards we illustrate the procedure with a flat background. For the perturbative analysis of an Einstein–Maxwell system in the background field method, one splits the original fields $(\hat{g}_{\mu\nu}, \hat{A})$ into

$$\hat{g}_{\mu\nu} \equiv h_{\mu\nu} + \tilde{g}_{\mu\nu} \quad , \quad \hat{A}_\mu \equiv a_\mu + \tilde{A}_\mu \tag{139}$$

where $(h_{\mu\nu}, a_\mu)$ denote the fluctuation fields. The graviton propagator associated with the traceless fluctuation mode [19,26,41] is given by

$$< h_{\mu\nu}(x_1)h_{\rho\sigma}(x_2) > \quad = \quad \tilde{P}_{\mu\nu\rho\sigma} \, \tilde{\Delta}(x_1 - x_2), \tag{140}$$

where the tensor $\tilde{P}_{\mu\nu\rho\sigma}$ is given by

$$\tilde{P}_{\mu\nu\rho\sigma} \quad \equiv \quad \frac{(2\kappa^2)}{2}\left(\tilde{g}_{\mu\rho}\tilde{g}_{\nu\sigma} + \tilde{g}_{\mu\sigma}\tilde{g}_{B\nu\rho} - \frac{1}{2}\tilde{g}_{\mu\nu}\tilde{g}_{\rho\sigma}\right). \tag{141}$$

The value $\tilde{\Delta}(x_1 - x_2)$ denotes the propagator for a scalar theory in the background metric $\tilde{g}_{\mu\nu}$. The full propagator for the vector field is

$$< a_\mu(x_1)a_\nu(x_2) >= e^2 \tilde{g}_{\mu\nu} \, \tilde{\Delta}(x_1 - x_2). \tag{142}$$

As will be shown later, it is possible to formally construct $\tilde{\Delta}(x_1 - x_2)$ in a closed-form and with it some of the diagrams can be effectively computed by employing the full propagator (140) (as well as the full propagator of the Maxwell sector). The perturbative analysis by employing the full propagators will be called the "first-layer" perturbation.[16] We will see the use of the full tensor $\tilde{P}_{\mu\nu\rho\sigma}$ and $\tilde{\Delta}(x_1 - x_2)$ in one of the computations, the example of the first-layer perturbation below. For other diagrams, in

---

[16] The first-layer perturbation should be particularly useful for two- and higher-loop analyses.

particular, the vacuum-to-vacuum amplitudes, one may employ the "second-layer" perturbation[17] by splitting $\tilde{g}, \tilde{A}_\mu$ into

$$\tilde{g}_{\mu\nu} \equiv \varphi_{\mu\nu} + g_{\mu\nu} \quad , \quad \tilde{A}_\mu \equiv A_\mu + A_{0\mu}, \tag{143}$$

where $\varphi_{\mu\nu}, A_\mu$ represents the background fields and $g_{\mu\nu}, A_{0\mu}$ the unperturbed fields—namely, the classical solutions. For instance, we will take $g_{\mu\nu} = \eta_{\mu\nu}, A_{0\mu} = 0$ in the flat spacetime analysis in the next subsection. For a given diagram it often suffices, for a low-order evaluation, to replace $\tilde{P}_{\mu\nu\rho\sigma}$ by

$$\tilde{P}_{\mu\nu\rho\sigma} \simeq P_{\mu\nu\rho\sigma} \equiv \frac{(2\kappa^2)}{2}\left(g_{\mu\rho}g_{\nu\sigma} + g_{\mu\sigma}g_{\nu\rho} - \frac{1}{2}g_{\mu\nu}g_{\rho\sigma}\right), \tag{144}$$

where $P_{\mu\nu\rho\sigma}$ is the leading-order $\varphi_{\mu\nu}$-expansion of $\tilde{P}_{\mu\nu\rho\sigma}$. For the divergence analysis one can use $\tilde{\Delta}(x_1 - x_2) \simeq \Delta(x_1 - x_2)$ where $\Delta(x_1 - x_2)$ denotes the scalar propagator for $g_{\mu\nu} = \eta_{\mu\nu}$,

$$\Delta(x_1 - x_2) = \int \frac{d^4k}{(2\pi)^4} \frac{e^{ik\cdot(x_1-x_2)}}{ik^2}. \tag{145}$$

In this "bottom-up" approach, the diagrams of the first-layer perturbation can be calculated through the second-layer perturbation.

Let us set the stage for the perturbative analysis by expanding the action in terms of the fluctuation fields $h_{\mu\nu}, a_\mu$—one gets, including the gauge-fixing and ghost terms,

$$S = \int \left(\frac{1}{\kappa^2}\mathcal{L}_{grav} + \mathcal{L}_{matter}\right) \tag{146}$$

where

$$\begin{aligned}
\kappa^2 \mathcal{L}_{grav} = {} &\frac{1}{2}\sqrt{-\tilde{g}}\left(-\frac{1}{2}\tilde{\nabla}_\gamma h^{\alpha\beta}\tilde{\nabla}^\gamma h_{\alpha\beta} + \frac{1}{4}\tilde{\nabla}_\gamma h^\alpha_\alpha \tilde{\nabla}^\gamma h^\beta_\beta\right.\\
&\left. + h_{\alpha\beta}h_{\gamma\delta}\tilde{R}^{\alpha\gamma\beta\delta} - h_{\alpha\beta}h^\beta_{\ \gamma}\tilde{R}^{\kappa\alpha\gamma}_{\quad\ \kappa} - h^\alpha_{\ \alpha}h_{\beta\gamma}\tilde{R}^{\beta\gamma} - \frac{1}{2}h^{\alpha\beta}h_{\alpha\beta}\tilde{R} + \frac{1}{4}h^\alpha_\alpha h^\beta_\beta \tilde{R} + \cdots\right)\\
&- \tilde{\nabla}^\nu \bar{C}^\mu \tilde{\nabla}_\nu C_\mu + \tilde{R}_{\mu\nu}\bar{C}^\mu C^\nu - \frac{1}{e^2}\omega^*\tilde{\nabla}^\mu \tilde{F}_{\mu\nu}C^\nu - \frac{1}{e^2}\omega^*\tilde{F}_{\mu\nu}\tilde{\nabla}^\mu C^\nu + \cdots
\end{aligned} \tag{147}$$

and

$$\begin{aligned}
e^2 \mathcal{L}_{matter} = {} &-\frac{1}{4}\sqrt{-\tilde{g}}\left[\tilde{g}^{\mu\nu}\tilde{g}^{\rho\sigma} - \tilde{g}^{\mu\nu}h^{\rho\sigma} - \tilde{g}^{\rho\sigma}h^{\mu\nu} + \frac{1}{2}\tilde{g}^{\mu\nu}\tilde{g}^{\rho\sigma}h + \tilde{g}^{\mu\nu}h^{\rho\kappa}h^\sigma_\kappa + \tilde{g}^{\rho\sigma}h^{\mu\kappa}h^\nu_\kappa \right.\\
&\left. -\frac{1}{2}\tilde{g}^{\mu\nu}hh^{\rho\sigma} - \frac{1}{2}\tilde{g}^{\rho\sigma}hh^{\mu\nu} + h^{\mu\nu}h^{\rho\sigma} + \frac{1}{8}\tilde{g}^{\mu\nu}\tilde{g}^{\rho\sigma}(h^2 - 2h_{\kappa_1\kappa_2}h^{\kappa_1\kappa_2})\right]\left(f_{\mu\rho}f_{\nu\sigma} + 2f_{\mu\rho}\tilde{F}_{\nu\sigma} + \tilde{F}_{\mu\rho}\tilde{F}_{\nu\sigma}\right)\\
&- \frac{1}{2}\sqrt{-\tilde{g}}\,(\tilde{\nabla}_\kappa a^\kappa)^2 - \tilde{\nabla}\omega^* \tilde{\nabla}\omega + \cdots,
\end{aligned} \tag{148}$$

---

[17]    The need for the second-layer perturbation for the gravity sector was discussed, e.g., in [25,26]. The second-layer perturbation is not necessary in non-gravitational theories.

where the indices are raised and lowered by $\tilde{g}^{\mu\nu}$ and $\tilde{g}_{\mu\nu}$, respectively; $(C^{\kappa}, \omega)$ are the ghosts for the diffeomorphism and vector gauge transformation, respectively.[18] Putting it all together, a more useful way of writing the action (146) is

$$S \equiv S_k + S_v, \tag{150}$$

where $S_k, S_v$ denote the kinetic part and the vertices, respectively; they are given by

$$
\begin{aligned}
S_k &= \int \sqrt{-\tilde{g}} \frac{1}{2\kappa^2} \Big( -\frac{1}{2} \tilde{\nabla}_\gamma h^{\alpha\beta} \tilde{\nabla}^\gamma h_{\alpha\beta} + \frac{1}{4} \tilde{\nabla}_\gamma h_\alpha^\alpha \tilde{\nabla}^\gamma h_\beta^\beta \Big) - \frac{1}{4e^2} \sqrt{-\tilde{g}} \left( \tilde{g}^{\mu\nu} \tilde{g}^{\rho\sigma} f_{\mu\rho} f_{\nu\sigma} \right) \\
&\quad - \frac{1}{2e^2} \sqrt{-\tilde{g}} \left( \tilde{\nabla}_\kappa a^\kappa \right)^2 + \frac{1}{2\kappa^2} \sqrt{-\tilde{g}} \left( -\tilde{\nabla}^\nu \bar{C}^\mu \tilde{\nabla}_\nu C_\mu \right) - \frac{1}{e^2} \sqrt{-\tilde{g}} \, \tilde{\nabla}^\rho \omega^* \tilde{\nabla}_\rho \omega
\end{aligned} \tag{151}
$$

and

$$
\begin{aligned}
S_v &= \int \sqrt{-\tilde{g}} \frac{1}{2\kappa^2} \Big( h_{\alpha\beta} h_{\gamma\delta} \tilde{R}^{\alpha\gamma\beta\delta} - h_{\alpha\beta} h^\beta_{\ \gamma} \tilde{R}^{\kappa\alpha\gamma}_{\ \ \ \kappa} - h^\alpha_{\ \alpha} h_{\beta\gamma} \tilde{R}^{\beta\gamma} - \frac{1}{2} h^{\alpha\beta} h_{\alpha\beta} \tilde{R} \\
&\quad + \frac{1}{4} h^\alpha_\beta h^\beta_\alpha \tilde{R} \Big) - \frac{1}{4e^2} \sqrt{-\tilde{g}} (\tilde{g}^{\mu\nu} \tilde{g}^{\rho\sigma}) \Big( 2 f_{\mu\rho} \tilde{F}_{\nu\sigma} + \tilde{F}_{\mu\rho} \tilde{F}_{\nu\sigma} \Big) - \frac{1}{4e^2} \sqrt{-\tilde{g}} \Big[ -\tilde{g}^{\mu\nu} h^{\rho\sigma} \\
&\quad - \tilde{g}^{\rho\sigma} h^{\mu\nu} + \frac{1}{2} \tilde{g}^{\mu\nu} \tilde{g}^{\rho\sigma} h + \tilde{g}^{\mu\nu} h^{\rho\kappa} h^\sigma_\kappa + \tilde{g}^{\rho\sigma} h^{\mu\kappa} h^\nu_\kappa - \frac{1}{2} \tilde{g}^{\mu\nu} h h^{\rho\sigma} - \frac{1}{2} \tilde{g}^{\rho\sigma} h h^{\mu\nu} \\
&\quad + h^{\mu\nu} h^{\rho\sigma} + \frac{1}{8} \tilde{g}^{\mu\nu} \tilde{g}^{\rho\sigma} (h^2 - 2 h_{\kappa_1\kappa_2} h^{\kappa_1\kappa_2}) \Big] \Big( f_{\mu\rho} f_{\nu\sigma} + 2 f_{\mu\rho} \tilde{F}_{\nu\sigma} + \tilde{F}_{\mu\rho} \tilde{F}_{\nu\sigma} \Big) \\
&\quad + \frac{1}{2\kappa^2} \sqrt{-\tilde{g}} \left( +\tilde{R}_{\mu\nu} \bar{C}^\mu C^\nu + \frac{1}{2e^2} \tilde{\nabla}^\mu \omega^* \tilde{F}_{\mu\nu} C^\nu \right) + \cdots .
\end{aligned} \tag{152}
$$

Worth mentioning is how the graviton gauge-fixing has been implemented:

$$- \frac{1}{2} \Big[ \tilde{\nabla}_\nu h^{\mu\nu} - \frac{1}{2} \tilde{\nabla}^\mu h \Big]^2 . \tag{153}$$

This is the refined BFM version of the usual gauge-fixing (137) that is $\tilde{g}_{\mu\nu}$-background non-covariant (although $g_{\mu\nu}$-background-covariant). The physical content of the gauge condition satisfied by $h_{\mu\nu}$ is still (137); the BFM is just a convenience device that allows one to conduct the analysis covariantly (or more so than otherwise). (The field $\varphi_{\mu\nu}$ satisfies the same gauge-fixing) One may expect that with the gauge-fixing (153) the 1PI effective action will come out to be $\tilde{g}_{\mu\nu}$-covariant. This turns out to be naive; we will see that some of the counterterms turn out non-covariant due to the presence of the factors that can be removed by enforcing the strong form of the gauge condition.

Let us outline the steps of the amplitude computation for an arbitrary solution metric $g_{\mu\nu}$ and the renormalization in the curved background $g_{\mu\nu}$. The diagrams that we will consider are classified into four categories in terms of the second-layer perturbation. The first class is the diagrams with both vertices from the graviton sector: the pure gravity sector two-point amplitude and the corresponding ghost-loop diagram in Figure 3. The second is the diagrams with both vertices from the matter sector; several relatively simple matter-involving diagrams are listed in Figure 4a–c. The third is the diagrams with one vertex from the graviton sector and the other from the matter sector, Figure 4d. All of the diagrams so far have "homogeneous" loops, whereas the fourth class of the diagrams in Figure 5 have "inhomogeneous" ones; as we will see they require special care.

---

18 These ghost terms correspond to the following transformations of the fluctuation fields [80]:

$$
\begin{aligned}
h'_{\mu\nu} &= h_{\mu\nu} + (\tilde{g}_{\mu\kappa} \tilde{D}_\nu + \tilde{g}_{\nu\kappa} \tilde{D}_\mu) \eta^\kappa + (h_{\mu\kappa} \tilde{D}_\nu + h_{\nu\kappa} \tilde{D}_\mu) \eta^\kappa + \eta^\kappa \tilde{D}_\kappa h^{\mu\nu} \\
a'_\mu &= a_\mu + \eta^\kappa \tilde{F}_{\kappa\mu} + \tilde{D}_\mu \eta^5 + a_\kappa \tilde{D}_\mu \eta^\mu + \eta^\kappa \tilde{D}_\kappa a_\mu,
\end{aligned} \tag{149}
$$

under $x'^\alpha = x^\alpha - \eta^\alpha$ and the vector gauge transformation with the parameter $-\eta^\kappa \tilde{A}_\kappa + \eta^5$.

To see in more detail how these diagrams arise and the precise form of the corresponding vertices in the Lagrangian let us further expand (150). Because the graviton vertex makes the computation lengthy and tedious, we restrict the maximum number of the graviton external legs to two. In the gravity sector, the simplest diagrams are the second-order (in $\varphi_{\alpha\beta}$) diagrams in Figure 3. With the split given in (143), the kinetic terms themselves yield the vertices for the second-layer perturbation expansion. In the gravity sector, the graviton kinetic term is expanded as

$$2\kappa^2 \mathcal{L}_{grav,kin} = -\frac{1}{2}\partial_\gamma h^{\alpha\beta}\partial^\gamma h_{\alpha\beta} + \frac{1}{4}\partial_\gamma h^\alpha_\alpha \partial^\gamma h^\beta_\beta \tag{154}$$

$$+ \left(2g^{\beta\beta'}\tilde{\Gamma}^{\alpha'\gamma\alpha} - g^{\alpha\beta}\tilde{\Gamma}^{\alpha'\gamma\beta'}\right)\partial_\gamma h_{\alpha\beta}\, h_{\alpha'\beta'} + \left[\frac{1}{2}(g^{\alpha\alpha'}g^{\beta\beta'}\varphi^{\gamma\gamma'} + g^{\beta\beta'}g^{\gamma\gamma'}\varphi^{\alpha\alpha'}\right.$$

$$+ g^{\alpha\alpha'}g^{\gamma\gamma'}\varphi^{\beta\beta'}) - \frac{1}{4}\varphi\, g^{\alpha\alpha'}g^{\beta\beta'}g^{\gamma\gamma'} - \frac{1}{2}g^{\gamma\gamma'}g^{\alpha'\beta'}\varphi^{\alpha\beta} + \frac{1}{4}(-\varphi^{\gamma\gamma'} + \frac{1}{2}\varphi g^{\gamma\gamma'})g^{\alpha\beta}g^{\alpha'\beta'}\right]\partial_\gamma h_{\alpha\beta}\, \partial_{\gamma'} h_{\alpha'\beta'},$$

where the raising and lowering are done by $g^{\mu\nu}$ and $g_{\mu\nu}$, respectively. The terms in the second and third lines serve as the vertices responsible for Figure 3a. The corresponding ghost diagram is given in Figure 3b.

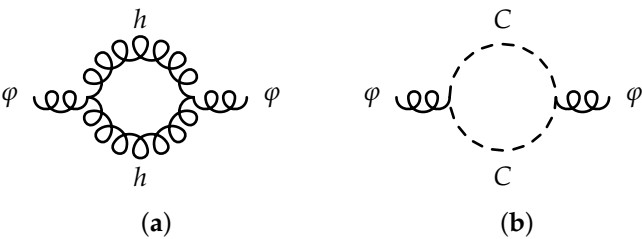

**Figure 3.** Graviton and ghost diagrams (indices on fields suppressed).

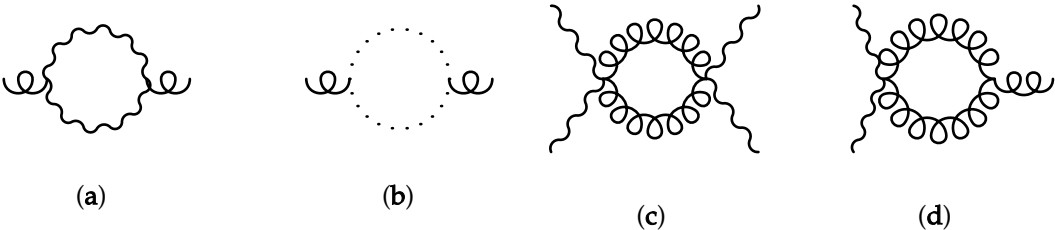

**Figure 4.** Matter-involving diagrams.

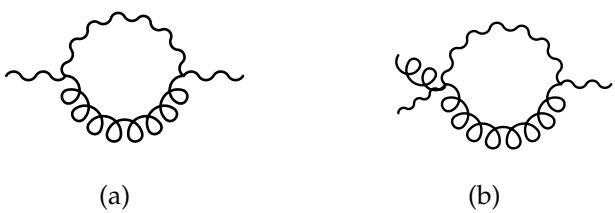

**Figure 5.** Diagrams with inhomogeneous loops.

The vertex, $V_g$, responsible for the diagrams in Figure 3a, is defined by rewriting (154) as

$$\mathcal{L} = \frac{1}{\kappa'^2}\left[-\frac{1}{2}\partial_\gamma h^{\alpha\beta}\partial^\gamma h_{\alpha\beta} + \frac{1}{4}\partial_\gamma h^\alpha_\alpha \partial^\gamma h^\beta_\beta + \mathcal{L}_{V_g}\right], \tag{155}$$

where

$$\kappa'^2 \equiv 2\kappa^2, \tag{156}$$

and ($\mathcal{L}_{V_g}$ and $V_g$ are related by $V_g = \sqrt{-g}\,\mathcal{L}_{V_g}$)

$$V_g \equiv \sqrt{-g}\Big(2g^{\beta\beta'}\tilde{\Gamma}^{\alpha'\gamma\alpha} - g^{\alpha\beta}\tilde{\Gamma}^{\alpha'\gamma\beta'}\Big)\partial_\gamma h_{\alpha\beta}\,h_{\alpha'\beta'} + \sqrt{-g}\Big[\frac{1}{2}(g^{\alpha\alpha'}g^{\beta\beta'}\varphi^{\gamma\gamma'} + g^{\beta\beta'}g^{\gamma\gamma'}\varphi^{\alpha\alpha'}$$

$$+ g^{\alpha\alpha'}g^{\gamma\gamma'}\varphi^{\beta\beta'}) - \frac{1}{4}\varphi\,g^{\alpha\alpha'}g^{\beta\beta'}g^{\gamma\gamma'} - \frac{1}{2}g^{\gamma\gamma'}g^{\alpha'\beta'}\varphi^{\alpha\beta} + \frac{1}{4}(-\varphi^{\gamma\gamma'} + \frac{1}{2}\varphi g^{\gamma\gamma'})g^{\alpha\beta}g^{\alpha'\beta'}\Big]\partial_\gamma h_{\alpha\beta}\,\partial_{\gamma'}h_{\alpha'\beta'}$$

$$+ \sqrt{-\tilde{g}}\Big(h_{\alpha\beta}h_{\gamma\delta}\tilde{R}^{\alpha\gamma\beta\delta} - h_{\alpha\beta}h^\beta{}_\gamma\tilde{R}^{\kappa\alpha\gamma}{}_\kappa - h^\alpha{}_\alpha h_{\beta\gamma}\tilde{R}^{\beta\gamma} - \frac{1}{2}h^{\alpha\beta}h_{\alpha\beta}\tilde{R} + \frac{1}{4}h^\alpha_\alpha h^\beta_\beta\tilde{R}\Big). \tag{157}$$

Concerning the $\tilde{g}_{\mu\nu}$-containing quantities, expansion in terms of $\varphi_{\mu\nu}$ to the appropriate orders is to be understood. The vertex relevant for the ghost-loop diagram can similarly be identified by expanding the terms quadratic in the ghost field:

$$V_C \equiv -\sqrt{-g}\Big[\frac{1}{2}\varphi\partial^\mu\bar{C}^\nu\partial_\mu C_\nu - \tilde{\Gamma}^\lambda_{\mu\nu}(\partial^\mu\bar{C}^\nu C_\lambda - \partial^\mu C^\nu\bar{C}_\lambda)$$

$$- (g^{\nu\beta}\varphi^{\mu\alpha} + g^{\mu\alpha}\varphi^{\nu\beta})\partial_\beta\bar{C}_\alpha\partial_\nu C_\mu\Big] + \sqrt{-g}\,R_{\mu\nu}\bar{C}^\mu C^\nu. \tag{158}$$

The vertices responsible for the diagrams in Figures 4 and 5 can also be obtained by examining the matter part of the action:

$$V_{m1} \equiv -\frac{1}{4}\sqrt{-g}\Big[-g^{\rho\sigma}\varphi^{\mu\nu} - g^{\mu\nu}\varphi^{\rho\sigma}\Big]f_{\mu\rho}f_{\nu\sigma}$$

$$V_{m2} \equiv -\frac{1}{4}\sqrt{-g}\Big[g^{\mu\nu}h^{\rho\kappa}h^\sigma_\kappa + g^{\rho\sigma}h^{\mu\kappa}h^\nu_\kappa + h^{\mu\nu}h^{\rho\sigma} - \frac{1}{4}g^{\mu\nu}g^{\rho\sigma}h_{\kappa_1\kappa_2}h^{\kappa_1\kappa_2}\Big]\tilde{F}_{\mu\rho}\tilde{F}_{\nu\sigma}$$

$$V_{m3} \equiv \frac{1}{2}\sqrt{-g}\Big[g^{\mu\nu}h^{\rho\sigma} + g^{\rho\sigma}h^{\mu\nu}\Big]f_{\mu\rho}F_{\nu\sigma}$$

$$V_{m4} \equiv -\frac{1}{2}\sqrt{-g}\Big[\varphi^{\mu\nu}h^{\rho\sigma} + \varphi^{\rho\sigma}h^{\mu\nu}\Big]f_{\mu\rho}F_{\nu\sigma} \tag{159}$$

Above the trace part of the fluctuation metric, $h \equiv \tilde{g}^{\alpha\beta}h_{\alpha\beta}$, has been set to zero [19,26,41]. Let us lay out the calculation of counterterms to the diagrams in Figures 3–5. The graviton and ghost contributions, respectively, are [26]

$$\Rightarrow \quad -\frac{1}{2}\frac{1}{\kappa'^4}\Big< \Big(\int\Big\{\Big(2g^{\beta\beta'}\tilde{\Gamma}^{\alpha'\gamma\alpha} - g^{\alpha\beta}\tilde{\Gamma}^{\alpha'\gamma\beta'}\Big)\partial_\gamma h_{\alpha\beta}\,h_{\alpha'\beta'}$$

$$+ \Big[\frac{1}{2}(g^{\alpha\alpha'}g^{\beta\beta'}\varphi^{\gamma\gamma'} + g^{\beta\beta'}g^{\gamma\gamma'}\varphi^{\alpha\alpha'} + g^{\alpha\alpha'}g^{\gamma\gamma'}\varphi^{\beta\beta'}) - \frac{1}{2}g^{\gamma\gamma'}g^{\alpha'\beta'}\varphi^{\alpha\beta}$$

$$- \frac{1}{4}\varphi^{\gamma\gamma'}g^{\alpha\beta}g^{\alpha'\beta'}\Big]\partial_\gamma h_{\alpha\beta}\,\partial_{\gamma'}h_{\alpha'\beta'} + \Big(h_{\alpha\beta}h_{\gamma\delta}\tilde{R}^{\alpha\gamma\beta\delta} - h_{\alpha\beta}h^\beta{}_\gamma\tilde{R}^{\kappa\alpha\gamma}{}_\kappa - \frac{1}{2}h^{\alpha\beta}h_{\alpha\beta}\tilde{R}\Big)\Big\}\Big)^2 \Big> \tag{160}$$

and

$$\Rightarrow \quad -\frac{1}{2}\frac{1}{\kappa'^4}\Big< \Big(\int -\Big[\frac{1}{2}\varphi\partial^\mu\bar{C}^\nu\partial_\mu C_\nu - \tilde{\Gamma}^\lambda_{\mu\nu}(\partial^\mu\bar{C}^\nu C_\lambda - \partial^\mu C^\nu\bar{C}_\lambda)$$

$$- (g^{\nu\beta}\varphi^{\mu\alpha} + g^{\mu\alpha}\varphi^{\nu\beta})\partial_\beta\bar{C}_\alpha\partial_\nu C_\mu\Big] + R_{\mu\nu}\bar{C}^\mu C^\nu\Big)^2 \Big>. \tag{161}$$

Above and in what follows "$\Rightarrow$" means that the diagram on the left-hand side leads to the counterterm(s) on the right-hand side after carrying out the contractions appropriately in reflection of the diagram under consideration.

The numerical factors, $-\frac{1}{2}$'s, are the combinatoric factors that arise when the vertices are brought down from the exponential of the path integral. For the gravity sector, the one-loop counterterms

are given by the sum of these two. (The result for the flat case is reviewed in the next subsection.) The diagram in Figure 4a,c has two of the vertices $V_{m1}$ ($V_{m2}$) inserted; the correlator expressions are[19]

$$\text{(diagram)} \Rightarrow -\frac{1}{2} < \left( \int V_{m1} \right)^2 > = -\frac{1}{2} < \left[ \int \frac{1}{4} (g^{\rho\sigma} \varphi^{\mu\nu} + g^{\mu\nu} \varphi^{\rho\sigma})(f_{\mu\rho} f_{\nu\sigma}) \right]^2 >$$

(162)

$$\text{(diagram)} \Rightarrow -\frac{1}{2} < \left( \int V_{m2} \right)^2 > = -\frac{1}{2} < \Big[ \int \frac{1}{4} (g^{\mu\nu} h^{\rho\kappa} h^{\sigma}_{\kappa} + g^{\rho\sigma} h^{\mu\kappa} h^{\nu}_{\kappa} + h^{\mu\nu} h^{\rho\sigma}$$
$$-\frac{1}{4} g^{\mu\nu} g^{\rho\sigma} h_{\kappa_1\kappa_2} h^{\kappa_1\kappa_2}) (\tilde{F}_{\mu\rho} \tilde{F}_{\nu\sigma}) \Big]^2 >$$

(163)

The cross-term diagram in Figure 4d is given by the vacuum expectation value of the two vertices, $V_{m_2}$ and $V_g$:

$$\text{(diagram)} \Rightarrow -< \int V_{m2} \int V_g > = -< \int \left(-\frac{1}{4}\right) \Big[ g^{\mu\nu} h^{\rho\kappa} h^{\sigma}_{\kappa} + g^{\rho\sigma} h^{\mu\kappa} h^{\nu}_{\kappa} + h^{\mu\nu} h^{\rho\sigma}$$
$$-\frac{1}{4} g^{\mu\nu} g^{\mu\nu} h_{\kappa_1\kappa_2} h^{\kappa_1\kappa_2} \Big] \left( \tilde{F}_{\mu\rho} \tilde{F}_{\nu\sigma} \right) \times \frac{1}{\kappa'^2} \int \Big\{ \left( 2 g^{\beta\beta'} \tilde{\Gamma}^{\alpha'\gamma\alpha} - g^{\alpha\beta} \tilde{\Gamma}^{\alpha'\gamma\beta'} \right) \partial_\gamma h_{\alpha\beta} \, h_{\alpha'\beta'}$$
$$+ \Big[ \frac{1}{2} (g^{\alpha\alpha'} g^{\beta\beta'} \varphi^{\gamma\gamma'} + g^{\beta\beta'} g^{\gamma\gamma'} \varphi^{\alpha\alpha'} + g^{\alpha\alpha'} g^{\gamma\gamma'} \varphi^{\beta\beta'}) - \frac{1}{2} g^{\gamma\gamma'} g^{\alpha'\beta'} \varphi^{\alpha\beta}$$
$$-\frac{1}{4} \varphi^{\gamma\gamma'} g^{\alpha\beta} g^{\alpha'\beta'} \Big] \partial_\gamma h_{\alpha\beta} \, \partial_{\gamma'} h_{\alpha'\beta'} + \left( h_{\alpha\beta} h_{\gamma\delta} \tilde{R}^{\alpha\gamma\beta\delta} - h_{\alpha\beta} h^{\beta}_{\gamma} \tilde{R}^{\kappa\alpha\gamma}_{\kappa} - \frac{1}{2} h^{\alpha\beta} h_{\alpha\beta} \tilde{R} \right) \Big\} >$$

(164)

For most of the diagrams that we will consider, the structures of the vertices allow one to use, for the given order, the approximate form of the tensor $\tilde{\Delta}(x_1 - x_2)$, namely, $\Delta(x_1 - x_2)$.

The diagrams with the inhomogeneous loops provide an example of the direct first-layer calculation. For them it is necessary, for covariance, to use the full propagator in (140), a step not needed for the other diagrams for which it was so far sufficient to use the leading-order propagator. In the first-layer perturbation, the graph to calculate can be represented by thickened lines:

The external lines represent the full fields, i.e., the fields with tildes, and by the same token, the internal lines the full propagators. The two diagrams in Figure 5 are the first two terms that result from, so to speak, $\varphi_{\alpha\beta}$-expanding the graph in Figure 6; there are additional contributions (not drawn here) coming from the internal lines when the full propagators are used. As for those diagrams in Figure 5, they can be set up in a manner similar to the others:

$$\text{(diagram)} \Rightarrow -\frac{1}{2} < \left( \int V_{m3} \right)^2 > = -\frac{1}{2} < \left( \int \frac{1}{2} (g^{\mu\nu} h^{\rho\sigma} + g^{\rho\sigma} h^{\mu\nu}) f_{\mu\rho} F_{\nu\sigma} \right)^2 >$$

$$\text{(diagram)} \Rightarrow -< \int V_{m3} \int V_{m4} > = -< \int \frac{1}{2} (g^{\mu\nu} h^{\rho\sigma} + g^{\rho\sigma} h^{\mu\nu}) f_{\mu\rho} F_{\nu\sigma}$$
$$\times \int \frac{-1}{2} (\varphi^{\mu'\nu'} h^{\rho'\sigma'} + \varphi^{\rho'\sigma'} h^{\mu'\nu'}) f_{\mu'\rho'} F_{\nu'\sigma'} > .$$

(165)

We will come back to these inhomogeneous ones in Section 5.3 where we show a convenient and effective way of calculating all the contributions, including those arising from the full internal propagators.

---

[19]  The vector coupling constant $e^2$ is often suppressed.

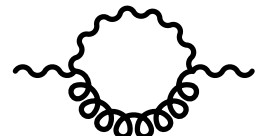

**Figure 6.** First-layer perturbation diagram.

*5.3. Flat Space Analysis*

In what follows we present the explicit flat spacetime computations for the two-point diagrams considered for a generic background $g_{\mu\nu}$ in the previous section. (The analysis of the vacuum-to-vacuum amplitudes and tadpoles will be presented in Section 5.4) Although the techniques employed are similar to those used in the pure gravity [26] and gravity-scalar [51] cases, the present analysis has several additional complications due to the fact that the matter part itself is a gauge system.

Let us consider a flat background:

$$\hat{g}_{\mu\nu} \equiv h_{\mu\nu} + \tilde{g}_{\mu\nu} \quad , \quad \hat{A}_\mu \equiv a_\mu + \tilde{A}_\mu, \tag{166}$$

where

$$\tilde{g}_{\mu\nu} \equiv \varphi_{\mu\nu} + g_{\mu\nu} \quad , \quad \tilde{A}_\mu \equiv A_\mu + A_{0\mu}, \tag{167}$$

with

$$g_{\mu\nu} = \eta_{\mu\nu} \quad , \quad A_{0\mu} = 0. \tag{168}$$

5.3.1. Two-Point Diagrams

Consider the ghost loop diagram in Figure 3b first. In the flat spacetime the ghost vertex takes

$$V_C = -\left[ -\tilde{\Gamma}^\lambda_{\mu\nu}(-C_\lambda \partial^\mu \bar{C}^\nu + \bar{C}_\lambda \partial^\mu C^\nu) - (\eta^{\nu\beta}\varphi^{\mu\alpha} + \eta^{\mu\alpha}\varphi^{\nu\beta})\partial_\beta \bar{C}_\alpha \partial_\nu C_\mu \right] + R_{\mu\nu}\bar{C}^\mu C^\nu.$$

It is convenient to define

$$V_C = V_{C,I} + V_{C,II}, \tag{169}$$

with

$$V_{C,I} \equiv -\left[ -\tilde{\Gamma}^\lambda_{\mu\nu}(-C_\lambda \partial^\mu \bar{C}^\nu + \bar{C}_\lambda \partial^\mu C^\nu) - (\eta^{\nu\beta}\varphi^{\mu\alpha} + \eta^{\mu\alpha}\varphi^{\nu\beta})\partial_\beta \bar{C}_\alpha \partial_\nu C_\mu \right]$$
$$V_{C,II} \equiv R_{\mu\nu}\bar{C}^\mu C^\nu. \tag{170}$$

The correlator to be computed is

$$-\frac{1}{2}\frac{1}{\kappa'^4} < \left( \int V_{C,I} + V_{C,II} \right)^2 > = -\frac{1}{2}\frac{1}{\kappa'^4} < \left\{ \int \left[ -\tilde{\Gamma}^\lambda_{\mu\nu}(\partial^\mu \bar{C}^\nu C_\lambda - \partial^\mu C^\nu \bar{C}_\lambda) \right. \right.$$
$$\left. \left. -(\eta^{\nu\beta}\varphi^{\mu\alpha} + \eta^{\mu\alpha}\varphi^{\nu\beta})\partial_\beta \bar{C}_\alpha \partial_\nu C_\mu \right] - R_{\mu\nu}\bar{C}^\mu C^\nu \right\}^2 > . \tag{171}$$

The dimensional analysis and covariance can be utilized to recognize the covariance of the final results. To see this consider, e.g., $< (\int V_{C,I})^2 >$; a direct calculation yields

$$-\frac{1}{2}\frac{1}{\kappa'^4} < \left( \int V_{C,I} \right)^2 > = -\frac{1}{2}\frac{\Gamma(\epsilon)}{(4\pi)^2} \int \left[ -\frac{2}{15}\partial^2 \varphi_{\mu\nu}\partial^2 \varphi^{\mu\nu} + \frac{4}{15}\partial^2 \varphi^{\alpha\kappa}\partial_\kappa \partial_\sigma \varphi^\sigma_\alpha - \frac{1}{30}(\partial_\alpha \partial_\beta \varphi^{\alpha\beta})^2 \right], \tag{172}$$

where the parameter $\varepsilon$ is given by

$$D = 4 - 2\varepsilon. \tag{173}$$

Lengthy index contractions were carried out with the help of the Mathematica package xAct'xTensor'. Invoking dimensional analysis and covariance, one expects the result to be a sum of $R^2$ and $R^2_{\mu\nu}$ to the second order of $\varphi_{\rho\sigma}$ with appropriate coefficients. With the traceless condition $\varphi = 0$ explicitly enforced, $R^2$ and $R^2_{\mu\nu}$ are given, to the second order in $\varphi_{\alpha\beta}$, by

$$\begin{aligned}
R^2 &= \partial_\mu \partial_\nu \varphi^{\mu\nu} \partial_\rho \partial_\sigma \varphi^{\rho\sigma} \\
R_{\alpha\beta} R^{\alpha\beta} &= \frac{1}{4}\Big[ \partial^2 \varphi^{\mu\nu} \partial^2 \varphi_{\mu\nu} - 2\partial^2 \varphi^{\alpha\kappa} \partial_\kappa \partial_\sigma \varphi_\alpha^\sigma + 2(\partial_\mu \partial_\nu \varphi^{\mu\nu})^2 \Big];
\end{aligned} \tag{174}$$

comparing with these it follows that

$$-\frac{1}{2}\frac{1}{\kappa'^4} < \left( \int V_{C,I} \right)^2 >= -\frac{1}{2}\frac{\Gamma(\epsilon)}{(4\pi)^2} \int \Big[ -\frac{8}{15}\tilde{R}_{\alpha\beta}\tilde{R}^{\alpha\beta} + \frac{7}{30}\tilde{R}^2 \Big]. \tag{175}$$

The tildes will be omitted from now on. Completing the other terms in (171), one gets

 $\Rightarrow$ $-\frac{1}{2}\frac{\Gamma(\epsilon)}{(4\pi)^2} \int \Big[ \frac{7}{15}R_{\mu\nu}R^{\mu\nu} + \frac{17}{30}R^2 \Big].$ $\qquad(176)$

The vertex $V_g$, which is relevant for the graviton-loop diagram in Figure 3a, takes, in the flat space,

$$\begin{aligned}
V_g \equiv & \left( 2\eta^{\beta\beta'}\tilde{\Gamma}^{\alpha'\gamma\alpha} - \eta^{\alpha\beta}\tilde{\Gamma}^{\alpha'\gamma\beta'} \right)\partial_\gamma h_{\alpha\beta}\, h_{\alpha'\beta'} + \Big[ \frac{1}{2}(\eta^{\alpha\alpha'}\eta^{\beta\beta'}\varphi^{\gamma\gamma'} + \eta^{\beta\beta'}\eta^{\gamma\gamma'}\varphi^{\alpha\alpha'} \\
& + \eta^{\alpha\alpha'}\eta^{\gamma\gamma'}\varphi^{\beta\beta'}) - \frac{1}{2}\eta^{\gamma\gamma'}\eta^{\alpha'\beta'}\varphi^{\alpha\beta} - \frac{1}{4}\varphi^{\gamma\gamma'}\eta^{\alpha\beta}\eta^{\alpha'\beta'} \Big]\partial_\gamma h_{\alpha\beta}\,\partial_{\gamma'} h_{\alpha'\beta'} \\
& + \left( h_{\alpha\beta}h_{\gamma\delta}\tilde{R}^{\alpha\gamma\beta\delta} - h_{\alpha\beta}h^\beta_{\ \gamma}\tilde{R}^{\kappa\alpha\gamma}_{\ \ \ \kappa} - \frac{1}{2}h^{\alpha\beta}h_{\alpha\beta}\tilde{R} \right).
\end{aligned} \tag{177}$$

Let us define:

$$\begin{aligned}
V_{g,I} &\equiv \left( 2\eta^{\beta\beta'}\tilde{\Gamma}^{\alpha'\gamma\alpha} - \eta^{\alpha\beta}\tilde{\Gamma}^{\alpha'\gamma\beta'} \right)\partial_\gamma h_{\alpha\beta}\, h_{\alpha'\beta'} \\
V_{g,II} &\equiv \Big[ \frac{1}{2}(\eta^{\alpha\alpha'}\eta^{\beta\beta'}\varphi^{\gamma\gamma'} + \eta^{\beta\beta'}\eta^{\gamma\gamma'}\varphi^{\alpha\alpha'} + \eta^{\alpha\alpha'}\eta^{\gamma\gamma'}\varphi^{\beta\beta'}) \\
& \quad - \frac{1}{4}\varphi\,\eta^{\alpha\alpha'}\eta^{\beta\beta'}\eta^{\gamma\gamma'} - \frac{1}{2}\eta^{\gamma\gamma'}\eta^{\alpha'\beta'}\varphi^{\alpha\beta} \\
& \quad + \frac{1}{4}(-\varphi^{\gamma\gamma'} + \frac{1}{2}\varphi\eta^{\gamma\gamma'})\eta^{\alpha\beta}\eta^{\alpha'\beta'} \Big]\partial_\gamma h_{\alpha\beta}\,\partial_{\gamma'} h_{\alpha'\beta'}
\end{aligned} \tag{178}$$

$$V_{g,III} = \sqrt{-\tilde{g}}\left( h_{\alpha\beta}h_{\gamma\delta}\tilde{R}^{\alpha\gamma\beta\delta} - h_{\alpha\beta}h^\beta_{\ \gamma}\tilde{R}^{\kappa\alpha\gamma}_{\ \ \ \kappa} - \frac{1}{2}h^{\alpha\beta}h_{\alpha\beta}\tilde{R} \right). \tag{179}$$

Again, by employing the traceless propagator one can show that

 $\Rightarrow$ $-\frac{1}{2}\frac{\Gamma(\epsilon)}{(4\pi)^2} \int \Big[ -\frac{23}{20}R_{\mu\nu}R^{\mu\nu} - \frac{23}{40}R^2 \Big].$ $\qquad(180)$

The correlators for the matter-involving sector have also been outlined for an arbitrary background in the previous section. The diagrams in Figure 4a–c lead, for the flat spacetime, to

$$\text{(diagram)} \quad \Rightarrow \quad \frac{\Gamma(\epsilon)}{(4\pi)^2} \int \left( \frac{1}{30} R^2 - \frac{1}{10} R_{\alpha\beta} R^{\alpha\beta} \right)$$

$$\text{(diagram)} \quad \Rightarrow \quad -\frac{\Gamma(\epsilon)}{(4\pi)^2} \frac{1}{15} \int R_{\alpha\beta} R^{\alpha\beta}$$

$$\text{(diagram)} \quad \Rightarrow \quad \frac{\kappa'^4 \Gamma(\epsilon)}{(4\pi)^2} \frac{3}{64} \int (F_{\alpha\beta} F^{\alpha\beta})^2. \tag{181}$$

These results are covariant as expected. On the other hand, the direct calculation of the diagram in Figure 4d yields

$$\text{(diagram)} \Big|_{V_{g,I}+V_{g,II}} \quad \Rightarrow \quad \frac{\kappa'^2 \Gamma(\epsilon)}{(4\pi)^2} \int \left( \frac{1}{16} F_{\mu\nu} F^{\mu\nu} \partial_\alpha \partial_\beta \varphi^{\alpha\beta} + \frac{1}{2} F_{\mu\kappa} F_\nu{}^\kappa \partial^2 \varphi^{\mu\nu} \right), \tag{182}$$

which is non-covariant. In fact, this is the diagram whose examination leads to the solution for the gauge choice-dependence problem as we will see below. The $V_{g,III}$ vertex also contributes to the diagram above:

$$\text{(diagram)} \Big|_{V_{g,III}} \quad \Rightarrow \quad \frac{\kappa'^2 \Gamma(\epsilon)}{(4\pi)^2} \int \left( \frac{3}{4} F_{\alpha\kappa} F_\beta{}^\kappa R^{\alpha\beta} + \frac{1}{8} F_{\alpha\beta} F^{\alpha\beta} R \right.$$

$$\left. + \frac{1}{4} F_{\alpha\delta} F_{\beta\gamma} R^{\alpha\beta\gamma\delta} - \frac{1}{4} F_{\alpha\beta} F_{\gamma\delta} R^{\alpha\beta\gamma\delta} \right). \tag{183}$$

Concerning the diagrams with the inhomogeneous loops, the first-layer diagram to be computed is the one in Figure 6. It corresponds to several second-layer diagrams, two of which are Figure 5a,b; one can show

$$\text{(diagram)} \quad \Rightarrow \quad \frac{\kappa'^2}{2} \frac{\Gamma(\epsilon)}{(4\pi)^2} \int \left( \frac{1}{3} \partial_\alpha F^\alpha{}_\kappa \partial_\beta F^{\beta\kappa} - \frac{1}{12} \partial_\rho F_{\alpha\beta} \partial^\rho F^{\alpha\beta} \right)$$

$$\text{(diagram)} \quad \Rightarrow \quad \kappa'^2 \frac{\Gamma(\epsilon)}{(4\pi)^2} \int \left( \frac{1}{3} F_{\alpha\kappa} \partial_\lambda \partial^\beta F_\beta{}^\kappa \varphi^{\alpha\lambda} - \frac{1}{12} F_{\alpha\kappa} \partial^2 F_\beta{}^\kappa \varphi^{\alpha\beta} \right), \tag{184}$$

where all of the index contractions are carried out with the flat metric. The first diagram is covariant at the leading order; the second diagram, however, is not at its given order, the $\varphi_{\alpha\beta}$-linear order. Moreover, there are also contributions arising from the higher-order internal propagators, and all of these three different contributions are required for the covariance since they altogether correspond to the single first-layer diagram in Figure 6. Keeping track of the higher-order internal propagators obviously requires the full (or at least higher-order) propagator expression $\tilde{\Delta}$. Because of these it will be more economical to compute them with one stroke by the first perturbation. For the first-layer perturbation calculation, the relevant vertex is

$$\mathbf{V} \equiv \frac{1}{2} \left[ \tilde{g}^{\rho\sigma} h^{\mu\nu} + \tilde{g}^{\mu\nu} h^{\rho\sigma} \right] f_{\mu\rho} F_{\nu\sigma}, \tag{185}$$

where the index contractions are carried out by $\tilde{g}_{\mu\nu}$. At this point let us introduce the orthonormal basis $e_{\underline{\alpha}}^\mu$:

$$e_{\underline{\alpha}}^\mu e_{\underline{\beta}}^\nu \tilde{g}_{\mu\nu} = \eta_{\underline{\alpha}\underline{\beta}}, \quad \underline{\alpha}, \underline{\beta} = 0, 1, 2, 3. \tag{186}$$

The full scalar propagator $\tilde{\Delta}$ can be written

$$\tilde{\Delta}(X_1 - X_2) = \int \frac{d^4L}{(2\pi)^4} \frac{e^{iL_{\underline{\delta}}(X_1 - X_2)^{\underline{\delta}}}}{iL_{\underline{\alpha}} L_{\underline{\beta}} \eta^{\underline{\alpha}\underline{\beta}}}, \tag{187}$$

where $X^{\underline{\alpha}}$ and $L_{\underline{\delta}}$ are the coordinates and momenta associated with the orthonormal basis. Then the computation of the two-point amplitude goes identically with that of Figure 5a. Once the result is obtained, one can switch back to the original frame. With this one gets

$$\Rightarrow -\frac{1}{2} < \left( \int \mathbf{v} \right)^2 > = \frac{\kappa'^2}{2} \frac{\Gamma(\epsilon)}{(4\pi)^2} \int \left( \frac{1}{3} \nabla_\alpha F^\alpha{}_\kappa \nabla_\beta F^{\beta\kappa} - \frac{1}{12} \nabla_\rho F_{\alpha\beta} \nabla^\rho F^{\alpha\beta} \right). \tag{188}$$

5.3.2. On Restoring Gauge-Choice Independence

Above, the counterterms for the diagrams in Figures 3–6 have been evaluated, leading to different types of the counterterms. One of them, i.e., Equation (182) (Figure 4c), has turned out non-covariant. This means that the effective action, as it stands, is non-covariant and gauge fixing-dependent. These two problems can be solved simply by enforcing the gauge-fixing

$$\partial_\mu \varphi^{\mu\nu} = 0 \tag{189}$$

explicitly on the effective action. One can see this by examining the non-covariant counter-terms given in (182). (Although this is just one example, we believe that the claim that the strong form of the gauge-fixing solves the problems will be generally true.) Note that the first term in (182) vanishes upon imposing the strong form of the gauge condition $\partial_\mu \varphi^{\mu\nu} = 0$, which implies

$$\partial_\nu \partial_\mu \varphi^{\mu\nu} = 0. \tag{190}$$

With this, Equation (182) now becomes

$$\left. \Rightarrow \right|_{V_{g,I} + V_{g,II}} \frac{\kappa'^2 \Gamma(\epsilon)}{(4\pi)^2} \int \left( \frac{1}{2} F_{\mu\kappa} F_\nu{}^\kappa \partial^2 \varphi^{\mu\nu} \right) = -\frac{\kappa'^2 \Gamma(\epsilon)}{(4\pi)^2} \int F_{\mu\kappa} F_\nu{}^\kappa R^{\mu\nu}, \tag{191}$$

where the second equality is valid, as usual, up to (and including) the linear order of $\varphi_{\alpha\beta}$. Above the following identity valid at $\varphi_{\rho\sigma}$-linear order has been used:

$$R_{\mu\nu} = \frac{1}{2} (\partial^\kappa \partial_\mu \varphi_{\kappa\nu} + \partial^\kappa \partial_\nu \varphi_{\kappa\mu} - \partial_\mu \partial_\nu \varphi - \partial^2 \varphi_{\mu\nu}) = -\frac{1}{2} \partial^2 \varphi_{\mu\nu}, \tag{192}$$

where the second equality results once the gauge conditions are enforced.

Among the terms explicitly evaluated above, only (182) has an issue; all the other results are gauge-fixing-independent. The analysis above suggests that after enforcing $\partial_\nu \varphi^{\mu\nu} = 0$, the effective action becomes fully covariant and gauge-choice-independent. In this sense, the gauge-choice-dependence found in the present work is milder compared to the gauge-choice-dependence found in the previous literature, and we attribute this to the use of the traceless propagator and refined background field method. Just as the classical action is fully covariant but is to be supplemented by a gauge-fixing condition, one should view the covariant 1PI action as still to be supplemented by the gauge-fixing. If one chooses a different gauge-fixing and carries out the amplitude computations in that gauge, one should get exactly the same covariant effective action up to the terms that can be removed by explicitly enforcing that gauge condition. In other words, this time, the covariant action is supplemented with the very gauge-fixing condition that one has chosen.

Therefore, the gauge-choice-independence of the effective action should be interpreted to mean that the action is covariant after enforcing the strong form of the gauge condition and that the covariant action is to be supplemented by the gauge-fixing condition of one's choice. The gauge-choice-dependence seems to have deep roots, having something to do with how the BFM itself works. More detailed discussion on this can be found in [82].

*5.4. Renormalization Procedure*

In this subsection we carry out two tasks: renormalization of the coupling constants and explicit one-loop renormalization via a metric field redefinition. We first analyze the renormalization of the three coupling constants: the cosmological constant, Newton's constant, and the vector coupling constant. The vacuum-to-vacuum and tadpole diagrams given in Figure 7 are responsible for the renormalization of the first two. (Unlike in a non-gravitational theory, the tadpole diagrams play an important role.) As for the vector coupling, the diagram in Figure 7 should be considered. Afterwards we carry out the renormalization via a metric field redefinition.

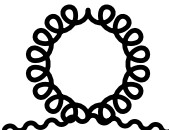

**Figure 7.** Diagram for vector coupling renormalization.

5.4.1. Renormalization of Coupling Constants

In terms of the first-layer perturbation, the loop corrections of the cosmological and Newton's constants are brought by the vacuum-to-vacuum and tadpole diagrams, respectively. See Figure 8 for a list of the diagrams for the pure gravity sector; there are similar diagrams for the matter-involving sector. For the graviton vacuum-to-vacuum amplitude, for example, one is to compute

$$\int \prod_x dh_{\kappa_1 \kappa_2} \, e^{\frac{i}{\kappa'^2} \int \sqrt{-\tilde{g}} \left( -\frac{1}{2} \tilde{\nabla}_\gamma h^{\alpha\beta} \tilde{\nabla}^\gamma h_{\alpha\beta} \right)}. \tag{193}$$

This vacuum energy amplitude in the first-layer perturbation will give a vacuum diagram and a tadpole diagram in the second-layer perturbation analysis. These diagrams as well as the genuine tadpole diagrams are analyzed below.

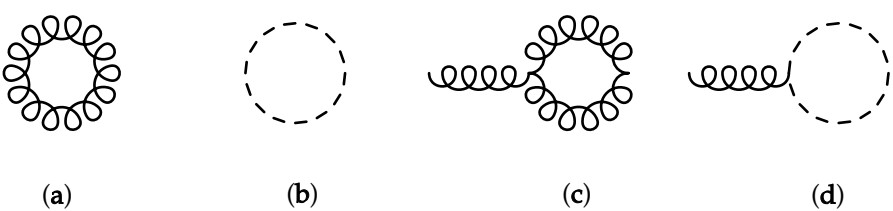

**Figure 8.** Vacuum and tadpol diagrams.

Let us first frame the analysis of the vacuum and tadpole diagrams in preparation for the flat space calculation. The vacuum-to-vacuum amplitude Figure 8a takes the form of the cosmological constant term and diverges (see, e.g., in [51,83]).[20] However, in dimensional regularization the vacuum energy diagram vanishes—which is an undesirable feature of dimensional regularization when dealing

---

[20] The discussion here is for a flat spacetime, but the divergence will be quite generically produced for an arbitrary background.

with a massless theory[21]—due to an identity Equation (199) below. (If we were dealing with a massive theory instead, a counterterm of the form of the cosmological constant of an infinite value would be required to remove the divergence.) The diagrams responsible for the renormalization of the Newton's constant are the tadpole diagrams. As we will see in detail below, the (would-be) shift in the Newton's constant is caused by a diagram that arises from self-contraction of two fluctuation fields within the given vertex. This time, the following identity makes the dimensional regularization unhandy for the tadpole diagrams:

$$\int d^D k \frac{1}{(k^2)^\omega} = 0, \tag{194}$$

where $\omega$ is an arbitrary number: the tadpole diagram vanishes due to this. In other words, the divergence that would otherwise renormalize the Newton's constant vanishes in dimensional regularization. For reasons to be explained, we will introduce the shifts in the cosmological and Newton's constants through finite renormalization.

The kinetic terms, which we quote here for convenience, are responsible for the first-layer vacuum-to-vacuum amplitudes:

$$2\kappa^2 \mathcal{L} = \sqrt{-\tilde{g}} \left( -\frac{1}{2} \tilde{\nabla}_\gamma h^{\alpha\beta} \tilde{\nabla}^\gamma h_{\alpha\beta} + \frac{1}{4} \tilde{\nabla}_\gamma h_\alpha^\alpha \tilde{\nabla}^\gamma h_\beta^\beta \right)$$
$$= -\frac{1}{2} \partial_\gamma h^{\alpha\beta} \partial^\gamma h_{\alpha\beta} + \frac{1}{4} \partial_\gamma h_\alpha^\alpha \partial^\gamma h_\beta^\beta + V_{g,I} + V_{g,II}, \tag{195}$$

where

$$V_{g,I} \equiv \left( 2\eta^{\beta\beta'} \tilde{\Gamma}^{\alpha'\gamma\alpha} - \eta^{\alpha\beta} \tilde{\Gamma}^{\alpha'\gamma\beta'} \right) \partial_\gamma h_{\alpha\beta} \, h_{\alpha'\beta'}$$

$$V_{g,II} \equiv \Big[ \frac{1}{2} (\eta^{\alpha\alpha'} \eta^{\beta\beta'} \varphi^{\gamma\gamma'} + \eta^{\beta\beta'} \eta^{\gamma\gamma'} \varphi^{\alpha\alpha'} + \eta^{\alpha\alpha'} \eta^{\gamma\gamma'} \varphi^{\beta\beta'})$$
$$- \frac{1}{4} \varphi \, \eta^{\alpha\alpha'} \eta^{\beta\beta'} \eta^{\gamma\gamma'} - \frac{1}{2} \eta^{\gamma\gamma'} \eta^{\alpha'\beta'} \varphi^{\alpha\beta}$$
$$+ \frac{1}{4} (-\varphi^{\gamma\gamma'} + \frac{1}{2} \varphi \eta^{\gamma\gamma'}) \eta^{\alpha\beta} \eta^{\alpha'\beta'} \Big] \partial_\gamma h_{\alpha\beta} \, \partial_{\gamma'} h_{\alpha'\beta'} \tag{196}$$

$$V_{g,III} = \sqrt{-\tilde{g}} \left( h_{\alpha\beta} h_{\gamma\delta} \tilde{R}^{\alpha\gamma\beta\delta} - h_{\alpha\beta} h^\beta_{\ \gamma} \tilde{R}^{\kappa\alpha\gamma}_{\ \ \ \kappa} - \frac{1}{2} h^{\alpha\beta} h_{\alpha\beta} \tilde{R} \right). \tag{197}$$

The first-layer vacuum-to-vacuum amplitudes are split into two parts in the second-layer perturbation; the vacuum-to-vacuum amplitudes and the tadpoles. Let us consider the second-layer vacuum-to-vacuum amplitudes. The vacuum energy leads to the cosmological constant renormalization and comes from

$$\int \prod_x dh_{\kappa_1 \kappa_2} \, e^{\frac{i}{\kappa'^2} \int \left( -\frac{1}{2} \partial_\gamma h^{\alpha\beta} \partial^\gamma h_{\alpha\beta} \right)} \tag{198}$$

The result is a constant term in the 1PI action (see, e.g., the analysis given in [83]): the calculation above leads to a quantum-level cosmological constant. The divergent part, which will be denoted by $A_0$ below, of the constant term is essentially the coefficient of the cosmological constant term. Here is the difference between gravity and a non-gravitational theory. In a non-gravitational theory, appearance of a term absent in the classical action would potentially signal non-renormalizability. In a gravitational theory one has an additional leverage of a metric field redefinition (more on this

---

[21]  For instance, the identities in (194) and (199) often obscure cancellations between the bosonic and fermionic amplitudes in a supersymmetric field theory, making them vanish separately.

later). (Even in a non-gravitational theory, appearance of a finite number of new couplings is taken to be compatible with renormalizability.)

The one-loop vacuum-to-vacuum amplitude, whether it is from the graviton or the ghost (or matter), involves the following integral that vanishes in dimensional regularization:

$$\int d^4 p \ln p^2 = 0. \tag{199}$$

Nevertheless, we introduce renormalization of the coupling constants through finite renormalization for the following reasons. Although the expression above is taken to vanish in dimensional regularization, the vacuum energy expression, in particular $A_0$ in (201), will not, in general, vanish in other regularizations for a curved background. To examine the behavior of the integral let us add a mass term $m^2$ that will be taken to $m^2 \to 0$ at the end,

$$\sim \int d^4 p \ln (p^2 + m^2). \tag{200}$$

For its evaluation, one can then take derivatives with respect to $m^2$; the result takes the form of

$$A_f + A_0 + A_1 m^2 + A_2 m^4, \tag{201}$$

where $A$'s are some $m$-independent constants; the finite piece, $A_f$, takes

$$A_f \sim m^4 \ln m^2. \tag{202}$$

With the limit $m^2 \to 0$, only the term with the constant $A_0$, which is infinite, survives, and dimensional regularization amounts to setting $A_0 = 0$. Although each term in (201) either vanishes or is taken to zero in dimensional regularization, not introducing nonvanishing finite pieces seems unnatural (and ultimately, unlikely to be consistent with the experiment); in a more general procedure of renormalization of a quantum field theory, one can always conduct finite renormalization regardless of the presence of the divergences. (As we will see below, not only the quantum shift but also a "classical" piece of the cosmological constant must be introduced.) Once a finite piece is introduced and the definition of the physical cosmological constant is made (say, as the coefficient of the $\int \sqrt{-\tilde{g}}$ term), the renormalized coupling will run basically due to the presence of the scale parameter $\mu$ (more details below).

Let us now consider the tadpole diagrams. For the second-layer tadpoles, the rest of the vertices in the kinetic term in (195)—which are nothing but $V_{g,I}$ and $V_{g,II}$—as well as $V_{g,III}$ are relevant; the former are part of the first-layer vacuum-to-vacuum amplitude whereas the latter is associated with a genuine first-layer tadpole. It turns out that $V_{g,I}$, $V_{g,II}$ lead to vanishing results in dimensional regularization. Let us illustrate that with $V_{g,I}$,

$$V_{g,I} = \left( 2\eta^{\beta\beta'} \tilde{\Gamma}^{\alpha'\gamma\alpha} - \eta^{\alpha\beta} \tilde{\Gamma}^{\alpha'\gamma\beta'} \right) \partial_\gamma h_{\alpha\beta} \, h_{\alpha'\beta'}. \tag{203}$$

The self-contraction of the $h_{\mu\nu}$'s in (203) leads to a momentum loop integral with an odd integrand, which thus vanishes. (The other terms in (195) vanish because the self-contraction leads to the trace of $\varphi_{\mu\nu}$.) The vertex $V_{g,III}$ similarly leads to a vanishing result. To see this, consider contraction of the $h_{\alpha\beta}$-fields in $V_{g,III}$. The index structures yield the Ricci scalar $R$ but the self-contraction vanishes in dimensional regularization due to the identity (194). Then, as in the case of the cosmological constant, the dimensional regularization does not lead to a divergence for the tadpole diagram; the shift is introduced through finite renormalization.

The diagram relevant for the vector coupling renormalization is given in Figure 7, a tadpole diagram with the graviton running on the loop. The relevant first-layer vertex is

$$-\frac{1}{4}\int\sqrt{-\tilde{g}}\left[\tilde{g}^{\mu\nu}h^{\rho\kappa}h^{\sigma}_{\kappa} + \tilde{g}^{\rho\sigma}h^{\mu\kappa}h^{\nu}_{\kappa} - \frac{1}{2}\tilde{g}^{\mu\nu}hh^{\rho\sigma} - \frac{1}{2}\tilde{g}^{\rho\sigma}hh^{\mu\nu} + h^{\mu\nu}h^{\rho\sigma} + \frac{1}{8}\tilde{g}^{\mu\nu}\tilde{g}^{\rho\sigma}(h^2 - 2h_{\kappa_1\kappa_2}h^{\kappa_1\kappa_2})\right]\tilde{F}_{\mu\rho}\tilde{F}_{\nu\sigma}. \quad (204)$$

The correlator to be computed is

$$-\frac{i}{4\mu^{2\epsilon}e^2}\int\sqrt{-\tilde{g}}\tilde{F}_{\mu\rho}\tilde{F}_{\nu\sigma}\left\langle\tilde{g}^{\mu\nu}h^{\rho\kappa}h^{\sigma}_{\kappa} + \tilde{g}^{\rho\sigma}h^{\mu\kappa}h^{\nu}_{\kappa} - \frac{1}{2}\tilde{g}^{\mu\nu}hh^{\rho\sigma} - \frac{1}{2}\tilde{g}^{\rho\sigma}hh^{\mu\nu} + h^{\mu\nu}h^{\rho\sigma} + \frac{1}{8}\tilde{g}^{\mu\nu}\tilde{g}^{\rho\sigma}(h^2 - 2h_{\kappa_1\kappa_2}h^{\kappa_1\kappa_2})\right\rangle, \quad (205)$$

where the self-contractions of the fluctuation fields are to be performed. The correlator leads to a counterterm of the form $\sim \tilde{F}^2_{\alpha\beta}$. Again the result vanishes due to the identity (194) in dimensional regularization. The shift can be introduced through finite renormalization. In Section 5.5 below, we revisit the renormalization of the coupling constants by employing an alternate renormalization scheme where the cosmological constant is treated as a formal graviton mass.

### 5.4.2. Renormalization through Metric Field Redefinition

The one-loop renormalization procedure is in order: we are ready to show that the Einstein–Hilbert action with the counter-terms can be rewritten as the same form of the Einstein–Hilbert action but now in terms of a redefined metric. Afterwards we contemplate on several possible alternative procedures of renormalization. We also comment on the higher-loop extension of the present work. The analysis here is to illustrate the renormalization procedure and is based on the computation that we have carried out in the sections so far. Some of the diagrams that we did not explicitly calculate will change the numerical values of some of the coefficients.

Collecting the results, the renormalized action plus the counterterms is given by

$$\int\sqrt{-g}\,(e_1 + e_2 R + e_3 R^2 + e_4 R^2_{\alpha\beta})$$
$$+ \int\sqrt{-g}\left(e_5 F_{\mu\kappa}F_\nu{}^\kappa R^{\mu\nu} + e_6 F_{\alpha\beta}F^{\alpha\beta}R + e_7 F_{\alpha\delta}F_{\beta\gamma}R^{\alpha\beta\gamma\delta}\right. \qquad (206)$$
$$\left. + e_8 F_{\alpha\beta}F_{\gamma\delta}R^{\alpha\beta\gamma\delta} + e_9\nabla^\alpha F_{\alpha\kappa}\nabla^\beta F_\beta{}^\kappa + e_{10}\nabla_\lambda F_{\mu\nu}\nabla^\lambda F^{\mu\nu} + e_{11}(F_{\alpha\beta}F^{\alpha\beta})^2 + \cdots\right),$$

where $e_1$ is the constant previously denoted by $A_0$. More precisely, $[e_1] = A_0$, where the square bracket $[e_i]$ denotes the infinite parts of the coefficient $e_i$ calculated in dimensional regularization. Similarly, the would-be divergence of the tadpole diagrams will be denoted $B_0 = [e_2]$. ($A_0, B_0$ are taken to vanish in dimensional regularization.) For the rest of the coefficients, one has, by collecting the results in the previous sections,

$$[e_3] = -\frac{17}{60} + \frac{23}{80} + \frac{1}{30}, \quad [e_4] = -\frac{7}{30} + \frac{23}{40} - \frac{1}{10} - \frac{1}{15},$$
$$[e_5] = \left(-1 + \frac{3}{4}\right)\kappa'^2, \quad [e_6] = \frac{\kappa'^2}{8}, \quad [e_7] = \frac{\kappa'^2}{4}, \quad [e_8] = -\frac{\kappa'^2}{4},$$
$$[e_9] = \frac{\kappa'^2}{6}, \quad [e_{10}] = -\frac{\kappa'^2}{24}, \quad [e_{11}] = \frac{3}{64}\kappa'^4, \qquad , \qquad (207)$$

where the common factor $\frac{\Gamma(\epsilon)}{(4\pi)^2}$ appearing in dimensional regularization has been suppressed. The finite pieces of each coefficient can be determined, say, by the $\overline{\text{MS}}$ scheme. Not all of the counter-terms are independent, due to the following relationships, the second of which is valid up to total derivative terms:

$$F_{\alpha\beta}F_{\gamma\delta}R^{\alpha\beta\gamma\delta} = \nabla_\mu F_{\nu\rho}\nabla^\mu F^{\nu\rho} + 2F_{\mu\kappa}F_\nu{}^\kappa R^{\mu\nu} - 2\nabla^\lambda F_{\lambda\kappa}\nabla^\sigma F_\sigma{}^\kappa$$
$$F_{\alpha\delta}F_{\beta\gamma}R^{\alpha\beta\gamma\delta} = -\frac{1}{2}F_{\alpha\beta}F_{\gamma\delta}R^{\alpha\beta\gamma\delta}. \qquad (208)$$

Upon substituting these into (206), one gets

$$
\int \sqrt{-g}\,\Big[ e_1 + e_2 R + e_3 R^2 + e_4 R_{\alpha\beta}^2 + (e_5 - e_7 + 2e_8) F_{\mu\kappa} F_\nu{}^\kappa R^{\mu\nu}
$$
$$
+ e_6 F_{\alpha\beta} F^{\alpha\beta} R + (e_7 - 2e_8 + e_9) \nabla^\alpha F_{\alpha\kappa} \nabla^\beta F_\beta{}^\kappa \tag{209}
$$
$$
+ (-e_7/2 + e_8 + e_{10}) \nabla_\lambda F_{\mu\nu} \nabla^\lambda F^{\mu\nu} + e_{11} (F_{\alpha\beta} F^{\alpha\beta})^2 + \cdots \Big].
$$

The strategy is to absorb these counterterms by redefining the metric in the bare action. Upon inspection one realizes that the counterterms of the forms $\nabla_\lambda F_{\mu\nu} \nabla^\lambda F^{\mu\nu}$, $\nabla^\alpha F_{\alpha\kappa} \nabla^\beta F_\beta{}^\kappa$ cannot be absorbed by a bare action that consists of the Einstein–Hilbert term and the Maxwell term. However, they can be absorbed by introducing the cosmological constant term as well. To see this in detail, consider a metric shift $g_{\mu\nu} \to g_{\mu\nu} + \delta g_{\mu\nu}$; the Einstein–Hilbert part shifts

$$
\sqrt{-g}\, R \to \sqrt{-g}\, R + R\, \delta g^{\mu\nu} \frac{\delta \sqrt{-g}}{\delta g_{\mu\nu}} + \sqrt{-g}\, \delta g^{\mu\nu} \frac{\delta R}{\delta g_{\mu\nu}} \tag{210}
$$

so the shifted part comes either with $R$ or $R_{\mu\nu}$, and is thus inadequate to absorb the aforementioned counterterms; as is the shifted part from the Maxwell's action. From now on we assume the presence of the cosmological constant in the bare action. Let us consider the following shifts [38,51],

$$
\kappa \to \kappa + \delta\kappa \quad , \quad \Lambda \to \Lambda + \delta\Lambda
$$

$$
\begin{aligned}
g_{\mu\nu} \quad \to \quad \mathscr{G}_{\mu\nu} \equiv\ & l_0 g_{\mu\nu} + l_1 g_{\mu\nu} R + l_2 R_{\mu\nu} + l_3 g_{\mu\nu} F_{\rho\sigma}^2 + l_4 F_{\mu\kappa} F_\nu{}^\kappa \\
& + l_5 R F_{\mu\kappa} F_\nu{}^\kappa + l_6 R_{\mu\nu} F_{\kappa_1\kappa_2}^2 + l_7 g_{\mu\nu} R F_{\rho\sigma}^2 + l_8 g_{\mu\nu} R^{\alpha\beta} F_{\alpha\kappa} F_\beta{}^\kappa \\
& + l_9 R_\mu{}^\alpha{}_\nu{}^\beta F_{\alpha\kappa} F_\beta{}^\kappa + l_{10} R (F_{\kappa_1\kappa_2} F^{\kappa_1\kappa_2})^2 \\
& + l_{11} \nabla_\mu F_{\kappa_1\kappa_2} \nabla_\nu F^{\kappa_1\kappa_2} + l_{12} \nabla^\lambda F_{\lambda\mu} \nabla^\kappa F_{\kappa\nu}.
\end{aligned} \tag{211}
$$

One can straightforwardly show that under these, the gravity and matter sectors shift, respectively,

$$
\begin{aligned}
& -\Big(\frac{2}{\kappa^2}\Lambda\Big) \int \sqrt{-g} + \frac{1}{\kappa^2} \int d^4x \sqrt{-g}\, R \to -2\Big(\frac{\Lambda}{\kappa^2} + \frac{\delta\Lambda}{\kappa^2} - \frac{2\delta\kappa\Lambda}{\kappa^3} + 2l_0\Lambda\Big) \int \sqrt{-g} \\
& + \Big(\frac{1}{\kappa^2} - \frac{2\delta\kappa}{\kappa^3} + \frac{l_0}{\kappa^2} - \frac{\Lambda}{\kappa^2}(4l_1 + l_2)\Big) \int \sqrt{-g}\, R + \frac{1}{\kappa^2} \int \sqrt{-g}\Big[(l_1 + \tfrac{1}{2}l_2)R^2 - l_2 R_{\mu\nu} R^{\mu\nu}\Big] \\
& + \frac{1}{\kappa^2} \int \sqrt{-g}\Big[ -\Lambda(4l_3 + l_4) F_{\alpha\beta} F^{\alpha\beta} + \Big(l_3 + l_4/2 - \Lambda[l_5 + l_6 + 4l_7]\Big) R F_{\alpha\beta} F^{\alpha\beta} \\
& -\Lambda(4l_8 + l_9) R^{\alpha\beta} F_{\alpha\kappa} F_\beta{}^\kappa - 4\Lambda l_{10}(F_{\rho\sigma} F^{\rho\sigma})^2 - \Lambda l_{11}(\nabla_\mu F_{\nu\rho})^2 - \Lambda l_{12}(\nabla^\kappa F_{\kappa\nu})^2 + \ldots \Big],
\end{aligned} \tag{212}
$$

and

$$
\begin{aligned}
-\frac{1}{4} \int \sqrt{-g}\, F_{\mu\nu}^2 \to\ & -\frac{1}{4} \int \sqrt{-g}\, F_{\mu\nu}^2 + \int \sqrt{-g}\Big[ -\frac{l_2}{8} R F_{\alpha\beta} F^{\alpha\beta} \\
& + \frac{l_2}{2} R^{\alpha\beta} F_{\alpha\kappa} F_\beta{}^\kappa + \frac{l_3}{2}(F_{\rho\sigma} F^{\rho\sigma})^2 + \frac{l_4}{2} F_{\alpha\kappa} F_\beta{}^\kappa F^{\alpha\kappa'} F^\beta{}_{\kappa'} + \cdots \Big].
\end{aligned} \tag{213}
$$

Combining these two, one gets

$$
\begin{aligned}
\frac{1}{\kappa^2} \int d^4x \sqrt{-g}\,(R - 2\Lambda) - \frac{1}{4} \int \sqrt{-g}\, F_{\mu\nu}^2 &\to -2\Big(\frac{\Lambda}{\kappa^2} + \frac{\delta\Lambda}{\kappa^2} - \frac{2\delta\kappa\Lambda}{\kappa^3} + 2l_0\Lambda\Big) \int \sqrt{-g} \\
&+ \Big(\frac{1}{\kappa^2} - \frac{2\delta\kappa}{\kappa^3} + \frac{1}{\kappa^2}l_0 - \frac{1}{\kappa^2}\Lambda(4l_1 + l_2)\Big) \int \sqrt{-g}\, R - \frac{1}{4} \int \sqrt{-g}\, F_{\mu\nu}^2 \\
&+ \frac{1}{\kappa^2} \int \sqrt{-g}\Big[(l_1 + \tfrac{1}{2}l_2)R^2 - l_2 R_{\mu\nu}R^{\mu\nu}\Big] + \frac{1}{\kappa^2} \int \sqrt{-g}\Big[ -\Lambda(4l_3 + l_4)F_{\alpha\beta}F^{\alpha\beta} \\
&+ \Big(l_3 + \frac{l_4}{2} - \Lambda[l_5 + l_6 + 4l_7] - \frac{l_2}{8}\kappa^2\Big)RF_{\alpha\beta}F^{\alpha\beta} + \Big(\kappa^2\frac{l_2}{2} - \Lambda[4l_8 + l_9]\Big)R^{\alpha\beta}F_{\alpha\kappa}F_{\beta}{}^{\kappa} \\
&(\kappa^2 l_3/2 - 4\Lambda l_{10})(F_{\rho\sigma}F^{\rho\sigma})^2 - l_{11}(\nabla_\mu F_{\nu\rho})^2 - l_{12}(\nabla^\kappa F_{\kappa\nu})^2 + \dots \Big]
\end{aligned}
\tag{214}
$$

Not all of the terms in the expansion have been explicitly recorded; some of them would be relevant for additional diagrams such as three-pt amplitudes.

Let us consider the first several coefficients of the shifted action and compare them with those appearing in (209). We start with the cosmological constant term and the Einstein–Hilbert term. Their counterterms can be absorbed by setting

$$
-\frac{2}{\kappa^2}\Big(\delta\Lambda - \frac{2\delta\kappa\Lambda}{\kappa}\Big) = A_0
\tag{215}
$$

and

$$
-\frac{2}{\kappa^3}\delta\kappa + \frac{1}{\kappa^2}l_0 - \frac{\Lambda}{\kappa^2}(4l_1 + l_2) = B_0,
\tag{216}
$$

respectively. Recall that the constants $A_0, B_0$ now contain the non-vanishing finite pieces introduced by the aforementioned finite renormalization. Equation (216) determines the infinite part of $\delta\kappa$

$$
\delta\kappa = \frac{\kappa}{2}l_0 - \frac{\kappa\Lambda}{2}(4l_1 + l_2) - \frac{\kappa^3}{2}B_0.
\tag{217}
$$

The value $\delta\Lambda$ is determined once this result is substituted into (215):

$$
\delta\Lambda = l_0\Lambda - \Lambda^2(4l_1 + l_2) - \frac{\kappa^2}{2}A_0.
\tag{218}
$$

The counterterms of the forms $R^2, R_{\mu\nu}^2$ can be absorbed by setting

$$
l_1 + \frac{1}{2}l_2 = e_3 \quad , \quad -l_2 = e_4,
\tag{219}
$$

which yields

$$
l_1 = e_3 + \frac{1}{2}e_4 \quad , \quad l_2 = -e_4.
\tag{220}
$$

Inspection of the coefficients of $F_{\alpha\beta}^2$ implies

$$
4l_3 + l_4 = \mathcal{O}(\kappa^4).
\tag{221}
$$

The coefficients of $RF_{\alpha\beta}F^{\alpha\beta}$, $R^{\alpha\beta}F_{\alpha\kappa}F_{\beta}{}^{\kappa}$ should match with the corresponding coefficients in (209):

$$
l_3 + \frac{l_4}{2} - \Lambda(l_5 + l_6 + 4l_7) - \frac{l_2}{8}\kappa^2 = e_6 \, , \quad \frac{\kappa^2}{2}l_2 - \Lambda(4l_8 + l_9) = e_5 - e_7 + 2e_8.
$$

These constraints are to be combined with those coming from the higher-order counter-terms.

As for the two- [84] and higher-loop-order renormalization, the following can be said: at two-loop, no such identity as (1) is available and one must therefore rely on the holography-inspired reduction. The two- and higher-loop renormalizability in an asymptotically flat background can be achieved by following the outlines presented in [36].

*5.5. Beta Function Analysis*

In the previous subsection we have initially carried out the analysis without including the cosmological constant. As we have reviewed, dimensional regularization has a technical subtlety: the flat propagator yields vanishing results for the vacuum and tadpole diagrams. Because of this, the shifts in the coupling constants were viewed as having been introduced through finite renormalization. Through the analysis in Section 5.4, it has been revealed that the cosmological constant is generically generated by the loop effects and the renormalizability requires its presence in the bare action. This status of the matter seems to suggest the possibility of carrying out an alternative renormalization procedure by including the cosmological constant in the starting renormalized action. Once the cosmological constant is included and expanded around the fluctuation metric, it can be treated as a source for additional vertices. As for the second-order fluctuation term, it can be treated as the "graviton mass" term. With this arrangement, the vacuum-to-vacuum and tadpole diagrams yield non-vanishing results. Here we carry out the beta function analysis of the vector coupling constant to illustrate this alternative procedure.

An analysis of renormalization of the vector coupling was previously carried out in [29,30] by employing the setup of [27]. An interesting role of the cosmological constant was noted: its presence led to running of the matter coupling constant that was absent when the cosmological constant was not present [85]. It was observed that the presence of the cosmological constant generates the formal mass terms for the photon and graviton. Our beta function calculation below confirms the result obtained in [85].

In the present context, treating a cosmological constant-type term as the graviton mass term was considered in [51] in an Einstein-scalar theory with a Higgs-type potential; the scalar part of the Lagrangian is

$$S = -\int d^4x \sqrt{-g} \left( \frac{1}{2} g^{\mu\nu} \partial_\mu \zeta \partial_\nu \zeta + V \right).$$
(222)

with the potential $V$ given by

$$V = \frac{\lambda}{4} \left( \zeta^2 + \frac{1}{\lambda} v^2 \right)^2,$$
(223)

where $\lambda$ is the scalar coupling and $v^2$ is the mass parameter. In [51] we treated the constant term from the potential as the graviton mass term:

$$m^2 = \frac{\kappa'^2}{8} \frac{v^4}{\lambda}.$$
(224)

In the present case we similarly treat the quadratic part of the cosmological constant term as a formal mass term for the graviton:

$$m^2 = -2\Lambda.$$
(225)

For the detailed analysis of the beta function, it is convenient, and common, to introduce a scale parameter $\mu$ by making the following scalings :

$$\kappa^2 \to \mu^{2\varepsilon} \kappa^2 \quad , \quad e^2 \to \mu^{2\varepsilon} e^2 \quad , \quad \Lambda \to \mu^{-2\varepsilon} \Lambda$$
(226)

With this, Equation (122) takes

$$S = \int \sqrt{-\hat{g}} \left( \frac{1}{\kappa^2 \mu^{2\varepsilon}} (\hat{R} - 2\Lambda \mu^{-2\varepsilon}) - \frac{1}{4e^2 \mu^{2\varepsilon}} \hat{F}_{\mu\nu}^2 \right). \tag{227}$$

Now the kinetic part of the gravity sector is

$$\mathcal{L}_{kin} = \frac{1}{2\kappa^2 \mu^{2\varepsilon}} \sqrt{-\tilde{g}} \left[ -\frac{1}{2} \tilde{\nabla}_\gamma h^{\alpha\beta} \tilde{\nabla}^\gamma h_{\alpha\beta} - \frac{1}{2} (-2\Lambda) h_{\mu\nu} h^{\mu\nu} \right]. \tag{228}$$

Treating the cosmological constant-containing term as the "mass" term has the effect of changing (187) to

$$\tilde{\Delta}(X_1 - X_2) = \int \frac{d^4 L}{(2\pi)^4} \frac{e^{iL_{\underline{\delta}}(X_1 - X_2)^{\underline{\delta}}}}{i(L_{\underline{\alpha}} L_{\underline{\beta}} \eta^{\underline{\alpha\beta}} - 2\Lambda)}. \tag{229}$$

The correlator relevant for the vector coupling renormalization is

$$-\frac{i}{4\mu^{2\varepsilon}e^2} \int \sqrt{-\tilde{g}} \tilde{F}_{\mu\rho} \tilde{F}_{\nu\sigma} \left\langle \tilde{g}^{\mu\nu} h^{\rho\kappa} h_\kappa^\sigma + \tilde{g}^{\rho\sigma} h^{\mu\kappa} h_\kappa^\nu - \frac{1}{2} \tilde{g}^{\mu\nu} h h^{\rho\sigma} - \frac{1}{2} \tilde{g}^{\rho\sigma} h h^{\mu\nu} + h^{\mu\nu} h^{\rho\sigma} + \frac{1}{8} \tilde{g}^{\mu\nu} \tilde{g}^{\rho\sigma} (h^2 - 2h_{\kappa_1\kappa_2} h^{\kappa_1\kappa_2}) \right\rangle \tag{230}$$

Carrying out the self-contractions of the fluctuation fields, one gets

$$
\begin{aligned}
&= -\frac{3i\mu^{2\varepsilon}\kappa'^2}{8\mu^{2\varepsilon}e^2} \int \sqrt{-\tilde{g}} \, \tilde{F}_{\mu\rho} \tilde{F}^{\mu\rho} \int \frac{d^4 L}{(2\pi)^4} \frac{1}{i(L^2 - 2\Lambda\mu^{-2\varepsilon})} \\
&= -\frac{3\mu^{2\varepsilon}\kappa'^2}{8\mu^{2\varepsilon}e^2} \frac{\Gamma(\varepsilon)(2\Lambda\mu^{-2\varepsilon})}{(4\pi)^2} \int \sqrt{-\tilde{g}} \, \tilde{F}_{\mu\rho} \tilde{F}^{\mu\rho} \\
&\simeq \frac{3}{32\pi^2(D-4)} \frac{\kappa^2\Lambda}{\mu^{2\varepsilon}e^2} \int \sqrt{-\tilde{g}} \, \tilde{F}_{\mu\rho} \tilde{F}^{\mu\rho},
\end{aligned}
\tag{231}
$$

where the second equality has been obtained by performing the momentum integration after a Wick rotation; the third equality is obtained by keeping only the pole term of $\Gamma(\varepsilon)$. The result above implies that the one-loop-corrected vector coupling $e_1$ is given by

$$e_1 = e\mu^\varepsilon \left( 1 + \frac{3}{8\pi^2(D-4)} \kappa^2 \Lambda \right)^{\frac{1}{2}} \simeq e\mu^\varepsilon \left( 1 + \frac{3}{16\pi^2(D-4)} \kappa^2 \Lambda \right). \tag{232}$$

From this it follows that

$$\mu \frac{\partial e_1}{\partial \mu} = \varepsilon e \mu^\varepsilon - \frac{3\mu^\varepsilon}{32\pi^2} \kappa^2 \Lambda e. \tag{233}$$

Taking $\varepsilon \to 0$, one gets the following beta function:

$$\beta(e) = -\frac{3}{32\pi^2} \kappa^2 \Lambda e. \tag{234}$$

This is the same as the result obtained in [85].[22] The result in [85] was obtained by employing the standard one-loop determinant formula.[23] Although, strictly speaking, the formula makes sense only when the traceless part of the fluctuation field is taken out, once the formula is used it doesn't matter how it is obtained. In other words, the result in [85] was obtained, so to speak, by bypassing the

---

[22] Note that $\kappa^2$ in [85] is twice $\kappa^2$ here.
[23] In a schematic notation, the formula reads

$$\int D\xi \, e^{-\frac{1}{2}\xi K \xi} = e^{-\frac{1}{2} \operatorname{tr} \ln \frac{K}{2\pi}}. \tag{235}$$

traceful propagator, and we believe that this is why our beta function result—obtained by employing the traceless propagator—agrees with that therein obtained.[24]

## 6. Future Astrophysical Applications

With the renormalization of gravity reasonably under control, we ponder astrophysical applications of the present quantization scheme. Although the quantum gravitational effects are viewed as small conventionally, one of the lessons learned through a series of the recent works is that this is essentially not true in general for the following two reasons. Firstly, there is an issue of the boundary conditions. Suppose one starts with a classical action with the standard Dirichlet boundary condition. At the quantum-level, the action comes to contain various higher-order terms, and for this the boundary conditions must be reexamined. As we will review below, the change in the boundary condition brings "order-1" effects to the classical picture. Secondly, there is a subtlety in taking the classical limit of the quantum-corrected quantities. Due to this there may again be "order-1" modifications to the classical picture. In [86,87] (see an earlier related work [88]), the energy measured by an infalling observer was studied for two different cases: quantum-corrected time-dependent dS-Schwarzschild and AdS black holes. It turns out that the infalling observer encounters a trans-Planckian energy near the horizon of the black hole, which is consistent with the Firewall proposal [89,90]. For this, the quantum effects and time-dependence are crucial. In particular, the so-called non-Dirichlet quantum modes play an important role in the trans-Planckian energy observed in [87]. Since these effects are non-perturbative and "big" they must be experimentally observable and have astrophysical significance. Below we discuss two potentially interesting astrophysical applications of the present quantization scheme: applications to AGN and gravitational wave physics.

As we will point out, the quantum gravitational effects substantially modify the geometry near the event horizon of a black hole. Because of this the horizon is no long featureless vacuum-like place but instead a potentially quite volatile place. Based on this we raise two questions and frame our future investigation. The first question pertains to the strength with which the quantum effects modify the near-horizon geometry. As we will show, the quantum effects do seem to cause an infalling observer to encounter a Firewall-type effect near the horizon when the black hole is time-dependent. Although no strictly Planck-scale physics has ever been directly observed experimentally, there is a phenomenon that seems fairly close—the high energy radiation by an AGN. Some of the extremely high-energy cosmic particles are believed to originate from AGNs. Their energy scales ($\sim 10^{19}$ ev) are not quite as high as the Planck energy. However, their energy losses on the way, e.g., the loss caused by climbing up the potential hill of the supermassive black hole, must be taken into account.

Another potentially interesting implication of the unconventional event horizon pertains to the boundary condition at the event horizon. In the conventional picture, one imposes the so-called perfect-infall boundary condition at the event horizon when studying the response of the black hole to an outside perturbation. Together with the entirely outgoing-wave boundary condition at the asymptotic region, the linearized metric field equation leads to the quasi-normal mode solutions. However, once the horizon is deformed by the quantum effects, other boundary conditions that would allow reflection at the event horizon are highly plausible possibilities. We frame future investigation by examining the Einstein-scalar system studied in [87].

The two issues above can be illustrated by taking the following AdS gravity-scalar system: at the classical level the system is given by

$$S = \frac{1}{\kappa^2} \int d^4x \sqrt{-g} \left[ R - 2\Lambda \right] - \int d^4x \sqrt{-g} \left[ \frac{1}{2}(\partial_\mu \zeta)^2 + \frac{1}{2} m^2 \zeta^2 \right] \tag{236}$$

---

[24]  Incidentally it also turns out that the result (230) remains the same even if one employs the traceful propagator, which should be a coincidence.

where $m^2 = -\frac{2}{L^2}$. It admits an AdS black hole solution,

$$\zeta = 0 \quad , \quad ds^2 = -\frac{1}{z^2}\left(Fdt^2 + 2dtdz\right) + W^2(dx^2 + dy^2) \tag{237}$$

with

$$F = -\frac{\Lambda}{3} - 2Mz^3 \quad , \quad W = \frac{1}{z} \quad , \quad \zeta = 0 \tag{238}$$

where $M$ is proportional to the mass of the black hole. At the classical level, the system admits the following form of a time-dependent solution [91]:

$$\begin{aligned} ds^2 &= -\frac{1}{z^2}\left[F(t,z)dt^2 + 2dt\,dz\right] + W(t,z)^2(dx^2 + dy^2) \\ \zeta &= \zeta(t,z) \end{aligned}$$

with

$$\begin{aligned} F(t,z) &= F_0(t) + F_1(t)z + F_2(t)z^2 + F_3(t)z^3 + \dots \\ W(t,z) &= \frac{1}{z} + W_0(t) + W_1(t)z + W_2(t)z^2 + W_3(t)z^3 + \dots \\ \zeta(t,z) &= \zeta_0(t) + \zeta_1(t)z + \zeta_2(t)z^2 + \zeta_3(t)z^3 + \dots . \end{aligned} \tag{239}$$

Substituting these ansatz into the field equations and imposing the Dirichlet boundary condition, one gets (we have set $L = 1$ and $\frac{6}{L^2} = -2\Lambda$):

$$\zeta_0 = W_0 = F_1 = 0, \quad F_0 = 1, \quad W_1 = -\frac{1}{8}\zeta_1^2, \quad F_2 = -\frac{1}{4}\zeta_1^2, \quad W_2 = -\frac{1}{6}\zeta_1\zeta_2$$

$$\zeta_3 = \frac{1}{2}\left(\frac{1}{2}\zeta_1^2\zeta_1 + 2\dot\zeta_2\right), \quad W_3 = \frac{1}{96}\left[-\frac{11}{4}\zeta_1{}^4 - 8\zeta_2^2 - 12\zeta_1\dot\zeta_2\right], \quad \dot F_3 = -\frac{1}{2}\zeta_1\dot\zeta_2 + \frac{1}{2}\zeta_1\ddot\zeta_2. \tag{240}$$

Unlike one's naive expectation, the quantum corrections modify the classical solution significantly, changing it to a qualitatively different black hole solution. The one-loop 1PI effective action is given by [51]

$$\begin{aligned} S = \frac{1}{\kappa^2}\int d^4x\sqrt{-g}\left[R - 2\Lambda\right] - \int d^4x\sqrt{-g}\left[\frac{1}{2}(\partial_\mu\zeta)^2 + \frac{1}{2}m^2\zeta^2\right] \\ + \frac{1}{\kappa^2}\int d^4x\sqrt{-g}\left[e_1\kappa^4 R\zeta^2 + e_2\kappa^2 R^2 + e_3\kappa^2 R_{\mu\nu}R^{\mu\nu} + e_4\kappa^6(\partial\zeta)^4 + e_5\kappa^6\zeta^4 + \cdots\right], \end{aligned} \tag{241}$$

where $e$'s are numerical constants that can be determined with fixed renormalization conditions. The quantum-level field equations can be obtained by varying the action (241). As a matter of fact, the boundary conditions must be considered before varying the action. As can be seen from the generalization of the GHY-term in Section 4.3, the quantum corrections in (241) will require various forms of the GHY-terms in case one wants to impose the Dirichlet boundary condition even at the quantum level. Recall that the quantum corrections above are generated after starting with a classical action with the Dirichlet boundary condition. In spite of the initial Dirichlet boundary condition, the quantum action comes to receive the higher-order terms that make the quantum-level field equation subject to the non-Dirichlet boundary condition if no additional GHY-terms are added. This seems to indicate that the quantum corrections are at odds with the Dirichlet boundary condition. It will

therefore be of some interest to examine the form of the solution that is not restricted by the Dirichlet boundary condition[25]—which is one of the two order-1 effects mentioned in the beginning.

The quantum system (241) admits the following form of the time-dependent solution:

$$ds^2 = -\frac{1}{z^2}\Big(F(t,z)dt^2 + 2dtdz\Big) + W^2(t,z)(dx^2 + dy^2), \tag{242}$$

with the quantum-corrected series

$$
\begin{aligned}
F(t,z) &= F_0(t) + F_1(t)z + F_2(t)z^2 + F_3(t)z^3 + \dots \\
&+ \kappa^2\Big[F_0^h(t) + F_1^h(t)z + F_2^h(t)z^2 + F_3^h(t)z^3 + \dots\Big] \\
W(t,z) &= \frac{1}{z} + W_0(t) + W_1(t)z + W_2(t)z^2 + W_3(t)z^3 + \dots \\
&+ \kappa^2\Big[\frac{W_{-1}^h(t)}{z} + W_0^h(t) + W_1^h(t)z + W_2^h(t)z^2 + W_3^h(t)z^3 + \dots\Big].
\end{aligned}
\tag{243}
$$

Similarly, for the scalar:

$$
\begin{aligned}
\zeta(t,z) &= \zeta_0(t) + \zeta_1(t)z + \zeta_2(t)z^2 + \zeta_3(t)z^3 + \dots \\
&+ \kappa^2\Big[\zeta_0^h(t) + \zeta_1^h(t)z + \zeta_2^h(t)z^2 + \zeta_3^h(t)z^3 + \dots\Big],
\end{aligned}
\tag{244}
$$

where the modes with superscript '*h*' represent the quantum modes. Upon substituting the ansatz into the field equations one gets, for the classical modes,

$$m^2 = \frac{2\Lambda_0}{3}, \quad \zeta_0 = 0, \quad F_0 = -\frac{\Lambda_0}{3}, \quad W_1 = 0, \quad F_1 = -F_0 W_0 - \Lambda_0 W_0$$

$$W_2 = 0, \quad F_2 = \frac{1}{4}\Big(4F_0 W_0{}^2 - 8\dot{W}_0\Big)$$

$$\zeta_3 = 0, \quad W_3 = 0, \quad F_3 = const, \quad \zeta_4 = 0, \quad W_4 = 0, \quad F_4 = -F_3 W_0; \tag{245}$$

and for the quantum modes,

$$\zeta_0^h = 0, \quad F_0^h = -\frac{1}{3}\kappa^2\Lambda_1, \quad W_1^h = 0, \quad F_1^h = \frac{2}{3}\Big(3F_0^h W_0 + \Lambda_0 W_0 W_{-1}^h - \Lambda_0 W_0^h - 3\dot{W}_{-1}^h\Big),$$

$$W_2^h = 0, \quad F_2^h = \frac{1}{3}\Big(-\kappa^2\Lambda_1 W_0{}^2 + 2\Lambda_0 W_0{}^2 W_{-1}^h - 2\Lambda_0 W_0 W_0^h + 6W_{-1}^h \dot{W}_0 - 6\dot{W}_0^h\Big),$$

$$\zeta_3^h = -\frac{1}{\Lambda_0}\Big(\Lambda_0 \zeta_1^h W_0{}^2 + 2\Lambda_0 \zeta_2^h W_0 + 3W_0 \dot{\zeta}_1^h + 3\zeta_1^h \dot{W}_0 + 3\ddot{\zeta}_2^h\Big), \quad W_3^h = 0, \quad \dot{F}_3^h = -3F_3 \dot{W}_{-1}^h$$

$$F_4^h = F_3 W_0 W_{-1}^h - F_3 W_0^h - F_3^h W_0, \quad W_4^h = -3e_2 F_3 W_0{}^2 + 3e_2 F_5 - 2e_3 F_3 W_0{}^2 + 2e_3 F_5$$

$$\zeta_4^h = \frac{F_3 \zeta_1^h}{2\Lambda_0} + \frac{12\dot{\zeta}_1^h \dot{W}_0}{\Lambda_0^2} + \frac{6W_0 \ddot{\zeta}_1^h}{\Lambda_0^2} + \frac{6\zeta_1^h \ddot{W}_0}{\Lambda_0^2} + \frac{6\ddot{\zeta}_2^h}{\Lambda_0^2} + \frac{9W_0{}^2 \dot{\zeta}_1^h}{\Lambda_0} + \frac{9W_0 \dot{\zeta}_2^h}{\Lambda_0} + \frac{9\zeta_1^h W_0 \dot{W}_0}{\Lambda_0}$$

$$+2\zeta_1^h W_0^3 + 3\zeta_2^h W_0^2. \tag{246}$$

The quantum corrections of the action imply a deformation of the geometry by quantum effects [74,86]. (See also [92,93] for related works.) One striking difference between the classical solution (240) and the quantum-level solution (245) and (246) is that although the building blocks for the classical solutions are the classical modes $(\zeta_1, \zeta_2)$, they are constrained to vanish once the

---

[25] The precise forms of the required boundary condition with the corresponding GHY-type terms will not be pursued in the present work. We will assume that such a boundary condition exists, and examine the implications of the series solution given in (243) and (244).

quantum-level field equations are considered; it is the quantum modes including $(\zeta_1^h, \zeta_2^h)$ that newly come to serve as the building blocks. This phenomenon is the other "order-1" effect and seems to have its origin in the subtlety in going to the classical limit [94]. In the present case, the subtlety is as follows. Reinserting the $\hbar$-dependence, one gets the following form for the $\hbar$-order field equation,

$$\hbar(\cdots) = 0 \tag{247}$$

as the $\hbar$-order parts of the field equations must vanish separately from the classical parts. Inside the parenthesis, some of the classical modes come to appear. If one takes the $\hbar \to 0$-limit too early, some of the quantum-level constraints will become omitted, which corresponds to the "usual" classical limit.

For the energy measured by an infalling observer when the observer is near the horizon, an expansion around the horizon should be useful. The metric field equation implies that the solution generically takes the form of

$$\zeta = \frac{\xi}{\kappa} \tag{248}$$

where $\xi$ represents a rescaled scalar field. As shown in [87], the classical part of the $\xi(t, z)$-series expansion identically vanishes and thus one gets

$$\xi(t, z) = \kappa^2 \left[ \tilde{\xi}_0^h(t) + \tilde{\xi}_1^h(t)(z - z_{EH}) + \tilde{\xi}_2^h(t)(z - z_{EH})^2 + \tilde{\xi}_3^h(z - z_{EH})^3 + \cdots \right], \tag{249}$$

where $z_{EH}$ denotes the location of the classical horizon. This expansion serves two purposes. Firstly, it is useful, as has just been mentioned, to demonstrate the trans-Planckian energy encountered by an infalling observer. Secondly, it allows one to examine the behavior of the solution near the horizon and thus the perfect-infall boundary condition that one typically imposes in the context of the quasi-normal modes. The leading-order energy comes from the scalar kinetic term in the stress-energy tensor—$\rho \equiv \partial_\mu \zeta \partial_\nu \zeta \, U^\mu U^\nu$, where $U^\mu$ denotes the four-velocity of the infalling observer. One can show that as $z \to z_{EH}^q$, with $z_{EH}^q$ denoting the quantum-corrected event horizon,

$$\rho \equiv \partial_\mu \zeta \partial_\nu \zeta \, U^\mu U^\nu \sim \frac{[\dot{\tilde{\xi}}_0^h(t)]^2}{\kappa^2}. \tag{250}$$

Note that it is the "horizon quantum mode" $\tilde{\xi}_0^h(t)$ that has led to this trans-Planckian energy. For a more realistic case, one should consider an Einstein–Maxwell case and investigate whether or not the energy density and Poynting vector reveal a similar behavior.

Finally, let us examine the behavior of the solution near the event horizon $z \sim z_{EH}$. Let us consider (249). With the quantum corrections, the $\tilde{\xi}_0^h$ mode is generically present (for a large class of boundary conditions) and this means that the boundary condition is such that there will be both transmitted and reflected waves. This shows that the quantum effects make the boundary condition deviate from the perfect-infall boundary condition. Therefore it will be of great interest to investigate various boundary conditions and how the associated physics departs from the quasi-normal mode physics.

**Funding:** This research received no external funding.

**Conflicts of Interest:** The author declares no conflict of interest.

## Abbreviations

The following abbreviations are used in this manuscript:

LQG　　Loop quantum gravity
BFM　　Background field method
ADM　　Arnowitt Deser Misner
LGT　　Large gauge transformation
BMS　　Bondi Metzner Sachs
GHY　　Gibbons–Hawking–York
AGN　　Active galactic nuclei

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
