# Peer review of "Foliation-Based Approach to Quantum Gravity and Applications to Astrophysics"

_universe, doi:10.3390/universe5030071_

Round 1
Reviewer 1 Report
Upon the first reading of the review (in the introduction section) there appears the impression that the new approach makes the general relativity to become a renormalizable theory. This is surely is not true. Even if the theory turns out to be asymptotically safe (numerical “proof” of this was claimed in the literature), it remains formally to be a non-renormalizable one. Moreover, the rigorous proof of renormalizability in all orders is absent. In the Introduction I would recommend to add a few words to state more definitely on what is the situation with the renoralizability within the author's approach.
In the rest I recommend to accept the review as it is.
Author Response
Dear Referee,
Thanks very much for your suggestions!
I have revised several places in the introduction (in particular the paragraph on the paper organization) for better clarity. The future task of the explicit two-loop renormalizability has been commented. Footnote 1 has been added.
I would like clarify several points here. At the same time I would like to invite the Referee to section 2. The claimed renormalizability concerns only the physical sector that has support on the holographic screen (see Fig. 1). In this sense, it is different from the full offshell renormalizability that people had tried to accomplish in the past. (My original papers do have detailed comments on this point.) As commented in section 2, the offshell non-renormalizability of a gravity theory, such as GR, is due to the appearance of the Riemann tensor in the counterterms. This was the observation by 't Hooft and 't Hooft-Veltman. My observation is that once one narrows down to the physical sector, the Riemann tensor becomes expressible in terms of the 3D Ricci tensor and metric, making the 't Hooft's program (of renormalization through the metric field redefinition) work again.
Finally, it would be of help if the Referee could point out the reference of the numerical proof of the asymptotic safety so that I can add it in the refs.

Reviewer 2 Report
In this review paper, the author mainly reviewed the foliation-based method (3+1 decomposition) for gravitational theory, including the Einstein gravity and Einstein-Maxwell gravity cases, with the emphasis on discussing the choice of Neumann boundary condition and the one-loop quantum effects and the renormalization of gravity.
I feel the paper is self-contained and clear. I only have one suggestion. In the abstract and the introduction, the author mentioned the gravity holography and the AdS/CFT correspondence. However, in the rest parts of the paper, the author didn't mention it again, I suggest the author to add a discussion on the relationship between the renormalization in Sec.5 and the renormalization studied from the holograhic perspective, i.e. the holograhpic renormalization.
Author Response
Thanks very much for the suggestion!
Tending to the suggestion, footnote 4 has been added with the corresponding ref. The last paragraph of section 5.4 has also been slightly revised.

Round 2
Reviewer 1 Report
Numerical "proof"
J. Laiho and D. Coumbe, “Evidence for asymptotic safety from
lattice quantum gravity,” Physical Review Letters, vol. 107, no. 16,
Article ID 161301, 2011.
Author Response
The ref has been added; it is ref[7] in the attached revised version. Thanks for the info!
